# Causal Effect Regularization: Automated Detection and Removal of Spurious Correlations

**Abhinav Kumar**
Microsoft Research
abhinavkumar.wk@gmail.com

**Amit Deshpande**
Microsoft Research
amitdesh@microsoft.com

**Amit Sharma**
Microsoft Research
amshar@microsoft.com

## Abstract

In many classification datasets, the task labels are *spuriously* correlated with some input attributes. Classifiers trained on such datasets often rely on these attributes for prediction, especially when the spurious correlation is high, and thus fail to generalize whenever there is a shift in the attributes' correlation at deployment. If we assume that the spurious attributes are known a priori, several methods have been proposed to learn a classifier that is invariant to the specified attributes. However, in real-world data, information about spurious attributes is typically unavailable. Therefore, we propose a method that automatically identifies spurious attributes by estimating their causal effect on the label and then uses a regularization objective to mitigate the classifier's reliance on them. Although causal effect of an attribute on the label is not always identified, we present two commonly occurring data-generating processes where the effect can be identified. Compared to recent work for identifying spurious attributes, we find that our method, AutoACER, is more accurate in removing the attribute from the learned model, especially when spurious correlation is high. Specifically, across synthetic, semi-synthetic, and real-world datasets, AutoACER shows significant improvement in a metric used to quantify the dependence of a classifier on spurious attributes ($\Delta$Prob), while obtaining better or similar accuracy. Empirically we find that AutoACER mitigates the reliance on spurious attributes even under noisy estimation of causal effects or when the causal effect is not identified. To explain the empirical robustness of our method, we create a simple linear classification task with two sets of attributes: causal and spurious. Under this setting, we prove that AutoACER only requires the *ranking* of estimated causal effects to be correct across attributes to select the correct classifier.

## 1 Introduction

When trained on datasets where the task label is spuriously correlated with some input attributes, machine learning classifiers have been shown to rely on these attributes (henceforth known as *spurious* attributes) for prediction [10, 23, 11]. For example in a sentiment classification dataset that we evaluate (Twitter-AAE [4]), a demographic attribute like race could be spuriously correlated with the sentiment of a sentence [9]. Classifiers trained on such datasets are at risk of failure during deployment when the correlation between the task label and the spurious attribute changes [2, 22, 30, 10].

Assuming that a set of auxiliary attributes is available at training time (but not at test time), several methods have been proposed to mitigate the classifier's reliance on the spurious attributes. The first category of methods assumes that the spurious attributes are known a priori. They develop regularization [35, 15, 22], optimization [30] or data-augmentation [17, 16, 14] strategies to train a classifier invariant to the specified spurious attributes. The second category of methods relaxes the assumption of known spurious attributes by automatically identifying the spurious attributes and

37th Conference on Neural Information Processing Systems (NeurIPS 2023).

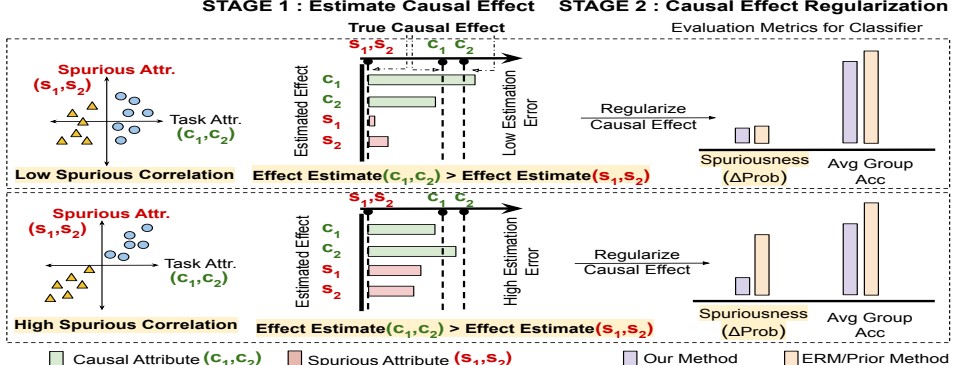

Figure 1: To reduce dependence on spurious attributes in a classifier, our method AutoACER has two stages. Stage 1 (middle panel) estimates the causal effect of attributes on the task label. Stage 2 (right panel) ensures that the causal effect of each attribute on the classifier's prediction matches its estimated causal effect on the task label. In the bottom panel, due to high spurious correlation the estimation of average causal effect is error-prone. AutoACER works well even in such high correlation scenarios as shown by the Spuriousness score, $\Delta$Prob (see definition in §4.2).

regularizing the classifier to be invariant to them. To identify spurious attributes, these methods impose assumptions on the type of spurious correlation they deal with. For example, they may assume that attributes are modified in input via symmetry transformation [25] or a group transformation [24], or the data-generating process follows a specific graph structure such as anti-causal [38]. However, all these methods assume a strict dichotomy—every attribute is either spurious or not—which makes them susceptible to imposing incorrect invariance whenever there is a mistake in identifying the spurious attribute.

In this paper, we propose a method that regularizes the effect of attributes on a classifier proportional to their *average causal effect* on the task label, instead of strictly binning them as spurious (or not) using an arbitrary threshold. We propose a two-step method that (**1**) uses an effect estimation algorithm to find the causal effect of a given attribute; (**2**) regularizes the classifier proportional to the estimated causal effect for each attribute estimated in the first step. We call the method *AutoACER*, Causal Effect Regularization for automated removal of spurious correlations. If the estimated causal effects are correct, AutoACER is perfect in removing a classifier's reliance on spurious attributes (i.e., attributes with causal effect close to zero), while retaining the classifier's acccuracy. But in practice it is difficult to have good estimates of the causal effect due to issues like non-identifiability, noise in the labels [39], or finite sample estimation error [3]. Under such conditions, we find that AutoACER is more robust than past work [25] since it does not group the attributes as spurious or causal and regularizes the classifier proportional to the estimated effect.

We analyze our method both theoretically and empirically. First, we provide the conditions under which causal effect is identified. An implication of our analysis is that causal effect is identified whenever the relationship between the attribute and label is fully mediated through the observed input (e.g, text or image). This is often the case in real-world datasets where human labellers use only the input to generate the label. Even if the causal effect is not identified, we show that AutoACER is robust to errors in effect estimation. Theoretically, on a simple classification setup with two sets of attributes—causal and spurious—we prove that only the correct ranking of the causal effect estimates is needed to learn a classifier invariant to spurious attributes. For the general case with multiple disentangled high-dimensional causal and spurious attributes, under the same condition on correct causal effect rankings, we prove that the desired classifier (that does not use spurious attributes) is preferred by AutoACER over the baseline ERM classifier.

To confirm our result empirically, we use three datasets that introduce different levels of difficulty in estimating the causal effect of attributes: (**1**) Syn-Text: a synthetic dataset that includes an unobserved confounding variable and violates the identifiability conditions of causal effect estimators; (**2**) MNIST34: a semi-synthetic dataset from [24], where we introduce noise in the image labels; (**3**) Twitter-AAE:, which is a real-world text dataset. To evaluate the robustness of methods to spurious correlation, we create multiple versions of each dataset with varying levels of correlation with the spurious attribute, thereby increasing the difficulty to estimate the causal effect. Even with a noisy

estimate of the causal effect of attributes, our method shows significant improvement over previous algorithms in reducing the dependence of a classifier on spurious attributes, especially in the high correlation regime. Our contributions include,

- Causal effect identifiability guarantee for two realistic causal data-generating processes, which is the first step to automatically distinguish between causal and spurious attributes.
- A method, AutoACER, to train a classifier that obeys the causal effect of attributes on the label. Under a simplified setting, we show that it requires only the correct ranking of attributes.
- Evaluation on three datasets showing that AutoACER is effective at reducing spurious correlation even under noisy, high correlation, or confounded settings.

## 2 OOD Generalization under Spurious Correlation: Problem Statement

For a classification task, let $(\boldsymbol{x}^i, y^i, \boldsymbol{a}^i)_{i=1}^n \sim \mathcal{P}_m$ be the set of example samples from the data distribution $\mathcal{P}$, where $\boldsymbol{x}^i \in \mathbf{X}$ are the input features, $y^i \in Y$ are the task labels and $\boldsymbol{a}^i = (a_1^i, \ldots, a_k^i)$, $a_j^i \in A_j$ are the auxiliary attributes, henceforth known as *attributes* for the example $\boldsymbol{x}^i$. We use "$\boldsymbol{x}_{a_j}$" to denote an example "$\boldsymbol{x}$" to have attribute $a_j \in A_j$. These attributes are only observed during the training time. The goal of the classification task referred to as *task* henceforth, is to predict the label $y^i$ from the given input $\boldsymbol{x}^i$. The task classifier can be written as $c(h(\boldsymbol{x}))$ where $h : \boldsymbol{X} \to \boldsymbol{Z}$ is an encoder mapping the input $\boldsymbol{x}$ to a latent representation $\boldsymbol{z} := h(\boldsymbol{x})$ and $c : \boldsymbol{Z} \to Y$ is the classifier on top of the representation $\boldsymbol{Z}$.

**Generalization to shifts in spurious correlation.** We are interested in the setting where the data-generating process is *causal* [31] i.e. there is a certain set of *causal* attributes that affect the task label $y$. Upon changing these causal attributes, the task label changes. Apart from these attributes, there could be other attributes defining the overall data-generating process (see Fig. 2 for examples). A attribute $c \in \mathcal{C}$ is called spurious attribute if it is correlated with the task label in the training dataset and thus could be used by a classifier trained to predict the task label, as a *shortcut* [38, 22]. But their correlation could change at the time of deployment, affecting the classifier's accuracy.

Using the attributes available at training time, our goal is to train a classifier $c(h(x))$ that is robust to shift in correlation between the task label and the spurious attributes. We use the fact that changing spurious attributes will not lead to a change in the task label i.e. they have zero causal effect on the task label $y$. Hence, we use the *estimated causal effect* of an attribute to automatically identify its degree of spuriousness. For a *spurious attribute*, its true causal effect on the label is zero and hence the goal is to ensure its causal effect on the classifier's prediction is also zero. More generally, we would like to regularize the effect of each attribute on the classifier's prediction to its causal effect on the label. In other words, unlike existing methods that aim to discover a subset of attributes that are spurious [24, 25, 38], we aim to estimate the causal effect of each attribute and match it. Since our method avoids hard categorization of attributes that downstream regularizers need to follow, we show that is a more robust way of handling estimation errors when spurious correlation may be high.

## 3 Causal Effect Regularization: Minimizing reliance on spurious correlation

We now describe our method to train a classifier that generalizes to shift in attribute-label correlation by automatically identifying and imposing invariance w.r.t to spurious attributes. In §3.1, we provide sufficient conditions for identifying the causal effect, a crucial first step to detect the spurious attributes. Next, in §3.2 and §3.3, we present the AutoACER method and its theoretical analysis.

### 3.1 Causal Effect Identification

Since our method relies on estimating the causal effect of attributes on the task label, we first show that the attributes' causal effect is *identified* for many commonly ocurring datasets. The identifiability of the causal effect of an attribute depends on the data-generating process (DGP). Below we present two illustrative DGPs (DGP-1 and DGP-2 from Fig. 2) under which the causal effect is identified. *DGP-1* refers to the case where the task label is generated based on the observed input $X$ which in turn may be caused by observed and unobserved attributes. *DGP-1* is common in many real-world datasets where the task labels are annotated based on the observed input $X$ either automatically using some deterministic function or using human annotators [17, 29]. Thus *DGP-1* is applicable for all

the settings where the input $X$ has all the sufficient information for creating the label. Mouli and Ribeiro [25] consider another DGP where a set of *transformations* (like rotation or vertical-flips in image) over a base (unobserved) image $x$ generates the input $X$. We adapt their graph to our setting where we include each observed attribute as a node in the graph, along with hidden confounders. Depending on whether they are spurious or not, these attributes may cause the task label $Y$. DGP-2 and DGP-3 represent two such adaptations from their work, with the difference that DGP-3 also allows unobserved attributes to cause the task label.

Generally for identifying the causal effect one assumes that we only have access to observational data ($\mathcal{P}$). But here we also assume access to the interventional distribution for the input, $P(X|do(A))$ where the attribute $A$ is set to a particular value, as in [24]. This is commonly available in vision datasets via data augmentation strategies over attributes like rotation or brightness, and also in text datasets using generative language models [5, 37]. Having access to interventional distribution $P(X|do(A))$ could help us identify the causal effect in certain cases where observational data ($\mathcal{P}$) alone cannot as we see below.

**Proposition 3.1.** *Let DGP-1 and DGP-2 in Fig. 2 be the causal graphs of two data-generating processes. Let $A, C, S$ be different attributes, $X$ be the observed input, $Y$ be the observed task label and $U$ be the unobserved confounding variable. In DGP-2, $x$ is the (unobserved) core input feature that affects the label $Y$. Then:*

1. **DGP-1 Causal Effect Identifiability:** *Given the interventional distribution $P(X|do(A))$, the causal effect of the attribute $A$ on task label $Y$ is identifiable using observed data distribution.*

2. **DGP-2 Causal Effect Identifiability:** *Let $C$ be a set of observed attributes that causally affect the task label $Y$ (unknown to us), $S$ be the set of observed attributes spuriously correlated with task label $Y$ (again unknown to us), and let $\mathcal{V} = C \cup S$ be the given set of all the attributes. Then if all the causal attributes are observed then the causal effect of all the attributes in $\mathcal{V}$ can be identified using observational data distribution alone.*

*Proof Sketch.* **(1)** We show that we can identify the interventional distribution $P(Y|do(A))$ which is needed to estimate the causal effect of $A$ on $Y$ using the given observational distribution $P(Y|X)$ and interventional distribution $P(X|do(A))$. **(2)** For both causal attribute $C$ and spurious attribute $S$, we show that we can identify the interventional distribution $P(Y|do(S))$ or $P(Y|do(C))$ using purely observational data using the same identity without the need to know whether the variable is causal or spurious a priori. See §B for proof. □

Note that *DGP-1* and *DGP-2* are not exhaustive. We provided them simply as illustrative examples to show that there exist DGPs, corresponding to commonly occurring datasets, where the attributes' effect is identified. However, the effect is not identified in *DGP-3* and possibly other DGPs. Moreover, in practice, the underlying DGP for a given task may not be known or mis-specified. Hence, in §3.2, we design our method AutoACER such that it does not depend on knowledge of the true DGP. When the effect is identified, our method is theoretically guaranteed to work well. Empirically, as we show in Section 4, our method also works to remove the spurious correlation for datasets corresponding to DGP-3 where the causal effect of A on Y is not identified. In comparison, prior methods like Mouli and Ribeiro [25] provably fail under DGP-3 (see §C for the proof that their method would fail to detect the spurious attribute).

### 3.2 AutoACER: Causal Effect Regularization for predictive models

Our proposed method proceeds in two stages. In the first stage, it uses a causal effect estimator to identify the causal effect of an attribute on the task label. Then in the second stage, it regularizes the classifier proportional to the causal effect of every attribute.

**Stage 1: Causal Effect Estimation.** Given a set of attributes $\mathcal{A} = \{a_1, \ldots, a_k\}$, the goal of this step is to estimate the causal effect ($TE_{a_i}$) of every attribute $a \in \mathcal{A}$. The causal effect of a attribute on the label $Y$ is defined as the expected change in the label $Y$ when we change the attribute (see §A for formal definition). If the data-generating process of the task is one of DGP-1 or DGP-2, our Prop 3.1 gives us sufficient conditions needed to identify the causal effect. Then one can use appropriate causal effect estimators that work under those conditions or build their own estimators using the closed form causal effect estimand given in the proof of Prop 3.1. However, to build a

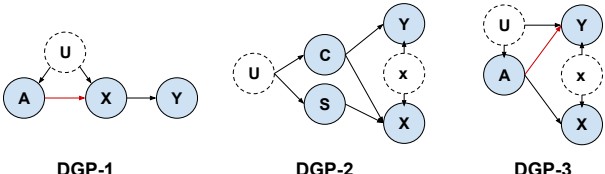

**DGP-1**          **DGP-2**          **DGP-3**

Figure 2: Causal graphs showing different data-generating processes. Shaded nodes denote observed variables; the red arrow denotes an unknown target relationship. Node X denotes the observed input features and node Y denotes the task label in all the DGPs. Node A refers to attributes; could denote either causal or spurious attribute. If A is a causal attribute then the red arrow will be present in the underlying true graph otherwise not. Node U denotes the unobserved confounding variable which could be the potential reason for spurious correlation between an attribute (A or S) and Y. The node C in DGP-2 denotes causal attributes and node S denotes spuriously correlated attributes. The node x in DGP-2 and DGP-3 denotes the unobserved causal attribute that creates the label Y along with other observed causal attributes (C or A). In the first two graphs, the causal effect of attributes (A and C, S respectively) on Y is identified (see Prop 3.1).

general method, we propose a technique that does not assume knowledge of the DGP. Specifically, we use a conditioning-based estimator (based on the backdoor criterion [27]) and assume the observed variables (X) as the backdoor variables.    There is a rich literature on estimating causal effects for high dimensional data [12, 33, 6]. We use a deep learning-based estimator from Chernozhukov et al. [6] to estimate the causal effect (henceforth called *Riesz*). Given the treatment along with the rest of the covariates, it learns a common representation that approximates backdoor adjustment to estimate the causal effect (see §E for details). Even if the causal effect in the relevant dataset is identifiable, we might get a noisy estimate of the effect due to finite sample error or noise in the labels. Later in §3.3 and §4 we will show that our method is robust to error in the causal effect estimate of attributes both theoretically and empirically. Finally, as a baseline effect estimator, we use the direct effect estimator (henceforth called *Direct*) defined as $\mathbb{E}_X(\mathbb{E}(Y|X, a = 1) - \mathbb{E}(Y|X, a = 0))$ for attribute $a \in \mathcal{A}$ that has limited identifiability guarantees (see §E for details).

**Stage2: Regularization.** Here our method regularizes the model prediction with the estimate of causal effect $= \{TE_{a_1}, \ldots, TE_{a_k}\}$ of each attribute $\mathcal{A} = \{a_1, \ldots, a_k\}$. The loss objective is,

$$\mathcal{L}_{AutoACER} := \mathbb{E}_{(\boldsymbol{x},y)\sim\mathcal{P}}\Big[\mathcal{L}_{task}\Big(c(h(\boldsymbol{x})), y\Big) + R \cdot \mathcal{L}_{Reg}\Big(\boldsymbol{x}, y\Big)\Big] \tag{1}$$

where $R$ is the regularization strength hyperparameter, $\mathcal{P}$ is the training data distribution (§2). The first term $\mathcal{L}_{task}(c(h(\boldsymbol{x})), y)$ can be any training objective e.g. cross-entropy or max-margin loss for training the encoder $h$ and task classifier $c$ jointly to predict the task label $y$ given input $\boldsymbol{x}$. Our regularization loss term $\mathcal{L}_{Reg}$ aims to regularize the model such that the causal effect of an attribute on the classifier's output matches the estimated causal effect of the attribute $a_i$ on the label. Formally,

$$\mathcal{L}_{Reg} := \sum_{i=\{1,2,\ldots,|\mathcal{A}|\}} \mathbb{E}_{(\boldsymbol{x}_{a_i'})\sim\mathcal{Q}(\boldsymbol{x}_{a_i})}\Big[\Big(c(h(\boldsymbol{x})_{a_i}) - c(h(\boldsymbol{x}_{a_i}'))\Big) - TE_{a_i}\Big]^2 \tag{2}$$

where $\boldsymbol{x}_{a_i'} \sim \mathcal{Q}(\boldsymbol{x}_{c_i})$ be a sample from counterfactual distribution $\mathcal{Q} := \mathbb{P}(\boldsymbol{x}_{c_i'}|\boldsymbol{x}_{c_i})$ and $\boldsymbol{x}_{c_i'}$ is the input had the attribute in input $\boldsymbol{x}_{c_i}$ been $c_i'$.

### 3.3 Robustness of AutoACER with noise in the causal effect estimates

Our proposed regularization method relies primarily on the estimates of the causal effect of any attribute $c_i$ to regularize the model. Thus, it becomes important to study the efficacy of our method under error or noise in causal effect estimation, which is expected in real-world tasks due to finite sample issues, incorrect choice of causal effect estimators, or due to properties of the unknown underlying DGP. We consider a simple setup to theoretically analyze the condition under which our method will train a better classifier than the standard ERM max-margin objective (following previous work [19, 13, 26]) in terms of generalization to the spurious correlations. Let $\mathcal{A} = \{ca_1, \ldots, ca_K, sp_1, \ldots, sp_J\}$ be the set of available attributes where $ca_k$ are causal attributes and $sp_j$ are the spurious attributes. For simplicity, we assume that the representation encoder mapping $\boldsymbol{X}$ to $\boldsymbol{Z}$ i.e $h : \boldsymbol{X} \to \boldsymbol{Z}$ is frozen and the final *task* classifier ($c$) is a linear operation over the representation. Following Kumar et al. [19], we also assume that $\boldsymbol{z}$ is a disentangled representation

w.r.t. the causal and spurious attributes, i.e., the representation vector $z$ can be divided into two subsets, features corresponding to the causal and spurious attribute respectively. Thus the task classifier takes the form $c(z) = \sum_{k=1}^{K} w_{ca_k} \cdot z_{ca_k} + \sum_{j=1}^{J} w_{sp_j} \cdot z_{sp_j}$. Note that we make the disentanglement assumption only for creating a simple setup so that we can theoretically study the effectiveness of our method in the presence of noisy estimates of the causal effects. Our method does not require this assumption.

Let $\mathcal{L}_{task}(\theta; (x, y_m))$ be the max-margin objective (see §A for details) used to train the task classifier "$c$" to predict the task label $y_m$ given the *frozen* latent space representation $z$. Let the task label $y$ and the attribute "$a$" where $a \in \mathcal{A}$ be binary taking value from $\{0, 1\}$. The causal effect of an attribute $a \in \mathcal{A}$ on the task label $Y$ is given by $TE_a = \mathbb{E}(Y|do(a) = 1) - \mathbb{E}(Y|do(a) = 0) = P(y = 1|do(a) = 1) - P(y = 1|do(a) = 0)$ (see §A for definition). Thus the value of the causal effect is bounded s.t. $\hat{TE}_a \in [-1, 1]$ where the ground truth causal effect of spuriously correlated attribute "$sp$" is $TE_{sp} = 0$ and for causal attribute "$ca$" is $|TE_{ca}| > 0$. Given that we assume a linear model, we instantiate a simpler form of the regularization term $\mathcal{L}_{Reg}$ for the training objective given in Eq. 1:

$$\mathcal{L}_{Reg} := \sum_{k=\{1,2,...,K\}} \lambda_{ca_k} \|w_{ca_k}\|_p + \sum_{j=\{1,2,...,J\}} \lambda_{sp_j} \|w_{sp_j}\|_p \qquad (3)$$

where $\lambda_{ca_k} := 1/|TE_{ca_k}|$ and $\lambda_{sp_j} := 1/|TE_{sp_j}|$ are the regularization strength for the causal and spurious features $z_{ca_k}$ and $z_{sp_j}$ respectively, $|\cdot|$ is the absolute value operator and $\|w\|_p$ is the $L_p$ norm of the vector $w$. Since $|TE_{(\cdot)}| \in [0, 1]$, we have $\lambda_{(\cdot)} = 1/|TE_{(\cdot)}| \geq 1$. In practice, we only have access to the empirical estimate of the causal effect $TE_{(\cdot)}$ denoted as $\hat{TE}_{(\cdot)}$ and the regularization coefficient becomes $\lambda_{(\cdot)} = 1/|\hat{TE}_{(\cdot)}|$. Now we are ready to state the main theoretical result that shows our regularization objective will learn the correct classifier which uses only the causal attribute for its prediction, given that the ranking of estimated treatment effect is correct up to some constant factor. Let $[S]$ denote the set $\{1, \ldots, S\}$.

**Theorem 3.1.** *Let the latent space be frozen and disentangled such that* $z = [z_{ca_1}, \ldots, z_{ca_K}, z_{sp_1}, \ldots, z_{sp_J}]$ *(Assm D.1). Let the desired classifier* $c^{des}(z) = \sum_{k=1}^{K} w_{ca_k}^{des} \cdot z_{ca_k}$ *be the max-margin classifier among all linear classifiers that use only the causal features* $z_{ca_k}$*'s for prediction. Let* $c^{mm}(z) = \sum_{k=1}^{K} w_{ca_k}^{mm} \cdot z_{ca_k} + \sum_{j=1}^{J} w_{sp_j}^{mm} \cdot z_{sp_j}$ *be the max-margin classifier that uses both the causal and the spurious features, and let* $w_{sp_j}^{mm} \neq 0, \forall j \in [J]$. *We assume* $w_{sp_j}^{mm} \neq 0$, $\forall j \in [J]$, *without loss of generality because otherwise, we can restrict our attention only to those* $j \in [J]$ *that have* $w_{sp_j}^{mm} \neq 0$. *Let the norm of the parameters of both the classifier be set to 1 i.e* $\sum_{k=1}^{K} \|w_{ca_k}^{mm}\|_{p=2}^2 + \sum_{j=1}^{J} \|w_{sp_j}^{mm}\|_{p=2}^2 = \sum_{k=1}^{K} \|w_{ca_k}^{des}\|_{p=2}^2 = 1$. *Then if regularization coefficients are related s.t.* $\text{mean}\left(\left\{\frac{\lambda_{ca_k}}{\lambda_{sp_j}} \cdot \eta_{k,j}\right\}_{k \in [K], j \in [J]}\right) < \frac{J}{K}$ *where* $\eta_{k,j} = \frac{\|w_{ca_k}^{des}\| - \|w_{ca_k}^{mm}\|_p}{\|w_{sp_j}^{mm}\|_p}$, *then*

1. **Preference:** $\mathcal{L}_{AutoACER}(c^{des}(z)) < \mathcal{L}_{AutoACER}(c^{mm}(z))$. *Thus, our causal effect regularization objective (Def 3) will choose the* $c^{des}(z)$ *classifier over the max-margin classifier* $c^{mm}(z)$ *which uses the spuriously correlated feature.*

2. **Global Optimum:** *The desired classifier* $c^{des}(z)$ *is the global optimum of our loss function* $\mathcal{L}_{AutoACER}$ *when* $J = 1$, $K = 1$, $p = 2$, *the regularization strength are related s.t.* $\lambda_{ca_1} < \lambda_{sp_1} \implies |\hat{TE}_{ca_1}| > |\hat{TE}_{sp_1}|$ *and search space of linear classifiers* $c(z)$ *are restricted to have the norm of parameters equal to 1.*

**Remark.** *The result of Theorem 3.1 holds under a more intuitive but stricter constraint on the regularization coefficient* $\lambda$ *which states that* $\lambda_{sp_j} > \left(\frac{K}{J}\eta_{k,j}\right)\lambda_{ca_k} \implies |\hat{TE}_{sp_j}| < \left(\frac{K}{J}\eta_{k,j}\right)|\hat{TE}_{ca_k}|$ $\forall k \in [K]$ *and* $j \in [J]$. *The above constraint states that if the treatment effect of the causal feature is more than that of the spurious feature by a constant factor then the claims in Theorem 3.1 hold.*

*Proof Sketch.* **(1)** We compare both the classifier $c^{des}(z)$ and $c^{mm}(z)$ using our overall training objective to our training objective $\mathcal{L}_{AutoACER}$ (Eq. 1). Given the relation between the regularization strength mentioned in the above theorem is satisfied, we then show that one can always choose a regularization strength "$R$" greater than a constant value s.t classifier $c^{mm}(z)$ incurs a greater

regularization penalty than $c^{des}(z)$ (Eq. 3) which is not compensated by the gain in the max-margin objective of $c^{mm}(z)$ over $c^{des}(z)$. Thus, the desired classifier $c^{des}(z)$ has lower overall loss than the $c^{mm}(z)$ in terms of our training objective $\mathcal{L}_{AutoACER}$. **(2)** We use the result from the first claim to show that $c^{des}(z)$ has a lower loss than any other classifier that uses the spurious feature $z_{sp_1}$. Then, among the classifier that only uses the causal feature $z_{ca_1}$, we show that again $c^{des}(z)$ has the lower loss w.r.t. $\mathcal{L}_{AutoACER}$. Thus the desired classifier has a lower loss than all other classifiers w.r.t $\mathcal{L}_{AutoACER}$ with parameter norm 1 hence a global optimum. Refer §D for proof.

# 4 Empirical Results

## 4.1 Datasets

Theorem 3.1 showed that our method can find the desired classifier in a simple linear setup. We now evaluate the method on a synthetic, semi-synthetic and real-world dataset. Details are in §E.

**Syn-Text.** We introduce a synthetic dataset where the ground truth causal graph is known and thus the ground truth spuriously correlated feature is known apriori (but unknown to all the methods). The dataset contains two random variables (*causal* and *confound*) that cause the binary main task label $y_m$ and the variable *spurious* is spuriously correlated with the *confound* variable. Given the values of spurious and causal features, we create a sentence as input. We define Syn-Text-Obs-Conf– a version of the Syn-Text dataset where all three variables/attributes are observed. Next, to increase difficulty, we create a version of this dataset — Syn-Text-Unobs-Conf— where the *confound* attribute is not observed in the training dataset, corresponding to DGP-3 from Fig. 2.

**MNIST34.** We use MNIST [20] to compare our method on a similar semi-synthetic dataset as used in Mouli and Ribeiro [25]. We define a new task over this dataset but use the same attributes, color, rotation, and digit, whose associated transformation satisfies the *lumpability* assumption in their work. We define the digit attribute (3 and 4) and the color attribute (red or green color of digit) as the causal attributes which creates the binary main task label using the $XOR$ operation. Then we add rotation ($0^{\circ}$ or $90^{\circ}$) to introduce spurious correlation with the main task label. This dataset corresponds to DGP-2, where $C$ is color and digit attribute, $S$ is rotation attribute and $x$ is an empty set.

**Twitter-AAE [4].** This is a real-world dataset where the main task is to predict a binary sentiment label from a tweet's text. The tweets are associated with *race* of the author which is spuriously correlated with the main task label. Since this is a real-world dataset where we have not artificially introduced the spurious attribute we don't have a ground truth causal effect of *race* on the sentiment label. But we expect it to be zero since changing *race* of a person should not change the sentiment of the tweet. We use GPT3 [5] to create the counterfactual example by prompting it to change the race-specific information in the given sentence (see §F for examples). This dataset corresponds to DGP-1, where the node $A$ is the spurious attribute race.

**Varying spurious correlation in the dataset.** Since the goal is to train a model that doesn't rely on the spuriously correlated attribute, we create multiple settings for every dataset with different levels of correlations between the main task labels $y_m$ and spurious attribute $y_{c_{sp}}$. Following [19], we use a *predictive correlation* metric to define the label-attribute correlation that we vary in our experiments. The predictive correlation ($\kappa$) measures how informative one label or attribute ($s$) is for predicting the other ($t$), $\kappa := Pr(s = t) = \sum_{i=1}^{N} \mathbf{1}[s = t]/N$, where $N$ is the size of the dataset and $\mathbf{1}[\cdot]$ is the indicator function that is 1 if the argument is true otherwise 0. Without loss of generality, assuming $s = 1$ is correlated with $t = 1$ and similarly, $s = 0$ is correlated with $t = 0$; predictive correlation lies in $\kappa \in [0.5, 1]$ where $\kappa = 0.5$ indicates no correlation and $\kappa = 1$ indicates that the attributes are fully correlated. For Syn-Text dataset, we vary the predictive correlation between the confound attribute and the spurious attribute; for MNIST34, between the combined causal attribute (digit and color) and the spurious attribute (rotation); and for Twitter-AAE, between the task label and the spurious attribute (race). See §E for details.

## 4.2 Baselines and evaluation metrics

**Baselines.** We compare our method with five baseline algorithms: (i) *ERM*: Base algorithm for training the main task classifier using cross-entropy loss without any additional regularization. (ii) *Mouli+CAD*: The method proposed in [25] to automatically detect the spurious attributes and

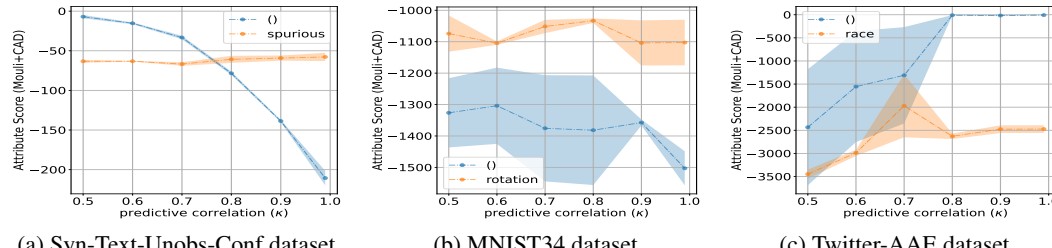

(a) Syn-Text-Unobs-Conf dataset     (b) MNIST34 dataset     (c) Twitter-AAE dataset

Figure 3: **Detecting spurious attribute using Mouli+CAD:** x-axis is the predictive correlation and y-axis shows the score defined by Mouli+CAD. The orange curve shows the score for true spurious attribute. The blue curve shows the score for setting when Mouli+CAD considers none of the given attributes as spurious. The attribute with the lowest score is selected as spurious. If the blue curve has the lowest score for a dataset with predictive correlations ($\kappa$), Mouli+CAD will not declare any attributes as spurious for that $\kappa$. **(a)** and **(b).** Mouli+CAD fails to detect the spurious attribute at high $\kappa$. **(c)** Mouli+CAD correctly identifies the race attribute as spurious.

train a model invariant to those attributes. Given a set of attributes, this method computes a score for every subset of attributes, selects the subset with a minimum score as the spurious subset, and finally enforces invariance with respect to those subsets using counterfactual data augmentation (CAD) [32, 16]. (iii) *Mouli+AutoACER* : Empirically, we observe that CAD does not correctly impose invariance for a given attribute (see §F for discussion). Thus we add a variant of Mouli+CAD's method where instead of using CAD it uses our regularization objective (Eq. 2) to impose invariance using the causal effect $TE_a = 0$ for some attribute "$a$". (iv) *Just Train Twice* [21] (*JTT*): Given a known spurious attribute, *JTT* aims to train a model that performs well on all subgroups of the dataset, where subgroups are defined based on different correlations between the task label and spurious attribute. Unlike methods like GroupDRO [30], *JTT* requires (expensive) subgroup labels only for the examples in the validation set. (v) *Invariant Risk Minimization [2]* (*IRM*): This method requires access to multiple *environments* or subsets of the dataset where the correlation between the spurious attribute and task label changes and then it trains a classifier that only uses the attributes with stable correlations across environments (see E for experimental setup).

**Metrics.** We use two metrics for evaluation. Since all datasets have binary task labels and attributes, we define a group-based metric (*average group accuracy*) to measure generalization under distribution shift. Specifically, given binary task label $y \in \{0, 1\}$ and spurious attribute $a \in \{0, 1\}$. Following Kumar et al. [19] we define $2 \times 2$ groups, one for each combination of $(y, a)$. The subset of the dataset with $(y = 1, a = 1)$ and $(y = 0, a = 0)$ are the majority group $S_{maj}$ while groups $(y = 1, a = 0)$ and $(y = 0, a = 1)$ make up the minority group $S_{min}$. We expect the main classifier to exploit this correlation and hence perform better on $S_{maj}$ but badly on $S_{min}$ where the correlation breaks. Thus we want a method that performs better on the average accuracy on both the groups i.e $\frac{Acc(S_{min}) + Acc(S_{maj})}{2}$, where $Acc(S_{maj})$ and $Acc(S_{min})$ are the accuracy on majority and minority group respectively. The second metric ($\Delta$Prob) measures the reliance of a classifier on the spurious feature. For every given input $\boldsymbol{x}_a$ we have access to the counterfactual distribution $(\boldsymbol{x}_{a'}) \sim \mathcal{Q}(\boldsymbol{x}_a)$ (§2) where the attribute $A = a$ is changed to $A = a'$. $\Delta$Prob is defined as the change in the prediction probability of the model on changing the spurious attribute $a$ in the input, thus directly measuring the reliance of the model on the spurious attribute. For background on baselines refer §A a detailed description of our experiment setup refer §E.

### 4.3 Evaluating Stage 1: Automatic Detection of Spurious Attributes

**Failure of Mouli+CAD in detecting the spurious attributes at high correlation.** In Fig. 3, we test the effectiveness of Mouli+CAD to detect the subset of attributes which are spurious on different datasets with varying levels of spurious correlation ($\kappa$). In Syn-Text dataset, at low correlations ($\kappa < 0.8$) Mouli+CAD correctly detects *spurious* attribute (orange line is lower than blue). As the correlation increases, their method incorrectly states that there is no spurious attribute (blue line lower than orange). In MNIST34 dataset, Mouli+CAD does not detect any attributes as spurious (shown by blue line for all $\kappa$). For Twitter-AAE dataset, Mouli+CAD's method is correctly able to detect the spuriously correlated attribute (*race*) for all the values of predictive correlation, perhaps because the spurious correlation is weak compared to the causal relationship from causal features to task label.

Table 1: **Causal Effect Estimate of spurious attribute**. *Direct* and *Riesz* are two different causal effect estimators as described in §3.2. Overall we see that *Riesz* performs better or comparable to the baseline *Direct* and closer to ground truth value 0. Since Twitter-AAE is a real dataset we don't have a ground causal effect of *race* attribute but we expect it to be zero (see §4.1).

| Dataset | True | Method | Predictive Correlation ($\kappa$) | | | | | |
| | | | 0.5 | 0.6 | 0.7 | 0.8 | 0.9 | 0.99 |
|---|---|---|---|---|---|---|---|---|
| Syn-Text-Obs-Conf | 0 | Direct | **0.00** | **0.00** | **0.01** | 0.08 | 0.16 | 0.22 |
| | | Riesz | **0.00** | 0.03 | 0.03 | **0.03** | **0.05** | **0.15** |
| Syn-Text-Unobs-Conf | 0 | Direct | **-0.01** | **0.13** | **0.24** | **0.37** | **0.49** | **0.67** |
| | | Riesz | 0.06 | 0.19 | 0.31 | 0.42 | 0.58 | 0.70 |
| MNIST34 | 0 | Direct | 0.02 | **0.03** | **0.05** | 0.06 | **0.15** | **0.27** |
| | | Riesz | **-0.01** | 0.06 | **0.05** | **0.04** | 0.2 | 0.31 |
| Twitter-AAE | - | Direct | **0.01** | **0.07** | **0.12** | 0.20 | **0.27** | **0.31** |
| | | Riesz | 0.02 | **0.07** | **0.12** | **0.17** | **0.27** | 0.37 |

(a) Syn-Text-Unobs-Conf     (b) MNIST34     (c) Twitter-AAE

Figure 4: **Average group accuracy (top row) and $\Delta$Prob (bottom row) for different methods.** x-axis denotes predictive correlation ($\kappa$) in the dataset. Compared to other baselines, **(a)** and **(b)** show that AutoACER performs better or comparable in average group accuracy and trains a classifier significantly less reliant to spurious attribute as measured by $\Delta$Prob. In **(a)**, IRM has lower $\Delta$Prob as compared to other methods but performs worse on average group accuracy. Mouli+CAD performs better in average group accuracy in **(c)** but relies heavily on spurious attribute shown by high $\Delta$Prob whereas AutoACER performs equivalent to all other baselines on average group accuracy with lowest $\Delta$Prob among all.

**AutoACER is robust to error in the estimation of spurious attributes.** Unlike Mouli+CAD's method that does a hard categorization, AutoACER estimates the causal effect of every attribute on the task label as a fine-grained measure of whether an attribute is spurious or not. Table 1 summarizes the estimated treatment effect of spurious attributes in every dataset for different levels of predictive correlation ($\kappa$). We use two different causal effect estimators named *Direct* and *Riesz* with the best estimate selected using validation loss (see §E). Overall, the Riesz estimator gives a better or comparable estimate of the causal effect of spurious attribute than Direct, except in the Syn-Text-Unobs-Conf dataset where the causal effect is not identified. At high predictive correlation($>= 0.9$), as expected, the causal effect estimates are incorrect. But as we will show next, since AutoACER uses a continuous effect value to detect the spurious attribute, it allows for errors in the first (detection) step to not affect later steps severely.

## 4.4 Evaluating Stage 2: Evaluation of AutoACER and other baselines

Fig. 4 compares the efficacy of our method with other baselines in removing the model's reliance on spurious attributes. On **Syn-Text-Unobs-Conf** dataset, our method (AutoACER) performs better than all the other baselines for all levels of predictive correlation ($\kappa$) on *average group accuracy*

(first row in Fig. 4a). In addition, $\Delta$Prob (the sensitivity of model on changing spurious attribute in input, see 4.1) for our method is the lowest compared to other baselines (see bottom row of Fig. 4a). For $\kappa \geq 0.8$, Mouli+CAD is the same as ERM since it fails to detect the spurious attribute and thus doesn't impose invariance w.r.t the spurious attribute. (Fig. 3a for details). On **MNIST34** dataset, the average group accuracy of all methods is comparable (except JTT that performs worse), but AutoACER has a substantially lower $\Delta$Prob than baselines for all values of $\kappa$. Again, the main reason why Mouli+CAD fails is that it is not able to detect the spurious attribute for all $\kappa$ and thus doesn't impose invariance w.r.t them (see Fig. 3b and §4.3). On **Twitter-AAE** dataset, Mouli+CAD correctly detects the *race* attribute as spurious and performs better in terms of Average Group Accuracy compared to all other baselines. But if we look at $\Delta$Prob, the gain in accuracy is not because of better invariance: in fact, the reliance on the spurious attribute (race) is worse than ERM. In contrast, AutoACER has lowest $\Delta$Prob among all while obtaining comparable accuracy to ERM, JTT and IRM. To summarize, in all datasets, AutoACER ensures a higher or comparable accuracy to ERM while yielding the lowest $\Delta$Prob. Note that AutoACER does not require knowledge of the spurious attribute whereas JTT and IRM do; hence we selected the best model using overall accuracy on the validation set for all the methods (see Eq. 33). For a fair comparison to JTT and IRM, in §F.2 (Fig. 7) we show results under the setting where all the methods (including AutoACER) assume that the spurious attribute is known. Even with access to this additional information, we observe a similar trend as in Fig. 4. AutoACER performs equivalent to all other baselines on average group accuracy with the lowest $\Delta$Prob among all. Finally, in §F we provide an in-depth analysis of our results, provide recommendations for selecting among different causal effect estimators used by our method (§F.1) and evaluate AutoACER and other baselines on an additional dataset —Civil-Comments [18]—a popular dataset for studying generalization under spurious correlation (§F.2).

## 5   Related Work

**Known spurious attributes.**   When the spuriously correlated attributes are known a priori, three major types of methods have been proposed, based on worst-group optimization [30], conditional independence constraints [35, 15, 22], or data augmentation [16, 17]. Methods like GroupDRO [30] create multiple groups in the training dataset based on the spurious attribute and optimize for worst group accuracy. To reduce labelling effort for the spurious attribute, methods like JTT [21] and EIIL [7] improve upon GroupDRO by requiring the spurious attribute labels only on the validation set. Other methods assume knowledge of causal graphs and impose conditional independence constraints for removing spurious attributes [35, 15, 22]. Methods based on data augmentation add counterfactual training data where only the spurious attribute is changed (and label remains the same) [16, 14, 17].

**Automatically discovering spurious attributes.**   The problem becomes harder if spurious attributes are not known. Mouli and Ribeiro's work [24, 25] provides a method assuming specific kinds of transformations on the spurious attributes, either transformations that form finite linear automorphism groups [24] or symmetry transformations over equivalence classes [25]. Any attribute changed via the corresponding transformation that does not hurt the training accuracy is considered spurious. However, they do not consider settings with correlation between the transformed attributes and the task labels. Our work considers a more realistic setup where we don't impose any constraint on the transformation or the attribute values, and allows attributes to be correlated with the task label (at different strengths). Using conditional independencies derived from a causal graph, Zheng and Makar [38] propose a method to automatically discover the spurious attributes under the anti-causal classification setting. Our work, considering the *causal* classification setting (features cause label) complements their work and allows soft regularization proportional to the causal effect.

## 6   Limitations and Conclusion

We presented a method for automatically detecting and removing spurious correlations while training a classifier. While we focused on spurious attributes, estimation of causal effects can be used to regularize the effect of non-spurious features too. That said, our work has limitations: we can guarantee the identification of causal effects only in certain DGPs. In future work, it will be useful to characterize the datasets on which our method is likely to work and where it fails. Another limitation is that our method is not guaranteed to work on datasets with an anti-causal data-generating process, wherein the observed input $x$ and other attributes are generated from the task label $y$.

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

# A Mathematical Preliminaries

## A.1 Causal Effect Estimation and RieszNet Estimator

Let $(X, Y, A)$ denote a random input point from the data distribution, where $A$ is the auxiliary attribute label and $X$ is the input covariate and $Y$ is the task label. Assuming binary $A$, the average causal effect (or equivalently called as average treatment effect) of attribute $A$ on the task label $Y$ is defined as (Chapter 3 in Pearl [27]):

$$TE_A = \mathbb{E}[Y|do(A) = 1] - \mathbb{E}[Y|do(A) = 0] \tag{4}$$

Given $X$ satisfies sufficient backdoor adjustment the causal effect can be estimated using (see Chapter 3 in Pearl [27] for details):

$$Direct := TE_A = \mathbb{E}_X\Big[\mathbb{E}[Y|X, A = 1] - \mathbb{E}[Y|X, A = 0]\Big] \tag{5}$$

Henceforth, we will refer to the above estimate of causal effect as *Direct*. Often in practice, the backdoor adjustment variable $X$ is high dimensional (e.g., vector encoding of text or images), and as a result, it becomes difficult to estimate the above quantity. Thus, several methods have been proposed to efficiently estimate and debias the causal effect for high dimensional data [6, 33]. In this paper, we use one of the recently proposed methods [6], henceforth denoted by *Riesz*, that uses insights from the Riesz representation theorem to create a multitasking neural network-based method to get a debiased estimate of the causal effect. Since this method uses a neural network to perform estimation, it is desirable for many data modalities such as text and image that require a neural network to get a good representation of the input. Let $g(X, A) := \mathbb{E}[Y|X, A]$, then the estimator used by *Riesz* is given by:

$$TE_A = \mathbb{E}[\alpha_0(X, A) \cdot g(X, A)] \tag{6}$$

where $\alpha_0(X, A)$ is called a Riesz representer (RR), whose existence is guaranteed by the Riesz representation theorem. Lemma 3.1 in Chernozhukov et al. [6] states that in order to estimate $g(X, A) = \mathbb{E}[Y|X, A]$, it suffices to estimate $g(X, A) = \mathbb{E}(Y|\alpha_0(X, A))$. Since $\alpha_0(X, A)$ is a scalar quantity, it is more efficient to estimate $\mathbb{E}(Y|\alpha_0(X, A))$ than to condition on high dimensional $(X, A)$. Thus, *Riesz* uses a multi-tasking objective to learn both $\alpha_0(X, A)$ and $g(X, A)$ jointly. First, a neural network is used to encode the input $(X, A)$ to a latent representation $Z$. Then, the architecture branches out into two heads; the first head trains the Riesz representer $\alpha_0(X, A)$ and the second branch is used to train $g(X, A)$ (see Figure 1 and Section 3 in Chernozhukov et al. [6] for details). Let the loss used to train the Riesz representer be called as *RRloss* and the loss used to train $g(X, A)$ be called as REGloss and L2Reg be the l2 regularization loss (see Section 3 in [6] for the exact form of loss). Therefore, the final loss can be written as

$$\text{RieszLoss} = \text{REGloss} + R_1 \cdot \text{RRloss} + R_2 \cdot \text{L2RegLoss}, \tag{7}$$

with $R_1$ and $R_2$ as the regularization hyper parameters. Chernozhukov et al. [6] also suggests using another term in the overall loss (TMLEloss) which we do not consider in the current setup. After training the whole neural network jointly with the above loss objective, the causal effect estimate could be calculated using:

$$Riesz := \mathbb{E}_X[g(X, A = 1) - g(X, A = 0)] \tag{8}$$

Chernozhukov et al. [6] also proposes another debiased measure of the causal effect given by:

$$DebiasedRiesz := \mathbb{E}_X\Big[\big(g(X, A = 1) - g(X, A = 0)\big) + \alpha(X, A)(Y - g(X, A))\Big] \tag{9}$$

We observe that this measure gives a worse estimate than *Riesz* on Syn-Text dataset where the ground truth causal effect is known. Thus, in all our experiments we only use the $Riesz$ estimator and leave the exploration of $DebiasedRiesz$ as future work. Please see §E.4 for the setup we use in our experiments to estimate the causal effect for different datasets.

## A.2 Max Margin Objective

Taking inspiration from the description of the max-margin classifier from Kumar et al. [19] and Nagarajan et al. [26], we give a brief introduction to the max-margin training objective that is used to train the classifier we consider in Theorem 3.1. Consider an encoder $h : \boldsymbol{X} \to \boldsymbol{Z}$ mapping

the input $x$ to latent space representation $z$, and a classifier $c : Z \to Y$ that uses the latent space representation $Z$ to predict the task label $y$ as $c(z) = w \cdot z$. The hyperplane $c(z) = 0$ is called the decision boundary of the classifier. Let the task label be binary, taking values $1$ or $-1$, without loss of generality (otherwise, the labels could be relabelled to conform to this notation). Then the points falling on one side of decision boundary $c(z < 0$ are assigned one predicted label (say $-1$) and the ones falling on the other side the other predicted label (say $+1$). Then the distance of a point $z$, having task label $y$, from the decision boundary is given by:

$$\mathcal{M}_c(z) = \frac{y \cdot c(z)}{\|w\|_2} \tag{10}$$

where $\|w\|_2$ is the L2 norm of the classifier. The *margin* of a classifier is the distance of the closest latent representation $z$ from the decision boundary. Thus, the goal of the max-margin objective is then to train a classifier that has the maximum margin. Equivalently, we want to minimize the following loss for training a classifier that has the largest margin.

$$\mathcal{L}_{mm}(c(Z), Y) = (-1) \min_i \frac{y^i c(z^i)}{\|c(z)\|_2} \tag{11}$$

where the minimum is over all the training data points indexed by $i$.

## B  Proof of Prop 3.1

**Proposition 3.1.** *Let DGP-1 and DGP-2 in Fig. 2 be the causal graphs of two data-generating processes. Let $A, C, S$ be different attributes, $X$ be the observed input, $Y$ be the observed task label and $U$ be the unobserved confounding variable. In DGP-2, $x$ is the (unobserved) core input feature that affects the label $Y$. Then:*

1. ***DGP-1 Causal Effect Identifiability:*** *Given the interventional distribution $P(X|do(A))$, the causal effect of the attribute $A$ on task label $Y$ is identifiable using observed data distribution.*

2. ***DGP-2 Causal Effect Identifiability:*** *Let $C$ be a set of observed attributes that causally affect the task label $Y$ (unknown to us), $S$ be the set of observed attributes spuriously correlated with task label $Y$ (again unknown to us), and let $\mathcal{V} = C \cup S$ be the given set of all the attributes. Then if all the causal attributes are observed then the causal effect of all the attributes in $V$ can be identified using observational data distribution alone.*

*Proof.* **First Claim:** To identify the causal effect of attribute $A$ on the label $Y$, we need access to the interventional distribution $P(Y|do(A))$. For example, in the case when $A$ is binary random variable the average causal effect of $A$ on $Y$ is given by $TE_A = \mathbb{E}[Y|do(A) = 1] - \mathbb{E}[Y|do(A) = 0]$. We can write $P(Y|do(A))$ as:

$$
\begin{aligned}
P(Y|do(A)) &= \sum_X \sum_U P(Y, X, U|do(A)) \\
&= \sum_X \sum_U \left\{ P(U)P(X|U, do(A))P(Y|X) \right\} \\
&= \sum_X P(Y|X) \left\{ \sum_U P(U)P(X|U, do(A)) \right\} \\
&= \sum_X P(Y|X)P(X|do(A))
\end{aligned} \tag{12}
$$

Since we have access to the interventional distribution $P(X|do(A))$ and access to observational distribution $P(Y, X, A)$, we can also estimate $P(Y|do(A))$ using the above identity completing the first part of the proof.

**Second Claim:** We are given the attributes $\mathcal{V} = \{C, S\}$, but we do not know the distinction between the causal and spurious attribute beforehand. We will show that we can identify the interventional

distribution $P(Y|do(A))$ where $A \in \mathcal{V}$. If $A = C$, then we have:

$$
\begin{aligned}
P(Y|do(C)) &= \sum_S P(Y, S|do(C)) \\
&= \sum_S P(S|do(C))P(Y|S, C) = \sum_S P(S)P(Y|S, C) \\
&= \sum_{\mathcal{V} \setminus A} P(\mathcal{V} \setminus A)P(Y|\mathcal{V})
\end{aligned} \tag{13}
$$

Then, when we have $A = S$, we have:

$$
\begin{aligned}
P(Y|do(S)) &= \sum_C P(Y, C|do(S)) = \sum_C P(C|do(S))P(Y|C, do(S)) \\
&= \sum_C P(C)P(Y|C, S) \qquad \because (Y \perp S|C) \\
&= \sum_{\mathcal{V} \setminus A} P(\mathcal{V} \setminus A)P(Y|\mathcal{V})
\end{aligned} \tag{14}
$$

Thus, $P(Y|do(A)) = \sum_{\mathcal{V} \setminus A} P(\mathcal{V} \setminus A)P(Y|\mathcal{V})$ for all $A \in \mathcal{V}$ and could be estimated from pure observational data.

$\square$

## C  Failure of Mouli on DGP-2 and DGP3

Mouli and Ribeiro [25] define a particular DGP for their task and propose a score that identifies the invariant transformations. DGP-2 and DGP-3 in Fig. 2 adapt the causal graph taken in their work to our setting, where the unobserved variable associated with every transformation is replaced with an observed attribute. In addition, we add an additional level of complexity to the DGP by introducing the unobserved confounding variable $U$ that introduces a spurious correlation between different attributes in DGP-2 and between an attribute and task label in DGP-3. The graph in DGP-1 is different from their setting and thus cannot use their method. This shows that the method proposed in their work doesn't generalize to different DGPs. Below, we show that their method will be able to identify the spurious attribute in the DGP-2 but fail to do so in DGP-3.

**Corollary C.0.1.** *Let the observed input $X$ be defined as the concatenation of the $C, S$ and $x$ in DGP-2 i.e. $X = [C, S, x]$ and $A$ and $x$ in DGP-3 i.e. $X = [A, x]$. Then, Theorem 1 in Mouli and Ribeiro [25] will correctly identify the spurious attribute in DGP-2. For DGP-3, it will incorrectly claim that there is an edge between $A$ and $Y$ even if it is non-existent in the original graph.*

*Proof Sketch.* For a attribute $A$ to be spurious, the score proposed in Theorem 1 in Mouli and Ribeiro [25] requires conditional independence of the $A$ with the task label $Y$ given the rest of the observed attributes and core input feature $x$. For DGP-2, we show that both the spurious attribute $S$ correctly satisfy that condition whereas the causal attribute $C$ correctly doesn't satisfy satisfy the condition. Then for DGP-3, we show that even if the attribute $A$ is spurious, it cannot be conditionally independent to Y given $x$ due to unblocked path $A \to U \to Y$ in the graph. See the complete proof below. $\square$

*Proof.* Theorem 1 in Mouli and Ribeiro [25] defines a score, henceforth called *mouli's score*, to identify whether there is an edge between any node $N$ in the graph and the main task label $Y$. There is no edge between $N$ and $Y$ iff for all $X$ :

$$
|P(Y|\Gamma_N(X), U_Y) - P(Y|X, U_Y)|_{TV} = 0 \tag{15}
$$

where $\Gamma_N$ is the most-expressive representation that is invariant with respect to the node $N$.

**Case 1 – DGP-2:** For DGP-2 when the node $N$ is causal ($C$), we have $X = [x, C, S], \Gamma_C(X) = [x, S]$ and $U_Y = \phi$. Thus, the mouli's score becomes:

$$
|P(Y|x, S) - P(Y|x, C, S)|_{TV} \neq 0 \tag{16}
$$

since for at least one combination of $x, C, S$ we have $P(Y|x, S) \neq P(Y|x, C, S)$ since $C \not\perp Y|S, x$ in DGP-2. Thus, mouli's score doesn't incorrectly mark causal node $C$ as spurious. Next for a spurious node $S$, we have $\Gamma_S(X) = [x, C]$; thus the mouli's score becomes:

$$|P(Y|x, C) - P(Y|x, C, S)|_{TV} = 0 \tag{17}$$

for all $x, C, S$ we have $P(Y|x, C) = P(Y|x, C, S)$ since $S \perp Y|C, x$. Thus the mouli's score correctly identifies the spurious node $S$.

**Case 2 – DGP-3:** For DGP-3, when the node $N = A$, we have $X = [x, A]$, $\Gamma_A(X) = [x]$, and if is spurious i.e. there is no edge from $A$ to $Y$ in the actual graph then the mouli's score becomes:

$$|P(Y|x) - P(Y|x, A)|_{TV} \neq 0 \tag{18}$$

for all $x, A$ since $A \not\perp Y|x \implies P(Y|x) \neq P(Y|x, A)$ for atleast one setting of $(x, A, Y)$. Thus the mouli's score will be non-zero and will fail to identify $A$ as a spurious node.

$\square$

# D   Proof of Theorem 3.1

We start by formalizing the assumption we made about the disentangled latent space in §3.3.

**Assumption D.1** (Disentangled latent space). *Let $\mathcal{A} = \{ca_1, \ldots, ca_K, sp_1, \ldots, sp_J\}$ be the set of attribute given to us. The latent representation $z$ is disentangled and is of form $[z_{ca_1}, \ldots, z_{ca_K}, z_{sp_1}, \ldots, z_{sp_J}]$, where $z_{ca_k} \in \mathbb{R}^{d_k}$ is the features corresponding to causal attribute "$ca_k$" and $z_{sp_j} \in \mathbb{R}^{d_j}$ are the features corresponding to spurious attribute "$sp_j$". Here $d_k$ and $d_j$ are the dimensions of $z_{ca_k}$ and $z_{sp_j}$ respectively.*

Now we are ready to formally state the main theorem and its proof that shows that it is sufficient to have the correct ranking of the causal effect of causal and spurious attributes to learn a classifier invariant to the spurious attributes using one instantiation of our regularization objective (3). Our proof goes in two steps:

1. **First Claim**: To prove that the desired classifier will be preferred by our loss objective over the classifier learned by the max-margin objective (henceforth denoted as undesired), we compare the loss incurred by both the classifier w.r.t to our training objective. Next, we show that given the ranking of the causal effect of causal and spurious attributes follows the relation mentioned in the theorem statement, there always exists a regularization strength when the desired classifier has a lower loss than the undesired one w.r.t to our training objective.
2. **Second Claim**: Using our first claim we show that the desired classifier will have a lower loss than any other classifier that uses the spurious attribute. Thus the desired classifier is preferred over any classifier that uses the spurious attribute using our loss objective. Then we show that among all the classifier that only uses the causal attribute for prediction, again our loss objective will select the desired classifier.

**Theorem 3.1.** *Let the latent space be frozen and disentangled such that $z = [z_{ca_1}, \ldots, z_{ca_K}, z_{sp_1}, \ldots, z_{sp_J}]$ (Assm D.1). Let the desired classifier $c^{des}(z) = \sum_{k=1}^{K} w_{ca_k}^{des} \cdot z_{ca_k}$ be the max-margin classifier among all linear classifiers that use only the causal features $z_{ca_k}$'s for prediction. Let $c^{mm}(z) = \sum_{k=1}^{K} w_{ca_k}^{mm} \cdot z_{ca_k} + \sum_{j=1}^{J} w_{sp_j}^{mm} \cdot z_{sp_j}$ be the max-margin classifier that uses both the causal and the spurious features, and let $w_{sp_j}^{mm} \neq \mathbf{0}$, $\forall j \in [J]$. We assume $w_{sp_j}^{mm} \neq \mathbf{0}$, $\forall j \in [J]$, without loss of generality because otherwise, we can restrict our attention only to those $j \in [J]$ that have $w_{sp_j}^{mm} \neq \mathbf{0}$. Let the norm of the parameters of both the classifier be set to 1 i.e $\sum_{k=1}^{K} \|w_{ca_k}^{mm}\|_{p=2}^2 + \sum_{j=1}^{J} \|w_{sp_j}^{mm}\|_{p=2}^2 = \sum_{k=1}^{K} \|w_{ca_k}^{des}\|_{p=2}^2 = 1$. Then if regularization coefficients are related s.t. $\text{mean}\left( \left\{ \frac{\lambda_{ca_k}}{\lambda_{sp_j}} \cdot \eta_{k,j} \right\}_{k \in [K], j \in [J]} \right) < \frac{J}{K}$ where $\eta_{k,j} = \frac{\|w_{ca_k}^{des}\| - \|w_{ca_k}^{mm}\|_p}{\|w_{sp_j}^{mm}\|_p}$, then*

1. **Preference:** $\mathcal{L}_{AutoACER}(c^{des}(z)) < \mathcal{L}_{AutoACER}(c^{mm}(z))$. *Thus, our causal effect regularization objective (Def 3) will choose the $c^{des}(z)$ classifier over the max-margin classifier $c^{mm}(z)$ which uses the spuriously correlated feature.*

2. **Global Optimum:** *The desired classifier $c^{des}(\boldsymbol{z})$ is the global optimum of our loss function $\mathcal{L}_{AutoACER}$ when $J = 1$, $K = 1$, $p = 2$, the regularization strength are related s.t. $\lambda_{ca_1} < \lambda_{sp_1} \implies |\hat{TE}_{ca_1}| > |\hat{TE}_{sp_1}|$ and search space of linear classifiers $c(\boldsymbol{z})$ are restricted to have the norm of parameters equal to 1.*

*Proof.* **Proof of Part 1:** Since we are training our task classifier using the max-margin objective (Eq. 11) thus over training objective becomes:

$$\mathcal{L}_{AutoACER}\big(c(h(\boldsymbol{x}))\big) \coloneqq \mathcal{L}_{mm}\big(c(h(\boldsymbol{x})), y\big) + R \cdot \sum_{k=1}^{K} \lambda_{ca_k} \|\boldsymbol{w}_{ca_k}\|_p + \sum_{j=1}^{J} \lambda_{sp_j} \|\boldsymbol{w}_{sp_j}\|_p \quad (19)$$

where $\|\cdot\|_p$ is the $L_p$ norm. For ease of exposition, we will denote $c(h(\boldsymbol{x}))$ as $c(\boldsymbol{z})$ where $\boldsymbol{z}$ is the latent space representation of $\boldsymbol{x}$ given by $\boldsymbol{z} = h(\boldsymbol{x})$. Next, we will show that there always exists a regularization strength $R \geq 0$ s.t. $\mathcal{L}_{AutoACER}(c^{des}(\boldsymbol{z})) < \mathcal{L}_{AutoACER}(c^{mm}(\boldsymbol{z}))$ given $\text{mean}\left(\left\{\frac{\lambda_{ca_k}}{\lambda_{sp_j}} \cdot \eta_{k,j}\right\}_{k \in [K], j \in [J]}\right) < \frac{J}{K}$ (from the main statement of Theorem 3.1). To have $\mathcal{L}_{AutoACER}(c^{des}(\boldsymbol{z})) < \mathcal{L}_{AutoACER}(c^{mm}(\boldsymbol{z}))$ we need to select the regularization strength $R \geq 0$ s.t:

$$\mathcal{L}_{mm}\big(c^{des}(\boldsymbol{z}), y\big) + R \cdot \left\{\sum_{k=1}^{K} \lambda_{ca_k} \|\boldsymbol{w}_{ca_k}^{des}\|_p\right\} <$$
$$\mathcal{L}_{mm}\big(c^{mm}(\boldsymbol{z}), y\big) + R \cdot \left\{\sum_{k=1}^{K} \lambda_{ca_k} \|\boldsymbol{w}_{ca_k}^{mm}\|_p + \sum_{j=1}^{J} \lambda_{sp_j} \|\boldsymbol{w}_{sp_j}^{mm}\|_p\right\} \quad (20)$$

The max-margin objective (see §A.2 for background) given is given by $\mathcal{L}_{mm}\big(c(\boldsymbol{z}), y\big) = (-1) \cdot \min_i \left\{\frac{y^i c(\boldsymbol{z}^i))}{\|c(\boldsymbol{z})\|_2}\right\}$ where $\boldsymbol{z}^i$ is the latent space representation of the input $\boldsymbol{x}^i$ with label $y^i$ and $\|c(\boldsymbol{z})\|_2$ is the L2-norm of the weight vector of the classifier $c(\boldsymbol{z})$. The norm of both the classifier $c^{des}(\boldsymbol{z})$ and $c^{mm}(\boldsymbol{z})$ is 1 (from the statement of this Theorem 3.1). Substituting the max-margin loss for both the classifier in Eq. 20 we get:

$$(-1) \cdot \min_i \left\{y^i c^{des}(\boldsymbol{z}^i)\right\} + R \cdot \left\{\sum_{k=1}^{K} \lambda_{ca_k} \|\boldsymbol{w}_{ca_k}^{des}\|_p\right\} <$$
$$(-1) \cdot \min_j \left\{y^j c^{mm}(\boldsymbol{z}^j)\right\} + R \cdot \left\{\sum_{k=1}^{K} \lambda_{ca_k} \|\boldsymbol{w}_{ca_k}^{mm}\|_p + \sum_{j=1}^{J} \lambda_{sp_j} \|\boldsymbol{w}_{sp_j}^{mm}\|_p\right\} \quad (21)$$

Rearranging the above equation we get:

$$R \cdot \underbrace{\left\{\sum_{k=1}^{K} \lambda_{ca_k} \left[\|\boldsymbol{w}_{ca_k}^{mm}\|_p - \|\boldsymbol{w}_{ca_k}^{des}\|_p\right] + \sum_{j=1}^{J} \lambda_{sp_j} \|\boldsymbol{w}_{sp_j}^{mm}\|_p\right\}}_{LHS-term1} >$$
$$\min_j \left\{y^j c^{mm}(\boldsymbol{z}^j)\right\} - \min_i \left\{y^i c^{des}(\boldsymbol{z}^i)\right\} \quad (22)$$

If the LHS-term 1 in the above equation is greater than 0, then we can always select a regularization strength $R > 0$ s.t. the above inequality is always satisfied, whenever

$$R > \frac{\min_j \left\{y^j c^{mm}(\boldsymbol{z}^j)\right\} - \min_i \left\{y^i c^{des}(\boldsymbol{z}^i)\right\}}{\left\{\sum_{k=1}^{K} \lambda_{ca_k} \left[\|\boldsymbol{w}_{ca_k}^{mm}\|_p - \|\boldsymbol{w}_{ca_k}^{des}\|_p\right] + \sum_{j=1}^{J} \lambda_{sp_j} \|\boldsymbol{w}_{sp_j}^{mm}\|_p\right\}} \quad (23)$$

Next we will show that given mean $\left( \left\{ \frac{\lambda_{ca_k}}{\lambda_{sp_j}} \cdot \eta_{k,j} \right\}_{k \in [K], j \in [J]} \right) < \frac{J}{K}$, the LHS-term1 in Eq. 22 is always greater than 0. For LHS-term1 to be greater than 0 we need:

$$\sum_{k=1}^{K} \lambda_{ca_k} \left[ \|\boldsymbol{w}_{ca_k}^{mm}\|_p - \|\boldsymbol{w}_{ca_k}^{des}\|_p \right] > (-1) \cdot \sum_{j=1}^{J} \lambda_{sp_j} \|\boldsymbol{w}_{sp_j}^{mm}\|_p$$

$$\implies \sum_{k=1}^{K} \lambda_{ca_k} \left[ \|\boldsymbol{w}_{ca_k}^{des}\|_p - \|\boldsymbol{w}_{ca_k}^{mm}\|_p \right] < \sum_{j=1}^{J} \lambda_{sp_j} \|\boldsymbol{w}_{sp_j}^{mm}\|_p \tag{24}$$

Since $\lambda_{(\cdot)} = 1/|\hat{TE}_{(\cdot)}|$ and $|\hat{TE}_{(\cdot)}| \in [0,1]$ we have $\lambda_{(\cdot)} > 0$ (see §3.3 for discussion). From the main statement of this theorem we have $\boldsymbol{w}_{sp_j}^{mm} \neq \boldsymbol{0} \implies \|\boldsymbol{w}_{sp_j}^{mm}\|_p > 0$ for all value of "$p$". Next, we have the following two cases:

**Case 1** $\left( \sum_{k=1}^{K} \lambda_{ca_k} \left[ \|\boldsymbol{w}_{ca_k}^{des}\|_p - \|\boldsymbol{w}_{ca_k}^{mm}\|_p \right] \leq 0 \right)$: For this case the above Eq. 24 is trivially satisfied since $\sum_{j=1}^{J} \lambda_{sp_j} \|\boldsymbol{w}_{sp_j}^{mm}\|_p > 0$ as $\forall j \in [J], \; \lambda_{sp_j} > 0$ and $\|\boldsymbol{w}_{sp_j}^{mm}\|_p > 0$.

**Case 2** $\left( \sum_{k=1}^{K} \lambda_{ca_k} \left[ \|\boldsymbol{w}_{ca_k}^{des}\|_p - \|\boldsymbol{w}_{ca_k}^{mm}\|_p \right] > 0 \right)$: Since for all $j \in [J]$ we have $\lambda_{sp_j} > 0$ and $\|\boldsymbol{w}_{sp_j}^{mm}\|_p > 0$, rearranging we get:

$$\sum_{j=1}^{J} \frac{\lambda_{sp_j} \|\boldsymbol{w}_{sp_j}^{mm}\|_p}{\sum_{k=1}^{K} \lambda_{ca_k} \left[ \|\boldsymbol{w}_{ca_k}^{des}\|_p - \|\boldsymbol{w}_{ca_k}^{mm}\|_p \right]} > 1$$

$$\sum_{j=1}^{J} \frac{1}{\sum_{k=1}^{K} \left( \frac{\lambda_{ca_k}}{\lambda_{sp_j}} \right) \left( \frac{\|\boldsymbol{w}_{ca_k}^{des}\|_p - \|\boldsymbol{w}_{ca_k}^{mm}\|_p}{\|\boldsymbol{w}_{sp_j}^{mm}\|_p} \right)} > 1 \tag{25}$$

$$\frac{1}{\sum_{j=1}^{J} \left\{ \underbrace{\frac{1}{\sum_{k=1}^{K} \left( \frac{\lambda_{ca_k}}{\lambda_{sp_j}} \right) \left( \frac{\|\boldsymbol{w}_{ca_k}^{des}\|_p - \|\boldsymbol{w}_{ca_k}^{mm}\|_p}{\|\boldsymbol{w}_{sp_j}^{mm}\|_p} \right)}}_{\text{LHS-term 3}} \right\}} \cdot J < J$$

The above equation is the harmonic mean of the LHS-term 3 and LHS-term 3 is $> 0$ for all values of $j$. We know that the harmonic mean is always less than or equal to the arithmetic mean for a set of positive numbers (Inequalities book by GH Hardy [34]). Thus the above inequality is satisfied if the following inequality is satisfied:

$$\sum_{j=1}^{J} \sum_{k=1}^{K} \frac{\left( \frac{\lambda_{ca_k}}{\lambda_{sp_j}} \right) \left( \frac{\|\boldsymbol{w}_{ca_k}^{des}\|_p - \|\boldsymbol{w}_{ca_k}^{mm}\|_p}{\|\boldsymbol{w}_{sp_j}^{mm}\|_p} \right)}{J \cdot K} < \frac{J}{K} \tag{26}$$

Since the above condition on regularization strength of features is satisfied (given in the main statement of this theorem), we have $\mathcal{L}_{AutoACER}(c^{des}(\boldsymbol{z})) < \mathcal{L}_{AutoACER}(c^{mm}(\boldsymbol{z}))$ thus completing the proof.

The above Eq. 26 states that the mean of $\left( \frac{\lambda_{ca_k}}{\lambda_{sp_j}} \right) \left( \frac{\|\boldsymbol{w}_{ca_k}^{des}\|_p - \|\boldsymbol{w}_{ca_k}^{mm}\|_p}{\|\boldsymbol{w}_{sp_j}^{mm}\|_p} \right)$ should be less than $J/K$. Thus the above equation is also satisfied if individually all the terms considered when taking the mean are individually less than $J/K$. Thus, a stricter but more intuitive condition on $\lambda$'s s.t Eq. 26 is satisfied is given by:

$$\left( \frac{\lambda_{ca_k}}{\lambda_{sp_j}} \right) \eta_{k,j} < \frac{J}{K}$$

$$\hat{TE}_{sp_j} \left( \frac{K}{J} \eta_{k,j} \right) < \hat{TE}_{ca_k} \tag{27}$$

where $\eta_{k,j} = \left( \frac{\|\boldsymbol{w}_{ca_k}^{des}\|_p - \|\boldsymbol{w}_{ca_k}^{mm}\|_p}{\|\boldsymbol{w}_{sp_j}^{mm}\|_p} \right)$.

**Proof of Part 2:** Given $p = 2$ and since $K = 1$ and $J = 1$, the desired classifier takes form $c^{des}(\boldsymbol{z}) = \boldsymbol{w}_{ca,1}^{des} \cdot \boldsymbol{z}_{ca,1} = \boldsymbol{w}_{ca}^{des} \cdot \boldsymbol{z}_{ca}$. For ease of exposition, we will drop "1" from the subscript which denotes

Table 2: **Conditional Probability Distribution for Syn-Text dataset** ($P(Y|\text{casual}, \text{confound})$): The rows represent the different settings of the parent attributes— causal and confound and the columns represent different values the task label $Y$ can take. Each cell represents the probability of observing the task label given the parent's setting.

| | $Y = 0$ | $Y = 1$ |
|---|---|---|
| *causal*=0, *confound*=0 | 0.99 | 0.01 |
| *causal*=0, *confound*=1 | 0.30 | 0.70 |
| *causal*=1, *confound*=0 | 0.70 | 0.30 |
| *causal*=1, *confound*=1 | 0.01 | 0.99 |

the feature number. Let $c^s(\boldsymbol{z}) = \boldsymbol{w}_{ca}^s \cdot \boldsymbol{z}_{ca} + \boldsymbol{w}_{sp}^s \cdot \boldsymbol{z}_{sp}$ be any classifier which uses spurious feature $\boldsymbol{z}_{sp}$ for its prediction s.t. $\boldsymbol{w}_{sp}^s \neq 0$. From statement 1 of this theorem, when the regularization strength are related such that $\lambda_{ca} \cdot \eta < \lambda_{sp}$, we have $\mathcal{L}_{AutoACER}(c^{des}(\boldsymbol{z})) < \mathcal{L}_{AutoACER}(c^s(\boldsymbol{z}))$ where $\eta = \frac{\|\boldsymbol{w}_{ca}^{des}\|_2 - \|\boldsymbol{w}_{ca}^s\|_2}{\|\boldsymbol{w}_{sp}^s\|_2}$. Since the search space of the linear classifiers is constrained to have the norm of parameters equal to 1 we have $\|\boldsymbol{w}_{ca}^s\|_2^2 + \|\boldsymbol{w}_{sp}^s\|_2^2 = 1$ and $\|\boldsymbol{w}_{ca}^{des}\|_2 = 1$. Let $\|\boldsymbol{w}_{ca}^s\| = \theta \in [0,1)$, then we have $\|\boldsymbol{w}_{sp}^s\| = \sqrt{1 - \theta^2}$. Substituting these values in $\eta$ we get:

$$
\begin{aligned}
\eta(\theta) &= \frac{1 - \theta}{\sqrt{1 - \theta^2}} \\
&= \sqrt{\frac{1 - \theta}{1 + \theta}} \qquad (\theta \neq 1)
\end{aligned} \tag{28}
$$

Thus $\eta(\theta)$ is a decreasing function for $\theta \in [0,1)$ and has its maximum value $\eta_{max} = 1$ at $\theta = 0$. Thus, if the regularization strengths are related such that $\lambda_{ca} \cdot \eta_{max} < \lambda_{sp} \implies \lambda_{ca} < \lambda_{sp}$ we have $\mathcal{L}_{AutoACER}(c^{des}(\boldsymbol{z})) < \mathcal{L}_{AutoACER}(c^s(\boldsymbol{z}))$ for all possible $c^s(\boldsymbol{z})$. Since we are given that $\lambda_{ca} < \lambda_{sp}$ in the second statement of the theorem $c^{des}(\boldsymbol{z})$ is preferred by our regularization objective among all possible $c^s(\boldsymbol{z})$. Next, among the classifier $c^\psi(\boldsymbol{z}) \neq c^{des}(\boldsymbol{z})$ which only uses the causal feature for prediction, we have:

$$
\mathcal{L}_{AutoACER}(c^{des}(\boldsymbol{z})) = \mathcal{L}_{mm}(c^{des}(\boldsymbol{z})) + R \cdot \left\{ \lambda_{ca} \|\boldsymbol{w}_{ca}^{des}\|_2 \right\} = \mathcal{L}_{mm}(c^{des}(\boldsymbol{z})) + R\lambda_{ca} \cdot 1 \tag{29}
$$

$$
\mathcal{L}_{AutoACER}(c^\psi(\boldsymbol{z})) = \mathcal{L}_{mm}(c^\psi(\boldsymbol{z})) + R \cdot \left\{ \lambda_{ca} \|\boldsymbol{w}_{ca}^\psi\|_2 \right\} = \mathcal{L}_{mm}(c^\psi(\boldsymbol{z})) + R\lambda_{ca} \cdot 1 \tag{30}
$$

Since the desired classifier has maximum margin when using only causal feature for prediction (by definition), we have $\mathcal{L}_{mm}(c^{des}(\boldsymbol{z})) < \mathcal{L}_{mm}(c^\psi(\boldsymbol{z}))$ for all other classifiers ($c^\psi(\boldsymbol{z})$) which only uses the causal feature for prediction. Thus the desired classifier is the global optimum for our loss function $\mathcal{L}_{AutoACER}$ when the classifiers are constrained to have parameters with a norm equal to 1 thereby completing the proof.

$\square$

# E   Experimental Setup

## E.1   Datasets

We perform extensive experiments on 6 datasets spanning synthetic, semi-synthetic, and real-world datasets. In §4 we give results for 3 such datasets — Syn-Text which is a synthetic dataset, MNIST34 which is a semi-synthetic dataset and Twitter-AAE which is a real-world dataset. In §F, we further evaluate our methods and other baselines on one additional real-world dataset (Civil-Comments) which has three different subsets. Below we give a detailed description of all the datasets.

**Syn-Text Dataset.**   To create this Syn-Text dataset we first create a tabular dataset with three binary attributes— *causal, spurious* and *confound*. The causal and confound attribute create the task label ($Y$) and the confound attribute creates the spurious attribute. Causal and confound attributes are independent, $P(\text{causal} = 0) = P(\text{causal} = 1) = 0.5$ and $P(\text{confound} = 0) = P(\text{confound} = 1) = $

Table 3: **World List corresponding to every Attribute for Syn-Text dataset.** Every value of a given attribute is associated with the following list of words. For creating a sentence we take the values of every attribute, randomly sample 3 words from the corresponding set, and concatenate them to form the final sentence i.e. sentence = [causal words, confound words, spurious words].

| Attribute | Value | Word List |
|---|---|---|
| Causal | 0 | apple, mango, tomato, cherry, pear, fruit, banana, pear, grapes |
| | 1 | rose, jasmine, tulip, lotus, daisy, sunflower, flower, marigold, dahlia, orchid |
| Confound | 0 | bad, inferior, substandard, inadequate, rotten, pathetic, faulty, defective, |
| | 1 | good, best, awesome, teriffic, mighty, gigantic, tremendous, mega, colossal, |
| Spurious | 0 | horror, gore, crime, thriller, mystery, gangster, drama, dark |
| | 1 | comedy, romance, fantasy, sports, epic, animated, adventure, science |

Table 4: **Automatically generated counterfactual example using GPT3.5 for Twitter-AAE dataset.**

| Input Race | Input Sentence | Generated Counterfactual |
|---|---|---|
| white | hey guess where i just ate at ? cracker barrel . they had plenty of mashed potatoes . just thought i'd let you know . twitter-entity | Hey, guess where I jus' ate at? Cracker Barrel! Dey had plenty o' mashed potatoes. Jus' thought I'd letchu know. #CrackerBarrelCravings |
| white | girl don't sit there and listen to him talk about lacrosse when we all know you have no clue what he is saying | Girl, don't siddity there listenin' to him talkin' 'bout lacrosse when we all know you ain't got no clue 'bout what he sayin'. |
| black | we don't wanna neva end | We don't want it to ever end. |
| black | i wonder if reggie evan still thnk blatche and joe johnson on the same level as lebron | I wonder if Reggie Evans still regard Blatche and Joe Johnson at the same level as LeBron. |

0.5. The conditional probability distribution (CPD) of the task label given to the parents is given in Table 2. The CPD for spurious attribute is not fixed i.e $P(\text{spurious}|\text{confound}) = \kappa$, which we vary in our experiment to change the overall predictive correlation of spurious attribute with the task label. We then create two versions of this dataset **(1)** Syn-Text-Obs-Conf and **(2)** Syn-Text-Unobs-Conf. In the Syn-Text-Obs-Conf dataset, we keep all the attributes in the dataset but for Syn-Text-Obs-Conf to simulate the real-world setting where there is an unobserved confounding variable we remove the *confound* attribute from the dataset. Post this, we use this tabular dataset to generate textual sentences for every example. For each of the values of (observed) attributes, we sample 3 words from a fixed set of words (separate for each value of attribute) that we append together to form the final sentence (see Table 3 for the set of words corresponding to every attribute). The Syn-Text-Unobs-Conf dataset corresponds to the DGP-3 in the Fig. 2 where the unobserved node $U$ is the confound attribute and $A$ is the spurious attribute and there is no edge between A and task label Y. In our experiment, we sample 1k examples using the above methods and create an 80-20 split for the train and test set. Note that the predictive correlation mentioned in experiments and other tables for all versions of Syn-Text dataset is between the confound and spurious attribute given by $\kappa = P(\text{spurious}|\text{confound})$. Our experiments require access to the counterfactual example $\boldsymbol{x}_{a'} \sim \mathcal{Q}(\boldsymbol{x}_a)$ where the attribute $A = a$ is changed to $A = a'$ in the input. To generate this counterfactual example we flip the value of

Table 5: **Words associated with different attributes in Civil-Comments dataset.**

| Attribute | Associated Word List |
|---|---|
| Race | black, white, supremacists, obama, clinton, trump, john, negro, blacks, whites, nigga |
| Gender | male, patriarchy, feminist, woman, men, man, female, women, he, she, him, his, mother, father, feminists, boy, girl |
| Religion | christian, muslim, allah, jesus, john, mohammed, catholic, priest, church, islamic, islam, hijab |

the attribute in the given input and generate the corresponding sentence using the same procedure mentioned above.

**MNIST34 Dataset.** We use the MNIST to evaluate the efficacy of our method on the vision dataset. Following Mouli and Ribeiro [25], we subsample only the digits 3 (digit attribute label=0) and 4 (digit attribute label=1) from this dataset and create a synthetic task. To create this dataset, we first take a grayscale image (with digit attribute either 3 or 4). Then we add background color — red labeled as 1 or green labeled as 0 — to the image uniformly randomly i.e. $P(\text{color}|\text{digit}) = 0.5$. We create the task label using a deterministic function over the digit and color attribute formally defined as $Y$ = color XOR digit where XOR is the exclusive OR operator. Thus the attribute, digit, and color are causal attributes for this dataset. Next, to introduce spurious correlation we add rotation transformation to the image — $0°$ labeled as 1 or $90°$ labelled as 0. We vary the correlation between the causal attributes (color and digit) and spurious attribute (rotation) by varying the CPD $P(\text{rotation}|\text{color, digit})$. Since the combined causal attribute label is the same as the task label the predictive correlation between the task label and spurious attribute is given by $\kappa = P(\text{rotation}|\text{color, digit})$ which is vary in all our experiments. The above data-generating process resembles DGP-2 in the Fig. 2 where the node $C$ is the combined causal attributes (color and digit) and the node $S$ is the spurious attribute (rotation). The core input feature $x$ is an empty set in this dataset. All the attributes combinedly create the final input image $X$. We sample 10k examples using the above process described above and create an 80-20 split for the train and test set. Our experiments require access to the counterfactual example $\boldsymbol{x}_{a'} \sim \mathcal{Q}(\boldsymbol{x}_a)$ where the attribute $A = a$ is changed to $A = a'$ in the input. To generate this counterfactual example we flip the value of the relevant attribute the attribute label and generate the counterfactual image.

**Twitter-AAE Dataset.** This is a real-world dataset where given a sentence the task is to predict the sentiment of the sentence. Following Elazar and Goldberg [9], Kumar et al. [19], we simplify the task to predict the binary sentiment (Positive or Negative) given the tweet. Every tweet is also associated with the demographic attribute "race" which is correlated with the task label in the dataset. Following Elazar and Goldberg [9], Kumar et al. [19] and considering that changing the race of the person in the tweet should not affect the sentiment, we consider race as the spurious attribute. We use the code made available by Elazar and Goldberg [9] to automatically label the tweet with the race that uses AAE (African-American English) and SAE (Standard American English) as a proxy for race. This data-generating process of this dataset resembles DGP-1 in Fig. 2 since the sentiment (task) labels are annotated using a deterministic function given the input tweet ([9]). The dataset [1] and code [2] to create the dataset are available online. We subsample 10k examples and use an 80-20 split for the train and test set. See §E.2 for details on how we vary the predictive correlation in this dataset between the task labels and spurious attribute in our experiment. Our experiments require access to the counterfactual example $\boldsymbol{x}_{a'} \sim \mathcal{Q}(\boldsymbol{x}_a)$ where the attribute $A = a$ is changed to $A = a'$ in the input. Since this is a real-world dataset where we don't have access to the data-generating process. Thus we take help from GPT3.5 (text-davinci-003 model) [5] to automatically generate the counterfactual tweet where the race attribute is changed. See Table 4 for a sample of generated counterfactual tweets.

**Civil-Comments Datasets.** To further evaluate our result on another real-world dataset we conduct use another dataset Civil-Comments[3] (WILDS dataset, [18]). Given a sentence, the task is to predict the toxicity of the sentence which is a continuous value in the original dataset. In our experiment,

---

[1]TwitterAAE dataset could be found online at: `http://slanglab.cs.umass.edu/TwitterAAE/`

[2]The code for Twitter-AAE dataset acquisition and automatically labeling race information is available at: `https://github.com/yanaiela/demog-text-removal`

[3]Civil Comments dataset is available online at `https://www.tensorflow.org/datasets/catalog/civil_comments`

we binarize this task to predict whether the sentence is toxic or not by labeling the sentence with toxicity score $\geq 0.5$ as toxic or otherwise non-toxic. We use a subset of the original dataset (CivilCommentsIdentities) which includes an extended set of auxiliary identity labels associated with the sentence. We finally select three different identity labels (*race*, *gender*, and *religion*). Race attribute takes two values *black* or *white*, Gender attribute takes two values *male* or *female*, and Religion attribute also takes two values *muslim* or *christian*. We expect that these identity attributes to have zero causal effect on the task label since changing a person's race, gender, or religion in the sentence should not change the toxicity of the sentence. For each considered identity label, we create a corresponding different subset of the dataset named Civil-Comments (Race), Civil-Comments (Gender), and Civil-Comments (Religion). This data-generating process of this dataset also follows DGP-1 since toxicity (task) labels were generated from human annotators. We subsample 5k examples for Civil-Comments (Race) and Civil-Comments (Gender) and 4k examples to create Civil-Comments (Religion) dataset that we use in our experiments. Then, we use an 80-20 split to create a train and test set. For details on how we vary the predictive correlation see §E.2. In our experiment, for every input $x$ we need access to the counterfactual where the attribute's value is changed in the input. To create such counterfactuals we use a deterministic function that remove the words related to the attribute from the sentence. Currently, we use a hand-crafted set of words for each of the attributes for removal, we plan to replace that more natural counterfactual generated from generative models like GPT3. Table 5 show the set of words associated with every attribute that we use for removal.

## E.2 Dataset with varying levels of spurious Correlation.

We create multiple subsets of the dataset with different levels of predictive correlation ($\kappa$) between the task label ($y$) and the attribute ($a$). The task label and all the attributes we consider in our work are binary taking values from 0 and 1. Following Kumar et al. [19], we define 4 subgroups in the dataset for each combination of $(y, a)$. The subset of the dataset with $(y = 1, a = 1)$ and $(y = 0, a = 0)$ is defined as majority group $S_{maj}$ where the task labels $y$ are correlated with the attribute. The remaining subset of the dataset where the correlation break is named $S_{min}$ which contains the dataset with $(y = 1, a = 0)$ and $(y = 0, a = 1)$. Next, we artificially vary the correlation between the attribute and the task label by varying $\kappa = Pr(y = a)$ i.e. the number of examples with the same task label and attribute. Following Kumar et al. [19], we can reformulate $\kappa$ in terms of the size of the majority and minority groups:

$$\kappa := \frac{|S_{maj}|}{|S_{maj}| + |S_{min}|} \tag{31}$$

For both Twitter-AAE and Civil-Comments dataset, we consider 6 different settings of $\kappa$ from the set $\{0.5, 0.6, 0.7, 0.8, 0.9, 0.99\}$ by artificially varying the size of $S_{maj}$ and $S_{min}$. To do so, we keep the size of $S_{maj}$ fixed, and then based on the desired value of $\kappa$ we determine the number of samples to take in $S_{min}$ using the above equation. Thus for different values of $\kappa$ the overall training dataset size ($|S_{maj}| + |S_{min}|$) changes in Twitter-AAE and Civil-Comments datasets. Thus it is important to only draw an independent conclusion from the different settings of $\kappa$ in these datasets.

## E.3 Encoder for the different datasets.

We give the details of how we encode different types of inputs to feed to the neural network in our experiment. For specific details of the rest of the architecture see the individual setup for every method in §E.4, E.5 and E.6.

**Syn-Text Dataset.** We tokenize the sentence into a list of words and use 100-dimensional pretrained GloVe word embedding [28] to get a vector representation for each word. Next, to get the final representation of a sentence we take the average of all the word embedding in the sentence. The word embedding is fixed and not trained with the model. Post this we add an additional trainable fully connected layer (with output dimension 50) without any activation to get the final representation for the sentence used by different methods with different loss objectives (see §E.4, E.5 and E.6).

**MNIST34 Dataset.** We directly take the image as input and normalize it by dividing it with a scalar 255. Post this we use a convolutional neural network for further processing the image. Specifically, we apply the following layers in sequence to get the final representation of the image.

1. 2D convolution layer, activation = relu, filter size = (3,3), channels = 32
2. 2D max pooling layer, with filter size (2,2)
3. 2D convolution layer, activation = relu, filter size = (3,3), channels = 64
4. 2D max pooling layer, with filter size (2,2)
5. 2D convolution layer, activation = relu, filter size = (3,3), channels = 128

Next, we flatten the output of the above last layer to get the final representation of the image used by different methods with different loss objectives (see §E.4, E.5 and E.6).

**Twitter-AAE and Civil-Comments Datasets.** We use Hugging Face [36] transformer implementation of BERT [8] *bert-base-uncased* model to encode the input sentence. We use the pooled output of the [CLS] token as the encoded representation of the input for further processing by other methods (see §E.4, E.5 and E.6). We start with the pretrained weight and fine-tune the model based on the specific task.

## E.4   Setup: Causal Effect Estimators

We use two different causal effect estimators — *Direct* (Eq. 5) and *Riesz* (Eq. 8) — to estimate the causal effect of a attribute on the task label $Y$ (see §A.1 for details of the individual estimators).

***Direct* estimator in practice.** For *Direct* estimator we use a neural network to estimate $\mathbb{E}[Y|X, A]$. To get the causal effect we select the best model based either based on validation loss or validation accuracy in the prediction of the task label $Y$ given $(X, A)$. We refer to these two versions of *Direct* estimators as *Direct(loss)* and *Direct(acc)* for the setting when validation loss and validation accuracy are used for selection respectively. To estimate $\mathbb{E}[Y|X, A]$ we use the following loss objective:

$$\Big[Y - g(X, A)\Big]^2 + R \cdot \text{L2loss} \tag{32}$$

where $g(X, A)$ is the neural network predicting the task label $Y$ given input $(X, A)$, L2loss is the L2 regularization loss and R is the regularization strength hyperparameter. For Syn-Text dataset, we don't apply L2loss, for MNIST34 dataset we do a hyperparameter search over $R \in \{0.0, 0.1, 1.0, 10.0, 100.0, 200.0, 1000.0\}$, for Twitter-AAE we use $R \in \{0.0, 10.0, 100.0, 1000.0\}$ and Civil-Comments dataset we use $R \in \{0.0, 1.0, 10.0, 100.0, 200.0, 1000.0\}$.

***Riesz* estimator in practice.** To get the causal effect from *Riesz* we optimize the loss function defined in Eq. 8. We fix the regularization hyperparameter $R_1$ for RRloss to a fixed value 1 in all the experiments and search over the L2loss regularization hyperparameter ($R_2$). For Syn-Text dataset, we don't use the L2loss at all, for MNIST34 dataset we search over $R_2 \in \{0.0, 0.1, 1.0, 10.0, 100.0, 200.0, 1000.0\}$, for Twitter-AAE dataset we use $R_2 \in \{0.0, 10.0, 100.0, 200.0, 1000.0\}$ and for Civil-Comments dataset we use $R_2 \in \{0.0, 1.0, 10.0, 100.0, 200.0, 1000.0\}$. Then similar to *Direct* estimator, for selecting the best model for estimating the causal effect we use validation loss and validation accuracy of $g(X, A)$ for predicting the task label $Y$ using input $(X, A)$. We call the estimators *Riesz(loss)* and *Riesz(acc)* based on this selection criteria.

For both the estimator we train the model for 200 epochs/iterations in Syn-Text dataset and 20 epochs for the rest of the datasets and use either validation loss or validation accuracy to select the best training epoch (as described above).

## E.5   Setup: AutoACER

We use the regularization objective defined in Eq. 1 and 2 to train the classifier using our method AutoACER. For each dataset, first, we encode the inputs to get a vector representation using the procedure described in §E.3. We use cross-entropy objectives to train the task prediction ($\mathcal{L}_{task}$). Next, we take the causal effect estimated from Stage 1 and map it to the closest value in set $\{-1.0, -0.5, -0.1, 0.0, 0.1, 0.3, 0.5, 1.0\}$. For Syn-Text and MNIST34 dataset we map the estimated treatment effect from Stage 1 to the closest value in set $\{-1.0, -0.7, -0.5, -0.3, -0.1, 0.0, 0.1, 0.3, 0.5, 0.7, 1.0\}$. The mapped causal effect is then used to regularize the model using the regularization term $\mathcal{L}_{Reg}$ (see Eq. 1 and 2). Also, we search over the regularization strength $R$ (Eq. 1) to select the best model. For all the datasets we choose the

value of $R$ from set $\{1, 10, 100, 1000\}$. We train the model for 20 epochs for all the datasets. Next, to compare the results with other methods we select the best model (across different regularization hyperparameters and training epochs) that has the highest accuracy on the validation set. Here accuracy of the model is defined as:

$$\text{accuracy of model} = \frac{\sum_{i=1}^{N} \mathbb{1}(f(\boldsymbol{x}_i) = y_i)}{N} \tag{33}$$

where $\mathbb{1}(\cdot)$ is the indicator function that is equal to 1 if the argument is true and 0 otherwise, "$f$" is the model, $\boldsymbol{x}_i$ is the input to the model and $y_i$ is the target label, and N is the total number of sample in the dataset.

The abovementioned selection criteria assume that we don't know the spurious attribute. However, some of the baselines we considered in our work (JTT and IRM) do assume that the spurious attributes are known and use them to train and select the best model that generalizes under spurious correlations. Thus for a fair comparison, in §F.2 and Fig. 7, we re-evaluate all the methods on all the datasets by relaxing this assumption. Now we select the configuration that performs the best on the following metric (see §4.2 for the definition of the individual metrics):

$$\text{selection criteria} = \frac{\text{Majority Group Accuracy} + \text{Minority Group Accuracy} + (1\text{-}\Delta\text{Prob})}{3} \tag{34}$$

### E.6  Setup: Mouli+CAD and other related baselines

As mentioned in Mouli and Ribeiro [25], given a set of attributes ($\mathcal{A}$), we train two models for each subset ($\alpha$) of the attributes ($\alpha \subset \mathcal{A}$) **(1)** model trained to predict the task label $Y$ while being invariant to the attributes in $\alpha$, **(2)** a model trained to predict the random label while being invariant to the attributes in $\alpha$. Using these two models we compute the score for every $\alpha$ as defined in Equation 8 in Mouli and Ribeiro [25]. Then we select the subset of attributes with the lowest score as spurious and train a classifier to predict the task label and be invariant to this subset of attributes. Similar to [25] we use counterfactual data augmentation (CAD) to impose invariance w.r.t to desired set of attributes (denoted as Mouli+CAD in all our experiments). We also experiment with using AutoACER to impose instead of CAD by using a causal effect equal to 0 for the attributes for which we want to impose invariance (see §4.2 for details). When using AutoACER we search over regularization hyperparameter $R$ from the set $\{1, 10, 100, 1000\}$. Unless otherwise specified, we use the same selection criteria to select the best model for comparison with other methods as described in Eq. 33.

### E.7  Setup: JTT baseline

We follow Algorithm 1 as described in Liu et al. [21] and train the model in two steps. First, we train a model with cross-entropy loss over all the examples (ERM). For Syn-Text-Unobs-Conf dataset, we tune the number epoch to train the model from set $\{4, 8\}$ using the model selection criteria defined later in step 2 of this method. For MNIST34 and Twitter-AAE we keep the number epochs fixed at 8 and 10 respectively for this first step. In the next step, we upsample the examples in the dataset that the ERM model predicted wrong in the first step by a factor of $\lambda_{up}$. We tuned the hyperparameter $\lambda_{up}$ over the set $\{2, 4, 8\}$ for Syn-Text-Unobs-Conf dataset and over set $\{2, 4\}$ for both MNIST34 and Twitter-AAE datasets. We train the final model using the oversampled dataset for 20 epochs for Syn-Text-Unobs-Conf, 20 epochs for MNIST34, and 10 epochs for Twitter-AAE dataset. Unless otherwise specified, we use the same selection criteria to select the best model using the validation dataset as described in Eq. 33.

### E.8  Setup: IRM baseline

We implement the IRMv1 to train the model using IRM objective as described in Arjovsky et al. [1]. We use the cross-entropy loss for estimating the prediction error ($R^e$ in the IRMv1 equation). We compute the norm of the gradient term directly by squaring the gradient of cross-entropy loss w.r.t. the dummy classifier $w = 1.0$ as described in the IRMv1 equation. We fine-tune the regularization coefficient $\lambda$ over the set $\{1e15, 5e15\}$ for Syn-Text-Unobs-Conf dataset, $\{1e14\}$ for MNIST34 dataset and $\{1e14, 5e14, 1e15\}$ for the Twitter-AAE dataset. We used such high regularization to scale up the gradient penalty term to have the same scale as the cross-entropy loss term which otherwise would have dominated the training objective. Unless otherwise specified, we use the same selection criteria to select the best model using the validation dataset as described in Eq. 33.

Table 6: **Estimated Causal Effect for *spurious* attribute in Syn-Text-Obs-Conf dataset:** The causal effect is estimated on multiple datasets with varying predictive correlation ($\kappa$). The true causal effect of spurious attribute is 0. In our experiment, we used two different causal effect estimators, *Direct* and *Riesz*. Each of them has two different ways to select the best estimate, using validation loss or validation accuracy (see §E.4 for details). The causal effect from *Direct* estimator is close to the correct value 0 for lower predictive correlation ($\kappa \leq 0.7$) but the error increases as $\kappa$ increases. The causal effect for *causal* and *spurious* attribute is identifiable in this dataset since the *confound* attribute is a sufficient backdoor adjustment. *Riesz* estimator learns a common representation that approximates the backdoor adjustment, thus, is expected to perform better even under a high value of $\kappa$. We observe *Riesz(loss)* estimator has the lowest or equivalent estimation error compared to other estimators for all values of $\kappa$.

| $\kappa$ | True | Direct | | Riesz | |
| --- | --- | --- | --- | --- | --- |
| | | Direct(acc) | Direct(loss) | Riesz(acc) | Riesz(loss) |
| 0.5 | 0 | -0.01 | -0.00 | 0.01 | **0.00** |
| 0.6 | 0 | -0.01 | **0.00** | 0.01 | 0.03 |
| 0.7 | 0 | **0.00** | 0.01 | 0.02 | 0.03 |
| 0.8 | 0 | 0.05 | 0.08 | 0.04 | **0.03** |
| 0.9 | 0 | 0.10 | 0.16 | 0.20 | **0.05** |
| 0.99 | 0 | 0.17 | 0.22 | 0.26 | **0.15** |

Table 7: **Estimated Causal Effect for *spurious* attribute in Syn-Text-Unobs-Conf dataset:** The causal effect is estimated on multiple datasets with varying predictive correlation ($\kappa$). The true causal effect of spurious attribute is 0. In our experiment, we used two different causal effect estimators, *Direct* and *Riesz*. Each of them has two different ways to select the best estimate, using validation loss or validation accuracy (see §E.4 for details). The data-generating process for this dataset is given by DGP-3 (Fig. 2) where the *confound* attribute is the unobserved variable (see §E.1 for details) and the identifiability of causal effect is not guaranteed. Thus, as expected, both the methods *Direct* and *Riesz* have a high estimation error and the error increases as the value of $\kappa$ increases. Among both, the Direct estimator performs better than Riesz for this dataset. Our method uses this estimated causal effect to regularize the classifier. We observe that our method is robust to this error in the estimation of causal effect and trains a classifier that performs better on average group accuracy and has significantly lower $\Delta$Prob compared to other baselines (see §4.4, F.2 and Fig. 4a, 14, 11 for discussion).

| $\kappa$ | True | Direct | | Riesz | |
| --- | --- | --- | --- | --- | --- |
| | | DE(acc) | DE(loss) | Riesz(acc) | Riesz(loss) |
| 0.5 | 0 | **-0.01** | **-0.01** | **0.01** | 0.06 |
| 0.6 | 0 | **0.10** | 0.13 | 0.16 | 0.19 |
| 0.7 | 0 | 0.20 | 0.24 | 0.29 | 0.31 |
| 0.8 | 0 | 0.38 | **0.37** | 0.43 | 0.42 |
| 0.9 | 0 | **0.49** | **0.49** | 0.57 | 0.58 |
| 0.99 | 0 | **0.65** | 0.67 | 0.67 | 0.70 |

### E.9 Computing Resources

We use an internal cluster of P40, P100, and V100 Nvidia GPUs to run all the experiments. We run each experiment for 3 random seeds and report the mean and standard error (using an error bar) for each of the metrics in our experiment of Stage 2 and other baselines and report the mean over 3 random runs for all the experiments in Stage 1 (Causal Effect Estimation).

## F    Additional Empirical Results

### F.1    Extended Evaluation of Stage 1: Detecting Spurious Attribute

**Causal Effect Estimator Recommendation.**     In this work we evaluate the performance of 2 different causal effect estimators — *Direct* and *Riesz* — each with two different ways of selecting the best mode either using validation loss or accuracy (see §E.4 for details) on 6 different datasets.

Table 8: **Estimated Causal Effect for *causal* attribute in Syn-Text-Unobs-Conf dataset:** The causal effect is estimated on multiple datasets with varying predictive correlation ($\kappa$). The true causal effect of spurious attribute is 0.29. In our experiment, we used two different causal effect estimators, *Direct* and *Riesz*. Each of them has two different ways to select the best estimate, using validation loss or validation accuracy (see §E.4 for details). The increasing value of $\kappa$ indicates the increasing predictive correlation of *spurious* attribute in the dataset (see E.1 for details). Even though there is no guarantee of the identifiability of causal effect, both the methods estimate almost correct causal effect for the *causal* feature. On the other hand, the previous method Mouli+CAD will incorrectly identify the *causal* feature as spurious for this dataset and thus will incorrectly impose invariance w.r.t to *causal* attribute. (see Fig. 5a and §F.t for more discussion).

| | | Direct | | Riesz | |
|---|---|---|---|---|---|
| $\kappa$ | True | DE(acc) | DE(loss) | Riesz(acc) | Riesz(loss) |
| 0.5 | 0.29 | 0.31 | 0.25 | 0.33 | **0.29** |
| 0.6 | 0.29 | **0.29** | 0.30 | 0.27 | 0.32 |
| 0.7 | 0.29 | 0.27 | **0.30** | 0.29 | **0.30** |
| 0.8 | 0.29 | 0.33 | 0.30 | 0.34 | **0.29** |
| 0.9 | 0.29 | 0.32 | 0.31 | 0.32 | **0.29** |
| 0.99 | 0.29 | 0.31 | 0.31 | 0.30 | **0.29** |

Table 9: **Estimated Causal Effect for spurious attribute (*rotation*) in MNIST34 dataset:** The causal effect is estimated on multiple datasets with varying predictive correlation ($\kappa$). The true causal effect of spurious attribute is $0$. In our experiment, we used two different causal effect estimators, *Direct* and *Riesz*. Each of them has two different ways to select the best estimate, using validation loss or validation accuracy (see §E.4 for details). The data-generating process of this dataset is given by DGP-2 (Fig. 2) and the causal effect is identifiable using only observational data (see Prop 3.1). The ground truth causal effect of spurious attribute is $0$. Overall the *Riesz(loss)* estimator has a lower or equivalent error in the estimation of the causal effect for most of $\kappa$ except very high $\kappa = 0.99$. As the predictive correlation ($\kappa$) of spurious attribute increases the error in the causal effect estimate of spurious attribute increases. When training the classifier, we show that our method (AutoACER) is robust to this error in the estimation of the causal effect of spurious attribute (see §4.4, F.2 and Fig. 4b, 15, 12 for discussion).

| | | Direct | | Riesz | |
|---|---|---|---|---|---|
| $\kappa$ | True | DE(acc) | DE(loss) | Riesz(acc) | Riesz(loss) |
| 0.5 | 0 | 0.02 | 0.02 | **0.01** | **-0.01** |
| 0.6 | 0 | 0.04 | 0.03 | **-0.01** | 0.06 |
| 0.7 | 0 | **0.05** | **0.05** | -0.10 | **0.05** |
| 0.8 | 0 | 0.07 | 0.06 | -0.07 | **0.04** |
| 0.9 | 0 | 0.11 | 0.15 | **0.09** | 0.20 |
| 0.99 | 0 | 0.28 | **0.27** | 0.50 | 0.31 |

Table 6 (Syn-Text-Obs-Conf), Table 7, 8 (Syn-Text-Unobs-Conf), Table 9(MNIST34) and Table 10 (Twitter-AAE and Civil-Comments) summarizes the performance of different causal effect estimator on these datasets. Different datasets bring different data-generating processes which bring different challenges for these estimators since the causal effect may or may not be identifiable (see individual tables for each dataset for discussion). We observe that one particular setting of the Riesz estimator i.e. *Riesz(loss)* performs consistently better or comparable to other estimators across the dataset. There are certain setting, especially at the high predictive correlation ($\kappa$) where even *Riesz(loss)* doesn't perform well and have a large error in the estimation of treatment effect. But we will show in §F.2 that our method AutoACER is robust to error in the estimation of the causal effect.

**Baseline Mouli+CAD can incorrectly detect causal attribute as spurious.** When given a spurious attribute we have seen that Mouli+CAD can sometimes fail to detect the spurious attribute (see §4.3 for discussions). But sometimes this Mouli+CAD can make an even more severe mistake — detect a causal attribute as spurious. Fig. 5a demonstrate on such failure in Syn-Text-Unobs-Conf dataset. When both the attribute— causal and spurious — is given to Mouli+CAD, it incorrectly identifies both the attributes as spurious for $\kappa \leq 0.7$ (shown by the red curve having the lowest score).

Table 10: **Estimated Causal Effect for spurious attribute in Twitter-AAE, Civil-Comments (Race), Civil-Comments (Gender) and Civil-Comments (Religion) dataset:** The causal effect is estimated on multiple version of each datasets with varying predictive correlation ($\kappa$). In our experiment, we used two different causal effect estimators, *Direct* and *Riesz*. Each of them has two different ways to select the best estimate, using validation loss or validation accuracy (see §E.4 for details). In Twitter-AAE dataset the spurious attribute in *race*, in Civil-Comments (Race) the spurious attribute is also *race*, in Civil-Comments (Gender) the spurious attribute is *gender* and in Civil-Comments (Religion) the spurious attribute is *religion*. The prediction task in Twitter-AAE dataset is sentiment classification and in Civil-Comments datasets it is toxicity prediction. The true causal effect of spurious attribute is unknown in every dataset. But we expect the spurious attribute to have a 0 causal effect on the task label since changing the race/gender/religion of a person should not change the sentiment or toxicity of the sentence. The data-generating process for all these dataset is given by DGP-1 (Fig. 2), where the attribute "A" is the spurious attribute (see §E.1 for details). The causal effect for any attribute "A" is identifiable given access to observational data and interventional distribution $P(X|do(A))$ (Prop 3.1). Since *Riesz* estimator assumes access to the counterfactual distribution $\mathcal{Q} := P(\boldsymbol{x}_{a'}|\boldsymbol{x}_a)$ (see §E.4), we expect it to have lower estimation error than *Direct*. Surprisingly, *Direct* performs comparably to *Riesz* estimator in all the datasets. When training the classifier, we show that our method (AutoACER) is robust to error in the estimation of the causal effect of spurious attribute (see §4.4, F.2 and Fig. 4c, 16, 13 for discussion).

| | | Direct | | Riesz | |
|---|---|---|---|---|---|
| Dataset | $\kappa$ | DE(acc) | DE(loss) | Riesz(acc) | Riesz(loss) |
| Twitter-AAE | 0.5 | **0.01** | **0.01** | 0.03 | 0.02 |
| | 0.6 | **0.07** | **0.07** | 0.12 | **0.07** |
| | 0.7 | **0.12** | **0.12** | **0.12** | **0.12** |
| | 0.8 | 0.21 | 0.20 | 0.18 | **0.17** |
| | 0.9 | 0.32 | 0.27 | **0.25** | 0.27 |
| | 0.99 | 0.35 | **0.31** | 0.40 | 0.37 |
| Civil-Comments (Race) | 0.5 | **0.02** | **0.02** | 0.06 | 0.06 |
| | 0.6 | **0.04** | **0.04** | 0.09 | 0.09 |
| | 0.7 | 0.07 | **0.05** | 0.08 | 0.09 |
| | 0.8 | **0.08** | **0.08** | 0.09 | 0.11 |
| | 0.9 | **0.15** | **0.15** | **0.15** | 0.18 |
| | 0.99 | **0.22** | **0.22** | 0.23 | 0.27 |
| Civil-Comments (Gender) | 0.5 | **0.00** | **0.00** | 0.03 | 0.01 |
| | 0.6 | **0.01** | **0.01** | 0.04 | 0.05 |
| | 0.7 | 0.03 | **0.02** | 0.09 | 0.09 |
| | 0.8 | **0.04** | 0.05 | 0.06 | 0.12 |
| | 0.9 | **0.09** | **0.09** | 0.12 | 0.12 |
| | 0.99 | 0.13 | **0.12** | 0.15 | 0.18 |
| Civil-Comments (Religion) | 0.5 | 0.01 | **0.00** | **0.00** | **0.00** |
| | 0.6 | **0.02** | 0.03 | **0.02** | **0.02** |
| | 0.7 | 0.02 | 0.03 | **0.01** | 0.04 |
| | 0.8 | 0.04 | 0.04 | 0.04 | **0.03** |
| | 0.9 | 0.06 | 0.06 | 0.06 | **0.05** |
| | 0.99 | **0.07** | 0.08 | 0.09 | 0.08 |

For $\kappa > 0.7$, again it incorrectly detects the true *causal* attribute as spurious. If we only give the causal attribute as input to Mouli+CAD, again causal attribute (shown in orange) will be detected as spurious attribute since it has a lower score than the subset with no attribute (blue curve) for all values of $\kappa$. But if we look at the causal effect estimate of causal attribute in Syn-Text-Unobs-Conf dataset it is close to the correct value 0.29 for all values of $\kappa$ even though the identifiability of causal effect for this dataset is not guaranteed (see Table 8). Thus using a continuous measure of the spuriousness of a attribute using the causal effect estimates can help mitigate the limitation of discrete measures (used in Mouli+CAD) like false positive and false negative detection of spurious attributes.

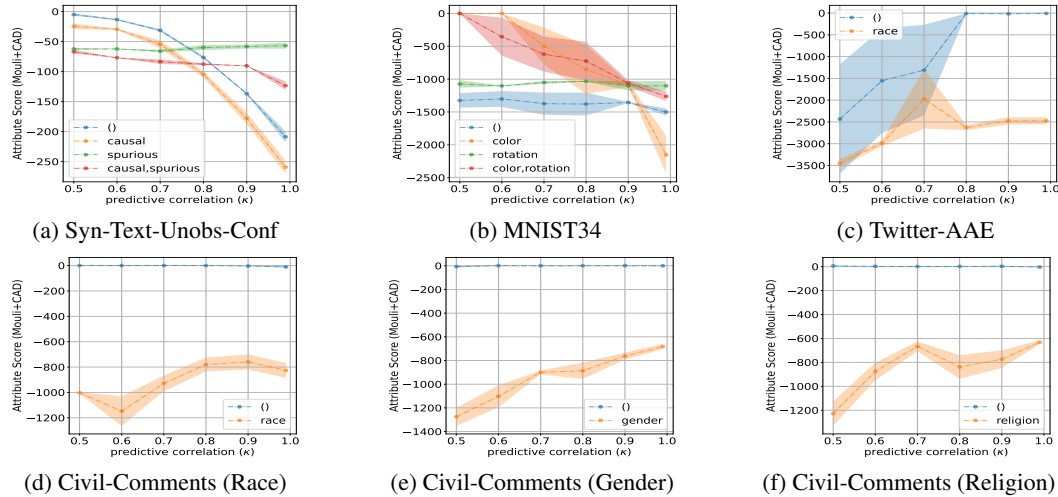

| (a) Syn-Text-Unobs-Conf | (b) MNIST34 | (c) Twitter-AAE |
| (d) Civil-Comments (Race) | (e) Civil-Comments (Gender) | (f) Civil-Comments (Religion) |

Figure 5: **Score used by Mouli+CAD for determining the spurious attribute.** Given a tuple of attributes in a given dataset, the baseline method Mouli+CAD defines a score for every subset of attribute and determines the least scoring subset as spurious. The empty subset "()" denotes that no attribute is considered spurious. The x-axis shows different levels of predictive correlation ($\kappa$) in the dataset and the y-axis shows the score assigned by Mouli+CAD to different subsets of attributes. We include some additional attribute not considered in Fig. 3 for Syn-Text-Unobs-Conf and MNIST34 datasets and summarize the score for Civil-Comments datasets. In **(a)**, we include the score for all the subsets of attributes (*causal* and *spurious*) in the Syn-Text dataset. When given all the attribute at the same time, Mouli+CAD will incorrectly detect both the attribute (causal and spurious) as spurious shown by the red curve for $\kappa < 0.7$ and the *causal* attribute as spurious for $\kappa > 0.8$. Thus Mouli+CAD will completely fail in determining the correct spurious attribute for Syn-Text dataset. When only given the *causal* attribute in the dataset, it will again incorrectly detect causal attribute as spurious since it will have a lower score compared to the subset with no attribute as shown by the orange curve being always lower than the blue curve. In **(b)**, we again include the score for every subset of attributes— rotation (spurious) and color (causal) which were partially included in Fig. 3. Again, Mouli+CAD will fail to identify any attribute as spurious for all $\kappa \leq 0.9$ (shown by the blue curve with the lowest score) and will incorrectly detect the causal attribute (color) as spurious for $\kappa = 0.99$ (shown by the orange curve). For **(c)** Twitter-AAE, **(d)** Civil-Comments (Race), **(e)** Civil-Comments (Gender) and **(f)** Civil-Comments (Religion) Mouli+AutoACER is able to correctly identify the spurious attribute shown by the orange curve with the lowest score. In Fig. 6, we show that Mouli+CAD doesn't always impose correct invariance with respect to spurious attribute and might fail to detect the correct spurious attribute once the correct invariance is imposed. See §F.1 for further discussions.

**Spuriousness Score used by Mouli+CAD can be misleading.** Given a set of attributes, Mouli+CAD creates multiple classifiers where they impose invariance with respect to different subsets of attributes. Using these invariant classifiers, they define a *spuriousness* score for all the subsets of attributes. Then the subset of attributes with a minimum score is selected as spurious. We observe in our experiment that Mouli+CAD when training the invariant classifier is not able to completely enforce invariance. Thus the score defined using the invariant model might be misleading. Fig. 6 shows one such instance of failure. The second row of Fig. 6c shows that Mouli+CAD doesn't impose correct invariance w.r.t the spurious *race* attribute (shown by high $\Delta$Prob of the orange curve). Next, we use AutoACER to enforce correct invariance w.r.t race attribute by regularizing the classifier with zero causal effect for race attribute. This enforces correct invariance shown by low $\Delta$Prob in the second row of Fig. 6f. As a result, we observe that the score for the spurious attribute (shown by the orange curve in the first row of Fig. 6f), is higher than blue. Thus enforcing the correct invariance Mouli+CAD will fail to detect the race attribute as spurious since the blue curve has a lower score than orange for all values of $\kappa < 0.9$.

### F.2 Extended Evaluation of Stage 2: AutoACER and other baselines assuming the Spurious attribute is known

In §4.4 and Fig. 4 we compared AutoACER with other baselines where we used overall accuracy as the selection criteria to select the best model (see Eq. 33). However, some of the baselines (IRM and JTT) assume that spurious attributes are known and use this information to train and select the best model that generalizes under spurious correlations. Thus, in this section for a fair comparison, we re-evaluate all the methods on all the datasets again by providing them with the knowledge of spurious attributes. Refer to Eq. 34 for details on how we select the best model with this information. Fig. 7 summarizes the result for AutoACER and other baseline with this selection criteria.

**Other evaluation metrics for Syn-Text-Unobs-Conf, MNIST34 and Twitter-AAE dataset**   We extend the evaluation in Fig. 7, by adding the comparison of the performance of AutoACER and other baselines over these accuracy metrics for Syn-Text-Unobs-Conf, MNIST34, and Twitter-AAE datasets in Fig. 8. Overall, AutoACER performs comparable or better than other baselines for Syn-Text-Unobs-Conf and MNIST34 datasets. In Twitter-AAE dataset, Mouli+CAD performs better in worst group and minority group accuracy than AutoACER. But AutoACER has significantly lower $\Delta$Prob compared to other baselines signifying that the learned classifier is invariant to the spurious attribute.

**Evaluation of AutoACER on Civil-Comments dataset.**   We evaluate our method on Civil-Comments dataset — a real-world dataset where given a sentence the task is to predict the toxicity of the sentence. We have three different subsets of this dataset, Civil-Comments (Race), Civil-Comments (Gender), and Civil-Comments (Religion) where the attribute *race*, *gender* and *religion* are the spuriously correlated with task label (see §E.1 for details). Fig. 10 compares AutoACER with other baselines over a number of different metrics. Unlike Twitter-AAE dataset where Mouli+CAD gave significant gains in the average group accuracy, AutoACER performs comparably to Mouli+CAD and improves over ERM over this metric on all the subsets. Although, $\Delta$Prob is low for all the baseline methods, AutoACER again gives a classifier with much lower $\Delta$Prob, especially for high values of predictive correlation ($\kappa$). On all subsets of Civil-Comments dataset, AutoACER also performs comparably to other baselines over other accuracy metrics — minority group accuracy and worst group accuracy — used to evaluate the efficacy of generalization over spurious attribute [30].

**Counterfactual Data Augmentation is not sufficient for imposing correct invariance.**   Fig. 7 gives us a hint that Mouli+CAD might not be sufficient for learning a model that is invariant to spurious attribute. For Twitter-AAE dataset (see Fig. 7c) and Civil-Comments dataset (see Fig. 10), AutoACER has comparable or slightly lower average group accuracy than Mouli+CAD. But AutoACER has a significantly lower value of $\Delta$Prob than other baselines and thus is able to impose correct invariance w.r.t spurious attribute. Thus it seems that counterfactual data augmentation used in Mouli+CAD might be using some other mechanism to perform better on average group accuracy than imposing invariance w.r.t to spurious attribute. In Fig. 9 we combine Mouli+CAD and AutoACER to get the best out of both worlds (denoted as *Mouli+AutoACER +CAD*). As expected we observed that the final classifier has a significant gain in the average group accuracy and has $\Delta$Prob than ERM classifier. We leave the task of understanding how CAD gives better gains in average group accuracy to future work. See Fig. 9 for details of other additional details of experiments with CAD that might shed some light on this question.

**Robustness of AutoACER on error in detection of spurious attributes.**   AutoACER follows a two-step procedure to train a classifier that generalizes to spurious correlation. First, in Stage 1, it estimates the causal effect of the given attribute on the task label to determine a continuous degree of *spuriouness* for that attribute. Then in Stage 2, it regularizes the classifier proportional to the estimated causal effect with the goal of imposing invariance with respect to spurious attribute. Since AutoACER is heavily reliant on the estimate of causal effect for correctly enforcing the invariance with respect to spurious attribute we need to analyze its robustness to noise in the estimation of causal effect. For simpler classifiers, our 3.1 shows that we don't need a correct causal effect estimate to learn a classifier invariant to spurious correlation. For the case when we have two high dimensional variables – one causal and another spurious — as long as the ranking of the causal effect of causal and spurious attribute is correct AutoACER will learn a classifier completely invariant to spurious attribute. This result on a simpler classification task hinted that we don't need a completely correct

estimate of attribute to learn a more complicated invariant classifier trained on a real-world dataset. We conduct an experiment where we knowing use a range of non-zero causal effects for spurious attribute (ground truth causal effect is zero) to regularize the model using AutoACER. We observe that AutoACER is less sensitive to even large noise in the causal effect ($> 0.5$) on the metrics like average group accuracy and $\Delta$Prob for a large range of regularization strength (R). Even with a large error in the causal effect AutoACER trains a classifier with high average group accuracy and low $\Delta$Prob. Fig. 11, 12 and 13 summarizes the result for Syn-Text-Unobs-Conf, MNIST34 and Twitter-AAE dataset. Also, we show the robustness of AutoACER when using the causal effect estimates from different estimators we consider in our work in Fig. 14, 15 and 16 for these datasets.

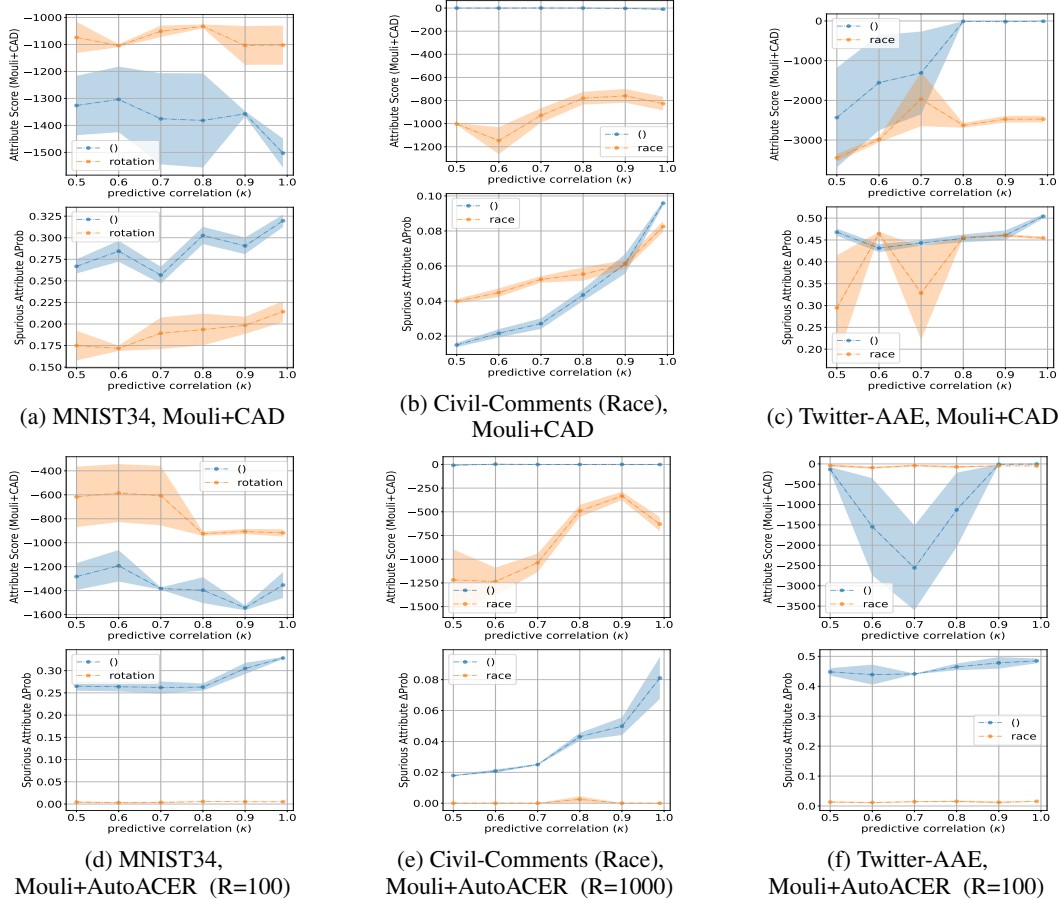

(a) MNIST34, Mouli+CAD

(b) Civil-Comments (Race), Mouli+CAD

(c) Twitter-AAE, Mouli+CAD

(d) MNIST34, Mouli+AutoACER (R=100)

(e) Civil-Comments (Race), Mouli+AutoACER (R=1000)

(f) Twitter-AAE, Mouli+AutoACER (R=100)

Figure 6: **Counterfactual Data Augmentation doesn't impose complete invariance:** The x-axis in every plot denotes the predictive correlation in the dataset, and the y-axis in the first row of every plot shows the score computed by the previous Mouli's method for a different subset of attributes. Different methods could be used to impose invariance w.r.t to a certain subset of attribute in the classifier and compute a score to quantify the *spuriousness* of the classifier. In **(a)**, **(b)** and **(c)** counterfactual data augmentation (CAD) is used to impose invariance whereas in **(d)**, **(e)** and **(f)** AutoACER is used to impose invariance. The empty subset "()" denotes that no attribute is considered spurious. Then the attribute with the lowest score is detected as spurious attribute. The y-axis in the second row of every plot measures the reliance of the classifier on spurious attributes on imposing invariance w.r.t to labeled attribute using ΔProb. The blue curve shows reliance on the spurious attribute when no invariance is imposed and the orange curve shows reliance on spurious attribute when we impose invariance w.r.t to spurious attribute. Sometimes CAD doesn't impose correct invariance in the classifier which may create mistakes in detecting the spurious attribute. For Civil-Comments (Race) and Twitter-AAE dataset the second row of **(b)** and **(c)** respectively shows that CAD is not able to impose correct invariance with respect to the spurious attribute shown by high ΔProb of spurious attribute in the classifier which is supposed to be invariant to spurious attribute (orange curve). AutoACER to impose the required invariance (second row in **(e)** and **(f)**) shown by ΔProb close to 0 for the orange curve. After the correct invariance is imposed using AutoACER, Mouli's score fails to determine the *race* attribute (which is actually spurious) as spurious for $\kappa \leq 0.9$ shown by a blue curve having a lower score in the first row of **(f)**. The relative ordering of score (first row) remains the same for MNIST34 and Civil-Comments (Race) after imposing invariance using either CAD (in **(a)**,**(b)**) or AutoACER (in **(d)**, **(e)**).

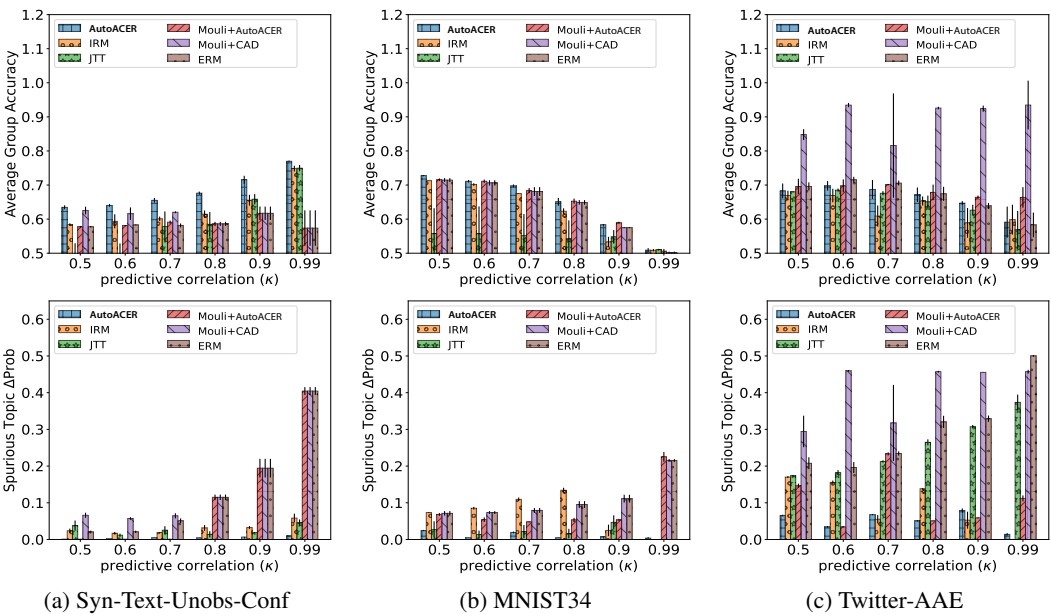

(a) Syn-Text-Unobs-Conf       (b) MNIST34       (c) Twitter-AAE

Figure 7: **Average group accuracy (top row) and ΔProb (bottom row) for different methods when the spurious attribute is known.** x-axis denotes predictive correlation ($\kappa$) in the dataset. We observe a similar trend as in Fig. 4 where none of the methods assumed that the spurious attribute is known. Compared to other baselines, **(a)** and **(b)** show that AutoACER performs better or comparable in average group accuracy and trains a classifier significantly less reliant to spurious attribute as measured by ΔProb. Mouli+CAD performs better in average group accuracy in **(c)** but relies heavily on spurious prop shown by high ΔProb whereas AutoACER performs equivalent to ERM on average group accuracy with lowest ΔProb among all. Refer to Eq. 34 for the model selection criteria used to select the best model and for further discussion see §F.2.

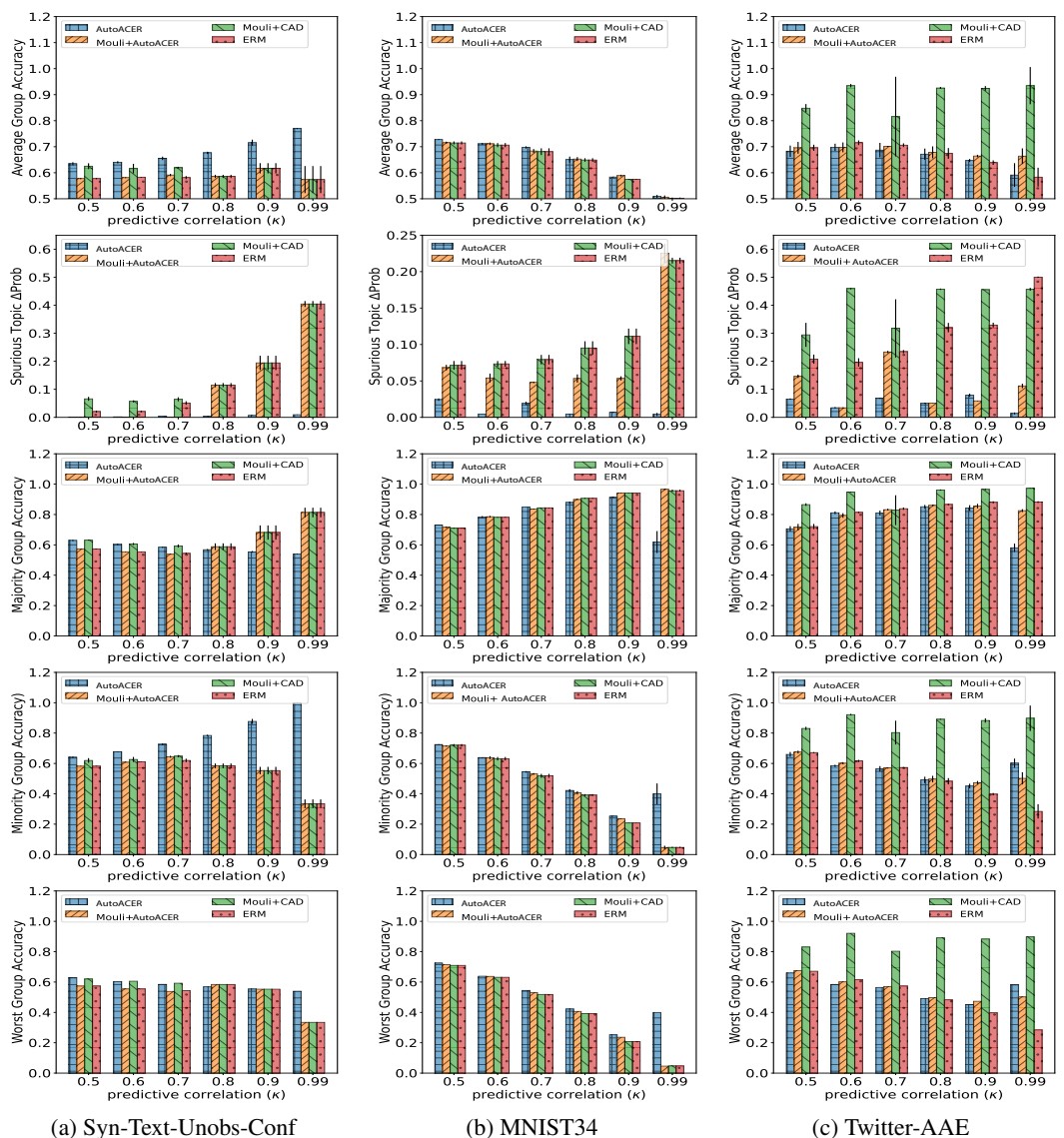

(a) Syn-Text-Unobs-Conf      (b) MNIST34      (c) Twitter-AAE

Figure 8: **Additional evaluation metrics.** We include three additional metrics used in the past work [30] to evaluate the generalization of a classifier to spurious correlation in addition to Average group accuracy and ΔProb as used in Fig. 7 — Majority group accuracy (third row), Minority Group Accuracy (fourth row), and Worst group accuracy (bottom row). x-axis denotes predictive correlation ($\kappa$) in the dataset. Compared to other baselines, **(a)** and **(b)** show that AutoACER performs better in minority group accuracy, comparable or better in the worst group accuracy, and trains a classifier significantly less reliant on spurious attribute as measured by ΔProb. The majority group's accuracy is expected to increase as the predictive correlation ($\kappa$) increases since the classifier could start using the spurious attribute for making predictions. In **(a)**, the majority group accuracy remains almost similar even as the $\kappa$ increases whereas it increases for other baselines. This shows that AutoACER trains a classifier that doesn't rely on spurious attribute even at high values of $\kappa$. Similarly, in **(b)**, the majority group accuracy of a classifier trained by other baselines is higher than AutoACER at higher values of $\kappa$ implying a higher reliance on spurious attribute. Mouli+CAD performs better in all group-wise accuracy in **(c)** but relies heavily on spurious prop shown by high ΔProb whereas AutoACER performs equivalent to ERM on all group-wise accuracy with lowest ΔProb among all. See §F.2 for further discussions and refer to Eq. 34 for the model selection criteria used to select the best model.

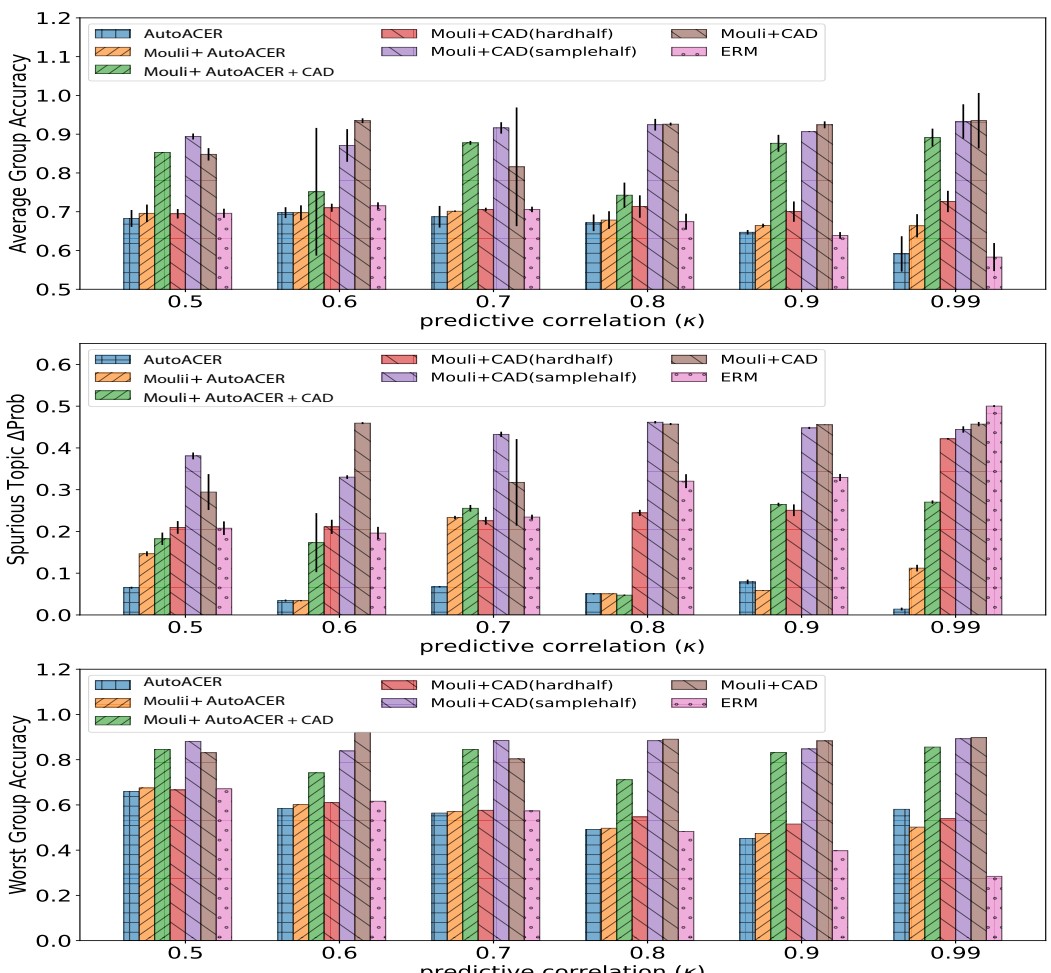

Figure 9: **Combining AutoACER and Mouli+CAD gives good trade-off in Accuracy and ΔProb:** The x-axis denotes predictive correlation ($\kappa$) in the dataset and the y-axis shows different evaluation metrics. Mouli+CAD performs better in average group accuracy whereas AutoACER performs better in ΔProb (see §F.2 and Fig. 7 for more discussion). Thus counterfactual data augmentation (CAD) might not be the best method to impose invariance with respect to a spurious attribute. Thus we augment Mouli+CAD with AutoACER (denoted as *Mouli+AutoACER +CAD* and shown by the green bar) to get the best out of both worlds. *Mouli+AutoACER +CAD* show similar increase in average group accuracy as Mouli+CAD and at the same time have lower ΔProb than Mouli+CAD and ERM. We experiment with two different settings where we halve the number of samples used to train the task classifier to test whether this gain in average group accuracy is due to increased sample size and thus the number of gradient updates when we perform CAD. Mouli+CAD (hardhalf) shows the setting when we reduce the initial dataset before augmentation to half and thus the effective dataset size and thus the number of gradient updates becomes the same after augmentation as used in other baselines. In this setting, we observe that the average group accuracy becomes equal to other baselines where we don't perform CAD. In the second setting, Mouli+CAD (samplehalf), we randomly subsample half of the counterfactually augmented dataset in every training iteration. Thus the effective number of gradient updates is the same as other baselines. In this setting, we observe similar behavior as Mouli+CAD in both average group accuracy and ΔProb metrics. These two experiments thus suggest that the number of gradient updates (better training) is not the main reason behind the gain of the average group accuracy when using counterfactual data augmentation. We leave the question for future work of understanding the mechanism behind how CAD is able to perform better on average group accuracy without completely imposing invariance with respect to spurious attributes. See §F.2 for further discussions and refer to Eq. 34 for the model selection criteria used to select the best model.

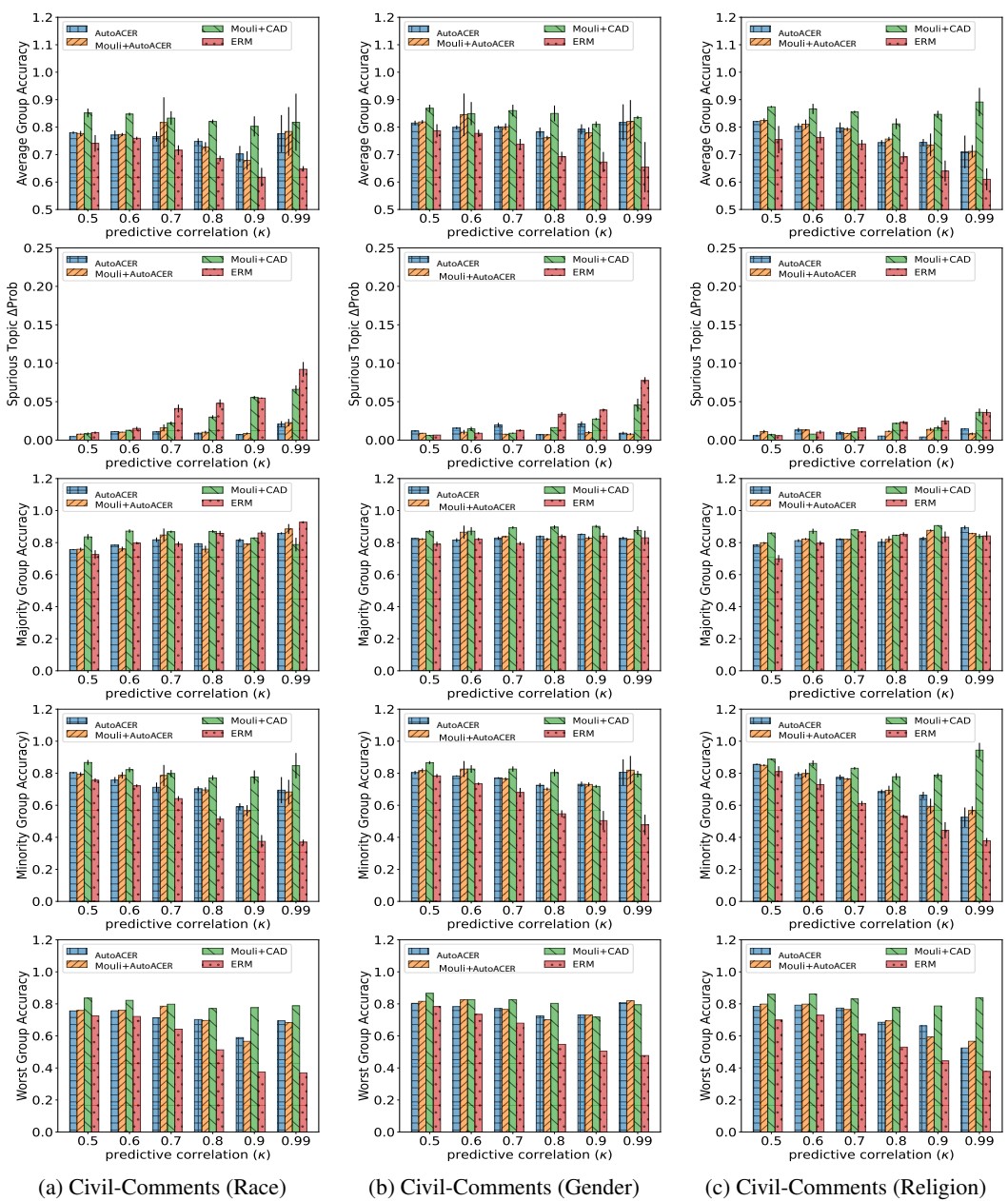

(a) Civil-Comments (Race)     (b) Civil-Comments (Gender)     (c) Civil-Comments (Religion)

Figure 10: **AutoACER and other baselines on Civil-Comments datasets:** Different columns show the results on different subsets of Civil-Comments dataset i.e. *race, gender* and *religion* (see E.1 for details dataset). Different rows show different evaluation metrics we consider to evaluate the performance. The x-axis denotes predictive correlation ($\kappa$) in the dataset. Unlike Twitter-AAE dataset (Fig. 8c and 7 where there was a significant gain in average group accuracy when using the Mouli+CAD, AutoACER performs comparably to Mouli+CAD on all the subsets of Civil-Comments datasets. Even though the $\Delta$Prob metric is already small for all the methods ($\leq 0.10$), AutoACER further reduces it close to 0. On all the individual group accuracy metrics AutoACER performs better than ERM and is comparable to other baselines. See §F.2 for more discussions and refer to Eq. 34 for the model selection criteria used to select the best model.

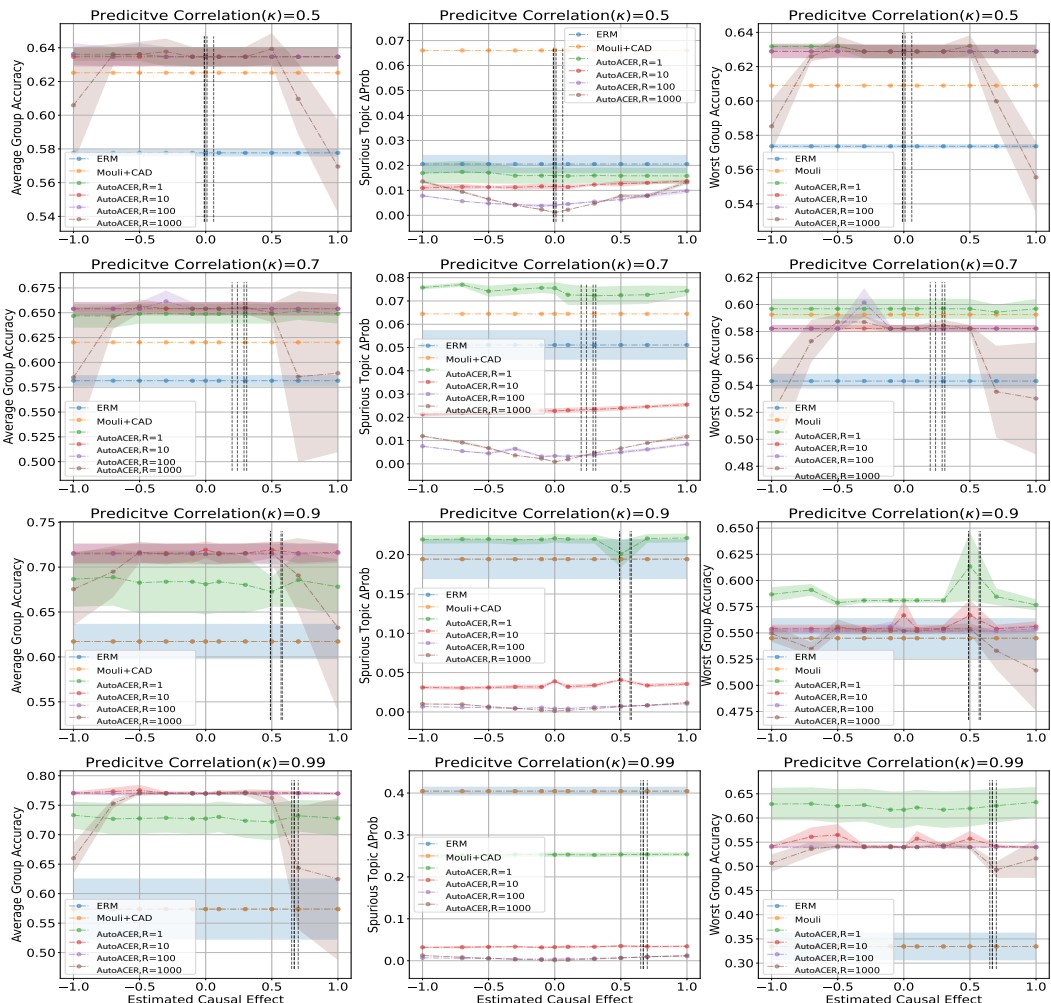

Figure 11: **Robustness of AutoACER w.r.t estimated causal effect on Syn-Text-Unobs-Conf:** Given a attribute, in Stage 1 our method AutoACER estimates the causal effect of attribute on the task label as a continuous measure of *spuriousness* of the attribute. Then in Stage 2, it regularizes the classifier proportional to the estimated causal effect. Since our model relies heavily on the estimate of the causal effect of the attribute it is important to analyze the robustness of AutoACER with noise in the estimation step. In this figure, we show the sensitivity of AutoACER with varying levels of noise in the causal effect estimate of the spurious attribute. The true causal effect of spurious attribute is 0 and the x-axis shows the value of the causal effect used in Stage 2. The different row shows the result for different levels of predictive correlation ($\kappa$). Different columns show different evaluation metrics — average group accuracy, $\Delta$Prob, and worst group accuracy. Different colored lines show different baselines or different regularization strengths ($R$) used by AutoACER. The dotted vertical black line in every plot shows the estimated causal effect using the Direct or Riesz estimator in Stage 1. We observe that AutoACER is less sensitive to the error in the estimated causal effect of spurious attribute. For example, for $\kappa = 0.99$ (last row), the green curve (AutoACER with $R = 10$) is almost constant for all values of causal effect used in Stage 2 for spurious attribute, has better average and worst group accuracy than ERM and Mouli+CAD and has a very low value of $\Delta$Prob. Even when we regularize the classifier to have a causal effect of 1 in Stage 2, the $\Delta$Prob remain close to zero demonstrating some form of *self-correction*. Theorem 3.1 hints a possible reason for this robustness — our method will always select the correct classifier (less reliant on spurious attribute) as long as the ranking of the causal effect of spurious and causal attribute is consistent up to some constant factor. See §F.2 for detailed discussion, Fig. 12 and 13 for similar analysis on MNIST34 and Twitter-AAE dataset respectively and refer to Eq. 34 for the model selection criteria used to select the best model.

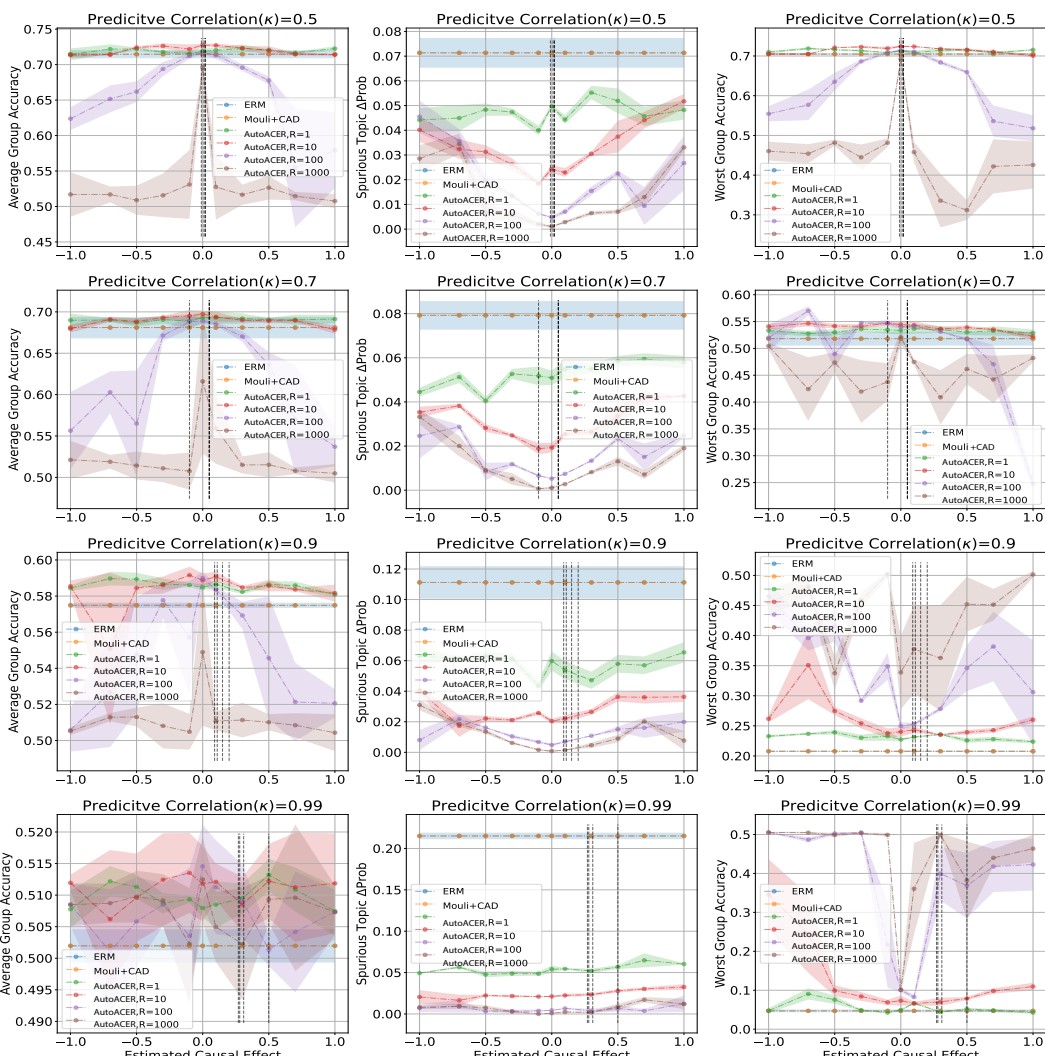

Figure 12: **Robustness of AutoACER w.r.t estimated causal effect on MNIST34:** In this figure, we show the sensitivity of AutoACER with varying levels of noise in the causal effect estimate of the spurious attribute. The true causal effect of spurious attribute (*rotation*) is 0 and the x-axis shows the value of the causal effect for this attribute used for regularization in Stage 2. The different row shows the result for different levels of predictive correlation ($\kappa$). Different columns show different evaluation metrics — average group accuracy, $\Delta$Prob, and worst group accuracy. Different colored lines in every plot show different baselines or different regularization strengths ($R$) used by AutoACER. The dotted vertical black line shows the estimated causal effect using the Direct or Riesz estimator in Stage 1. Similar to Fig. 11 we observe that AutoACER is robust to error in the estimated causal effect of the spurious attribute. For example, when $\kappa = 0.99$, for all the settings of AutoACER (with different regularization strength $R$), the average group accuracy is almost constant and better than the ERM and Mouli+CAD for all values of causal effect used for the spurious attribute. Similarly, the $\Delta$Prob is constant and lower than ERM and Mouli+CAD for all values of the estimated causal effect. For discussion on possible explanation see Fig. 11 and §F.2, and refer to Eq. 34 for the model selection criteria used to select the best model.

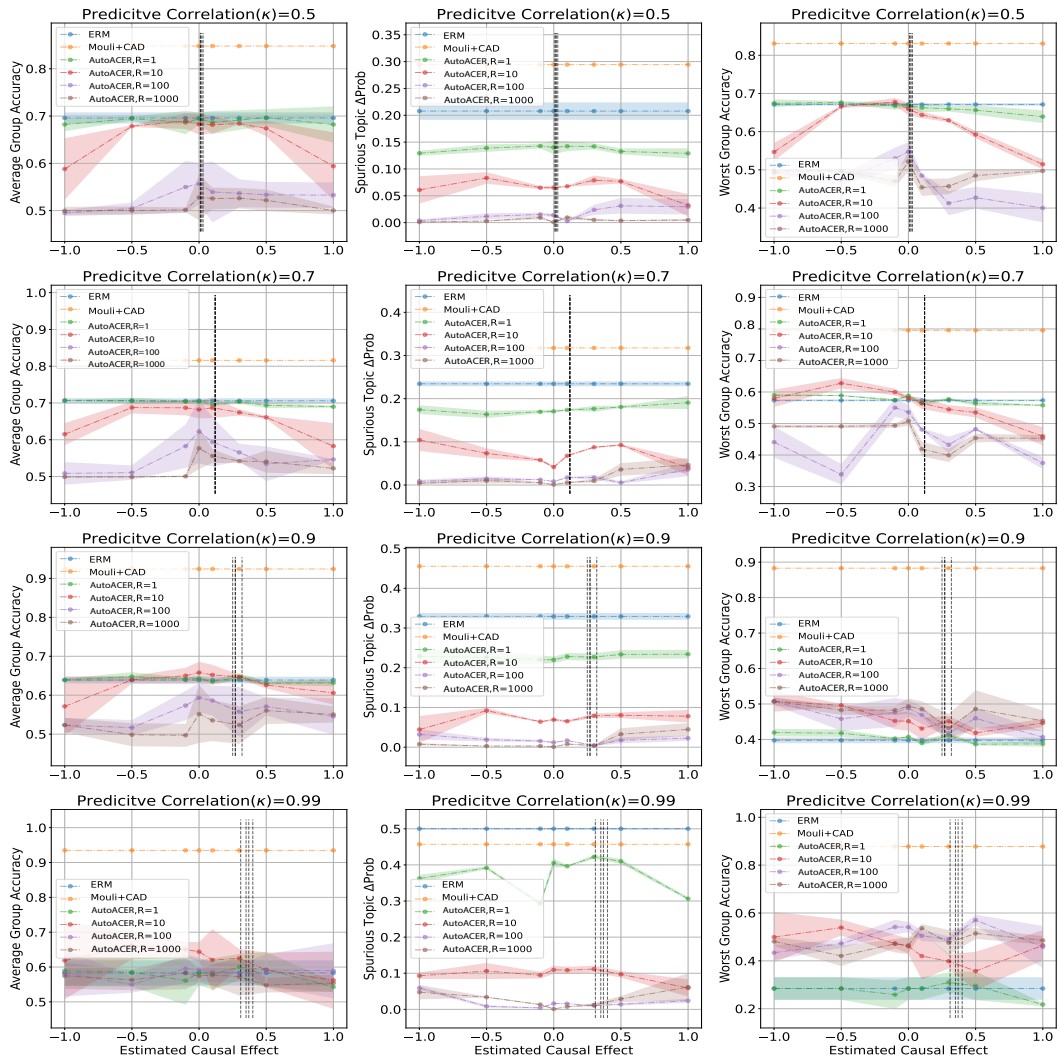

Figure 13: **Robustness of AutoACER w.r.t estimated causal effect on Twitter-AAE:** In this figure, we show the sensitivity of AutoACER with varying levels of noise in the causal effect estimate of the spurious attribute. The expected true causal effect of spurious attribute (*race*) is 0 and the x-axis shows the value of the causal effect for this attribute used for regularization in Stage 2. The different row shows the result for different levels of predictive correlation ($\kappa$). Different columns show different evaluation metrics — average group accuracy, $\Delta$Prob, and worst group accuracy. Different colored lines in every plot show different baselines or different regularization strengths ($R$) used by AutoACER. The dotted vertical black line shows the estimated causal effect using the Direct or Riesz estimator in Stage 1. Similar to Fig. 11 and 12, we observe that AutoACER is not sensitive to noise in the estimated causal effect of the spurious attribute. The average group accuracy and worst group accuracy remain almost constant with increasing noise in the estimate, though they are lower as compared to Mouli+CAD. But AutoACER is less sensitive to noise in estimated effect and is significantly lower than ERM and Mouli+CAD for all values of $\kappa$ in $\Delta$Prob metric. For further discussions see Fig. 11, 12 and §F.2, and refer to Eq. 34 for the model selection criteria used to select the best model.

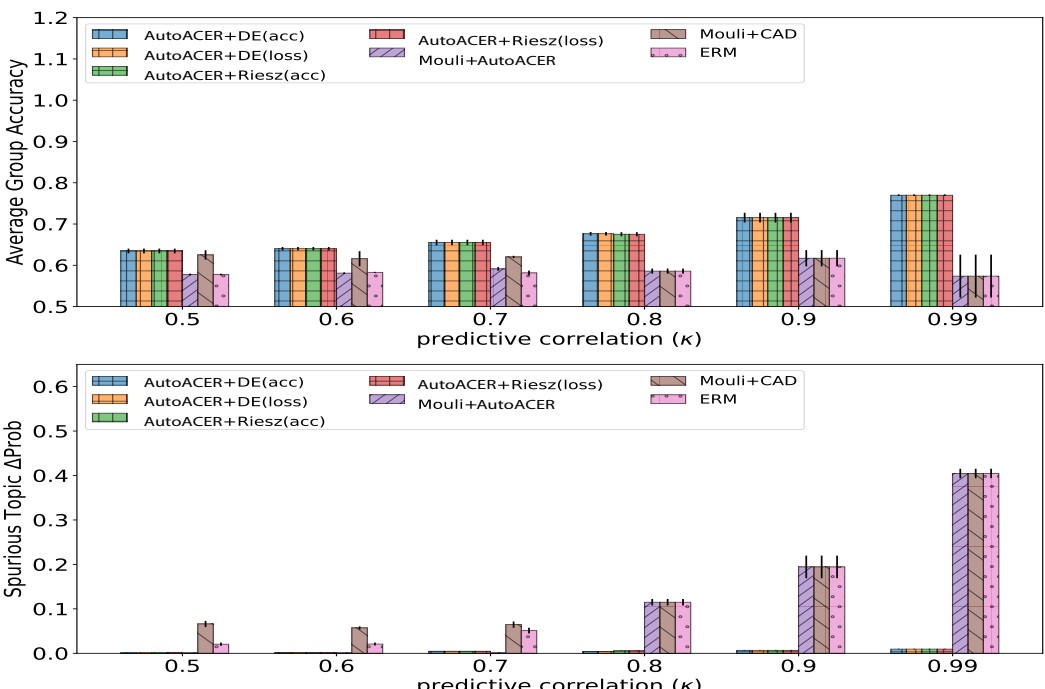

Figure 14: **Performance of AutoACER using different causal effect estimators for Syn-Text-Unobs-Conf:** AutoACER uses causal effect estimator to determine the *spuriousness* of a given attribute in Stage 1 and then regularize the classifier proportional to the estimated effect in Stage 2. In this work, we consider two different estimators — Direct and Riesz — each with two ways of selecting the best estimate (using validation loss or accuracy, see §E.4 for details). In Fig. 7a, we select the causal effect estimator that performs best in Stage 2. Thus it is not clear from Fig. 7a if any particular estimator is more suitable to estimate the causal effect in a particular dataset. In this figure, we show the performance of AutoACER using all the different estimators separately. $DE(\cdot)$ is the *Direct* estimator and $Riesz(\cdot)$ is the *Riesz* estimator with their two different settings. The x-axis shows the predictive correlation ($\kappa$) in the dataset and the y-axis shows average group accuracy and $\Delta$Prob in the top and bottom rows respectively. We observe that AutoACER performs equivalently when using any of the available estimators on both metrics. Compared to other baselines using any of the estimator AutoACER performs better in both average group accuracy and $\Delta$Prob. This demonstrates the robustness of the method in choosing different causal effect estimators and noise in the estimated causal effect for training a classifier that generalizes to spurious attributes. For more discussion see §F.2 and Fig. 15 and 16 for results on MNIST34 and Twitter-AAE dataset respectively. Refer to Eq. 34 for the model selection criteria used to select the best model.

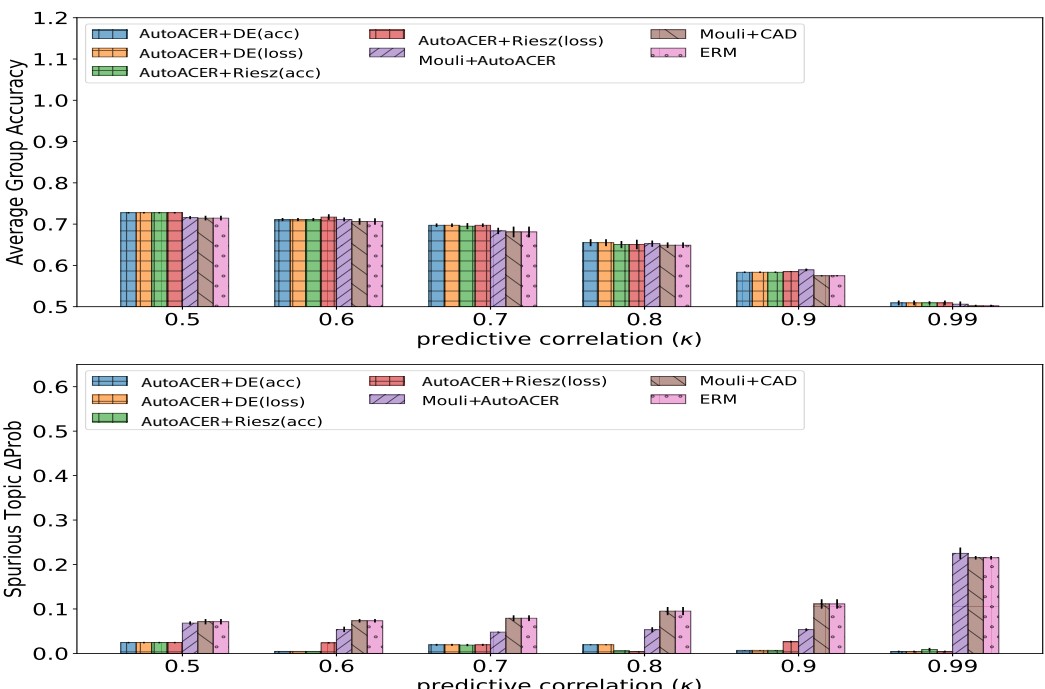

Figure 15: **Performance of AutoACER using different causal effect estimators for MNIST34:** In this work, we have considered four different causal effect estimators to determine the *spuriousness* of a given attribute (see §E.4). Fig. 7b aggregates the performance of AutoACER by selecting the estimator that gives best performance. Thus we expand the results in Fig. 7b to demonstrate the performance of AutoACER when using the causal effect estimate from different estimators individually. DE($\cdot$) is the *Direct* estimator and Riesz($\cdot$) is the *Riesz* estimator with their two different settings. The x-axis shows the predictive correlation ($\kappa$) in the dataset and the y-axis shows average group accuracy and $\Delta$Prob in the top and bottom rows respectively. We observe that AutoACER performs equivalently when using any of the available estimators on both metrics. Compared to other baselines using any of the estimator AutoACER performs better in both average group accuracy and $\Delta$Prob. This demonstrates the robustness of the method in choosing different causal effect estimators and noise in the estimated causal effect for training a classifier that generalizes to spurious attributes. For more discussion see §F.2 and Fig. 14 and 16 for results on Syn-Text-Unobs-Conf and Twitter-AAE dataset respectively. Refer to Eq. 34 for the model selection criteria used to select the best model.

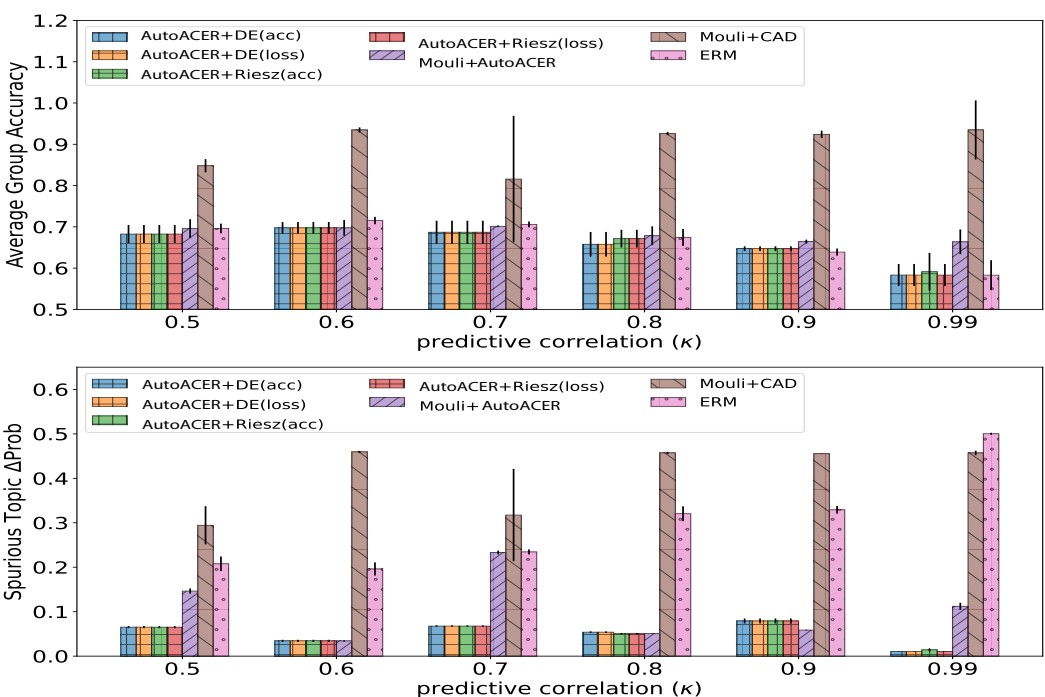

Figure 16: **Performance of AutoACER using different causal effect estimators for Twitter-AAE:** In this work, we have considered four different causal effect estimators to determine the *spuriousness* of a given attribute (see §E.4). Fig. 7c aggregates the performance of AutoACER by selecting the estimator that gives best performance. Thus we expand the results in Fig. 7c to demonstrate the performance of AutoACER when using the causal effect estimate from different estimators individually. DE($\cdot$) is the *Direct* estimator and Riesz($\cdot$) is the *Riesz* estimator with their two different settings. The x-axis shows the predictive correlation ($\kappa$) in the dataset and the y-axis shows average group accuracy and $\Delta$Prob in the top and bottom rows respectively. Similar to the trend in Fig. 14 and 15, AutoACER performs equivalently when using different causal effect estimators. For more discussions see §F.2 and refer to Eq. 34 for the model selection criteria used to select the best model.

