# OpenReview forum: "Causal Effect Regularization: Automated Detection and Removal of Spurious Correlations"
_NeurIPS.cc/2023/Conference — NeurIPS 2023 poster_

### Official Review · Reviewer_zxmG · 2023-07-02

**Soundness:** 3 good
**Presentation:** 4 excellent
**Contribution:** 3 good
**Rating:** 6
**Confidence:** 3

**Summary:**

In this paper, the authors proposed a causal effect regularization technique - CausalReg, which can effectively identify and remove the spurious but unknown attributes. CausalReg is robust to no-identifiable issues, finite sample error, and noise. The experimental results demonstrate the superior performance of CausalReg in identifying and exclude the spurious attributes.

**Strengths:**

The paper is well-written, with a strong theoretical and experimental backend to justify the ability of the CausalReg to automatically remove the spurious correlated features.  Specifically, in theory, the authors have proved that under certain data generation structures, the causal effects are guaranteed to be identified. In practice,  the algorithm demonstrates significant control of the spurious correlated feature compared to other approaches (ERM, Mouli, CAD).

**Weaknesses:**

Major:

1. Precision and Recall experiment is missing: Some experiments should be done to reflect the precision and recall of the discovery of spurious and causal attributes. Specifically, the paper claims its advantage compared to other methods that require binning with thresholds to identify the spurious and causal effect; the authors then should compare with these algorithms and demonstrate comparable results even without binning techniques.

2. The mathematical definition of spurious attributes is not completely clear.



Minor:

1. The figures seem to be twisted
2. Legends in Fig 3 is not clear





**Questions:**

1. The effect of CausalReg seems to rely on the ML model in stage 1 can learn a fairly good causal relationship. Is it true? If so, how much would the authors anticipate the performance change when adapting other deep learning models in Stage 1?

2. How was the counterfactual distribution learned (mentioned in line 176, equation (2))

3. What would happen if some causal features were missing.





**Limitations:**

As the authors have mentioned, one major limitation is that the theoretical guarantee of causal discovery works only under certain data generation procedures.

---

> ### Author Rebuttal · Authors · 2023-08-09
>
> > **W1: Precision and Recall experiment is missing: Some experiments should be done to reflect the precision and recall of the discovery of spurious and causal attributes.** (part of the remark left out for brevity)
>
> A: We thank the reviewer for suggesting this experiment. However, classifying an attribute as spurious or causal based on its causal effect will require some thresholding criteria. To the best of our knowledge, no such previous work exists that uses the causal effect of an attribute to classify them as causal or spurious. One exception is the Mouli+CAD baseline that defines a score for every attribute to classify them as spurious or not. We observe that for two datasets, Syn-Text-Unobs-Conf and MNIST34, Mouli+CAD is not able to detect the spurious attribute correctly. Comparatively, for our method even though the causal effect estimated for the spurious feature is not exactly zero, our method performs better than the Mouli+CAD baseline since we don't perform any hard thresholding for the spurious feature (see Section 4 for details).
>
>
> That said, we have plotted the AUROC curve (see Fig 2 in the uploaded document in the global comment of rebuttal) for the identification of causal and spurious attributes by thresholding the causal effect estimated for a different feature. In the plot, the orange curve shows the AUROC curve for Riesz Estimator and the blue curve is for the Direct Estimator (see Section 3.2, Stage 1 and Appendix E.4 for details on the estimator). Riesz Estimator performs better than the Direct as shown by the higher area under the curve. The “star” marked in the plot shows the TPR (true positive rate) and FPR (false positive rate) of the identification of causal and spurious attributes by the Mouli+CAD method.
>
>
>
>
>
>
> > **W2: The mathematical definition of spurious attributes is not completely clear.**
>
> A: We thank the reviewer for pointing this out. We have defined the spurious attribute in Lines 94-96 stating “We use the fact that changing spurious attributes will not lead to a change in the task label i.e. they have zero causal effect on the task label y.” Thus a spurious attribute has zero true causal effect on the task label $y$. We will make this more precise in the camera-ready submission if accepted.
>
>
>
> > **W3: The figures seem to be twisted**
>
> > **W4: Legends in Fig 3 is not clear**
>
> A: We thank the reviewer for pointing this out. We will make the necessary changes in the camera-ready submission if accepted.
>
>
>
>
> > **Q1: The effect of CausalReg seems to rely on the ML model in stage 1 can learn a fairly good causal relationship. Is it true?** (part of the remark left out for brevity)
>
> A:  It is true that having a good estimate of the causal effect will definitely help the subsequent steps of our method. But since we use a continuous value of the causal effect of the attribute for regularization instead of hard thresholding, our method allows for error in the first step not to affect the later steps severely. Thus our method is robust to error in the estimation of causal effect in Stage 1.
>
> In stage 1 of our experiments, we used two different deep-learning-based causal effect estimators, Direct and Reisz (see Section E.4 in the appendix for details). In practice, one can replace these estimators with any reasonable causal effect estimator of their choice based on the dataset considered. We observe that both estimators give a similar estimate on all the datasets whereas Riesz performs relatively better when the predictive correlation $\kappa$ is high. At higher $\kappa$, the causal effect of the spurious attribute is noisy i.e. have a non-zero value. In spite of this noise and slight difference in the causal effect estimate of both the estimators in the first step, in the second stage our method is able to perform equally well for both the estimators used (see Fig 13, 14, and 15 in the appendix for the individual performance of our method when using the Direct and Riesz estimator in stage 1). Given these observations, we conjecture that our method should be robust to the choice of causal effect estimator used in stage 1. Theorem 3.1 also states that we only need the ranking of the causal effect of causal and spurious attributes to be correct in order to learn an optimal classifier further demonstrating the robustness of our method to the noise in the causal effect estimation that could happen due to choice of the estimator. See “**Robustness to noise in the estimated causal effect**” in the global comment of the rebuttal for further discussion.
>
>
>
> > **Q2: How was the counterfactual distribution learned (mentioned in line 176, equation (2))**
>
> A: We have provided a detailed answer titled “**Learning counterfactual distribution**” in the global comment of the rebuttal.
>
>
> > **Q3: What would happen if some causal features were missing.**
>
> A: There are two potential problems that could happen:
> 1. **Drop in the overall accuracy**: It is possible that since some causal features were missing, spurious attributes may have been providing the missing information/compensating for the drop in the accuracy of a learned classifier. Now, if we remove those spurious features, the overall accuracy might drop.
> 2. **Identifiability of causal effect**: Given an attribute, we find the causal effect of the attribute on the task label $y$. For estimating the causal effect, we use a causal effect estimator that may need access to other covariates/attributes for correctly estimating the causal effect. In case some of the other attributes (causal or spurious) are missing, it might affect the performance of the causal effect estimator in Stage 1. That said, we have shown that our method is robust to the noise in the estimation of the causal effect in Stage 1 (see “**Robustness to Noise in the estimated causal effect**” discussion in the global comment of the rebuttal for further detail).
>
> We will add a brief discussion on this in the camera-ready version if accepted.

---

### Official Review · Reviewer_quUc · 2023-07-04

**Soundness:** 3 good
**Presentation:** 4 excellent
**Contribution:** 3 good
**Rating:** 5
**Confidence:** 4

**Summary:**

The authors propose to reduce the effect of spurious attributes on the classification of, say, images, by first estimating the true causal effects and then regularizing the effect of spurious attributes to be closer to the estimated effects. They show, in a theoretical scenario, that this approach is sound and robust to noisy estimates in the first stage.

**Strengths:**

- The paper is clearly written, and tackles an important problem of reducing the effects of spuriously correlated variables on prediction tasks.
- The theoretical results are interesting.


**Weaknesses:**

- The proof sketches do not currently give any intuition about the shape of the proofs, the simply restate what has to be proved.
- It is unclear what the benefit of using the authors' approach is over doing the same first stage and discarding the spurious attributes.
- If the causal effect estimation relies on knowledge about the causal graph (DGP-1, DGP-2), then why do we need to do all this? Should it not already be known which attributes are causal and which are spurious through graphical criteria?


**Questions:**

- Given that the entire point of using regularization instead of feature selection is to ensure that causal attributes are not accidentally discarded due to our uncertainty as to which attributes are spurious, is it reasonable to assume in Section 3.3 that the representation disentangles between the two categories of attributes?
- In particular, is there a "smart" encoder architecture which would ensure that the representation correctly disentangles with "high probability"?
- Similarly, since the penalty weights $\lambda$ are specifically chosen to satisfy a certain inequality, then if our causal estimates are wrong, what would be the benefit of using regularisation?
- Overall, what are the upsides and downsides of using regularization over feature selection?
- Can the authors explain what prevents the desired classifier from being optimal in Theorem 3.1 when the representation is of higher dimension?
- Perhaps relatedly, could a consistent estimator of causal feature effect for step 1 be combined with a consistent step 2 (e.g., in the setting of Theorem 3.1) to provide a consistent estimate in which the desired classifier is indeed the global optimum (also in the higher dimensional case)?



**Limitations:**

The authors do not discuss the limitations of their approach in detail.

---

> ### Author Rebuttal · Authors · 2023-08-09
>
> > **W1: The proof sketches do not currently give any intuition about the shape of the proofs, they simply restate what has to be proved.**
>
> A: We thank the reviewer for pointing this out. We will update the paper to include a more technical proof sketch.
>
>
>
>
> > **W2: It is unclear what the benefit of using the authors' approach is over doing the same first stage and discarding the spurious attributes.**
>
> > **Q4: Overall, what are the upsides and downsides of using regularization over feature selection?**
>
> A: Compared to doing the first stage and discarding spurious attributes, there are two key benefits of using regularization over feature selection:
> 1. **Noise in estimating the causal vs spurious attribute (Lines 38-39)**: The causal effect of attributes obtained in the first step could be noisy due to various reasons like non-identifiability of causal effect, etc. Thus it is possible that the causal effect for the spurious attribute could be non-zero. Simple thresholding techniques might lead to some false positive or false negative spurious attributes. As a result, this approach of estimating the causal effect and then discarding the spurious attribute may not work. See “**Robustness to noise in the estimated causal effect**” in the global comment of the rebuttal for discussion on the robustness of our method to noise.
> 2. **Attributes could have a continuous range of causal effects**: An example to justify this is the MNLI dataset [1], a popular dataset used in the spurious correlation literature. Given a premise and a hypothesis, the task is to predict whether the hypothesis entails, contradicts, or is neutral with respect to the premise. It has been observed that the presence of negation words (“nobody”, “no”, “never”, and “nothing”) is correlated with the contradiction label. At the same time, these features are sometimes important for prediction since they change the meaning of the sentence and thus have an aggregate non-zero causal effect. Thus regularizing with the correct causal effect is important to learn a correct classifier.
>
> [1] Adina Williams, Nikita Nangia, Samuel R. Bowman, 2018
>
>
>
>
> > **W3: If the causal effect estimation relies on knowledge about the causal graph (DGP-1, DGP-2), then why do we need to do all this?** (part of the remark left out for brevity)
>
> A: We politely disagree with this comment. In summary, our method doesn’t assume the knowledge of underlying DGP. We consider two commonly occurring DGPs (DGP1 and DGP2) only to illustrate that causal effects are only identifiable in certain causal graphs.
> Also, the DGP considered in our causal graph is general/has unknown edges. There are certain edges that are unknown (shown by the red color in the graph) and thus one cannot use the graphical criteria to find the causal and spurious attribute. Please see the “**Our method does not assume knowledge of the underlying DGP**” discussion in the global comment of the rebuttal.
>
>
>
>
>
> > **Q1 and Q2: Is it reasonable to assume in Section 3.3 that the representation disentangles between the two categories of attributes?** (part of the remark left out for brevity)
>
> A: We agree with the reviewer’s comment that it is not reasonable to assume that the latent space will always disentangle. But, we only make this assumption for creating a simple setup where we could theoretically study the effectiveness of our method in the presence of noisy estimates of the causal effects. But empirically, we don't make the assumption that the encoder disentangles the latent space into such attributes and show that our method is still effective and robust to noisy estimates of the causal effects.
>
>
> > **Q3: Similarly, since the penalty weights $\lambda$ are specifically chosen to satisfy a certain inequality, then if our causal estimates are wrong, what would be the benefit of using regularisation?**
>
> A: We believe there is a slight misunderstanding in the interpretation of Theorem 3.1. The theorem states that we don’t need a precise estimate of the causal effect of the attributes for learning the correct classifier. Rather, as long as the penalty weights $\lambda$ (inverse of the estimated causal effect of the attribute, Line 199) satisfy the specified inequality, our method will select the correct classifier. As long as the estimated ranking of causal effects is correct, our method will select the correct classifier (Remark on line 223). This shows that our method is robust to the error in the estimates of the causal effect of the attributes.
>
>
>
>
>
> > **Q5 and Q6: Can the authors explain what prevents the desired classifier from being optimal in Theorem 3.1 when the representation is of higher dimension?** (part of the remark left out for brevity)
>
> A: We believe there is a slight misunderstanding in the interpretation of Part 2 of Theorem 3.1. We only make the assumption that there is **one high-dimensional causal** and **one high-dimensional spurious** feature in the representation for ease of exposition. In case, there are multiple high-dimensional causal and spurious attributes, all the different causal attributes can be concatenated into a single bigger high-dimensional vector and similarly for multiple spurious attributes and the same proof will hold. We leave the extension of our current proof for the case with multiple high-dimensional causal and spurious attributes (each with different causal effect) for future work.
>
>
>
>
> > **Limitations: The authors do not discuss the limitations of their approach in detail.**
>
> A: We thank the reviewer for pointing this out and we will elaborate on them further in the camera-ready version. Here is the summary:
> 1. Our method only works when the underlying dataset-generating process is causal.
> 2. We can guarantee the identification of causal effects only in certain kinds of DGPs. For a general DGP, while we explore this limitation both theoretically (Theorem 3.1) and empirically (Syn-Text-Unobs-Conf), a more extensive study of robustness is needed.

---

> > ### Comment · Reviewer_quUc · 2023-08-14
> > **Response to rebuttal**
> >
> > Responses to W3 and Q1+2: While the empirical results here show good performance, is there any way of ascertaining how well the provided estimates perform on future datasets? Since in general neither the DGP nor the disentanglement assumptions will hold, can we tell how well our correction for spurious features performs when we don't have access to the ground truth data? Relatedly, if the identifiability assumptions *do* hold, can we verify this after having run the algorithm?

---

> > > ### Author Response · Authors · 2023-08-14
> > >
> > > > C1: While the empirical results here show good performance, is there any way of ascertaining how well the provided estimates perform on future datasets? Since in general neither the DGP nor the disentanglement assumptions will hold, can we tell how well our correction for spurious features performs when we don't have access to the ground truth data?
> > >
> > > We thank the reviewer for raising these thoughtful questions. We agree with the reviewer that in general both the identifiability and disentanglement assumptions may not hold. Below we give pointers that can help ascertain the kind of datasets where the theoretical assumptions are expected to hold.
> > >
> > > **Identification assumption**:
> > > 1. **Datasets built using human annotation where input X is sufficient for predicting label (DGP-1)**: For certain datasets like CivilComments and Twitter-AAE (considered in this work), the **data-collection process** tells us that the task labels were created by showing the input sentence to human annotators or through a deterministic function given the input and thus resembles the DGP-1 considered in this work. In fact, DGP-1 is a common data-generating process covering datasets with human-annotated labels where the task label Y is generated using the observed input X and this input is sufficient for the prediction of the label. Thus, **for many-real world NLP datasets that use human annotation or deterministic functions to create ground-truth labels using input**, it is indeed possible to **estimate the correct causal effect** using the estimator given in the proof of Proposition 3.1 (part 1).
> > > 2. **Datasets with multiple correlated attributes where the label depends only on a subset of attributes (DGP-2)**: Such datasets are common in computer vision, e.g., the MNIST dataset considered in our work. Here there may be multiple attributes that are correlated with each other. They can be divided into two subsets: the causal attributes (e.g., shape) that affect the task label Y and the spurious/”style” attributes (e.g., color, rotation) that do not affect Y. This corresponds to DGP-2 from Figure 2 and common in image datasets (see, e.g., Von Kugelgen et al. (2021)). Here too, identification of attributes’ effect is possible whenever the correlated causal and spurious attributes are observed. Note that other independent causal attributes can be unobserved.
> > >
> > > More generally, given a dataset with a correlation between spurious attributes and the task label, we conjecture based on Prop. 3.1 that as long as all the causal attributes correlated with the spurious attribute are observed, then the effect of attributes is identified (causal attributes refer to the attributes that cause Y; causal attributes independent of the spurious attribute can be unobserved). So given a future dataset, if we can determine such a property based on domain knowledge, then the theoretical applicability of our method can be ascertained.
> > >
> > > **Disentanglement assumption**:
> > >
> > > In addition, we would like to emphasize that the **identification of the correct causal effect is a more fundamental assumption** for our method to work as desired whereas the disentanglement assumption is merely made for the convenience of the proof.  Neither the general-purpose causal effect estimation algorithm used in the first step (e.g. RieszNet or Direct Estimator) nor our regularization term in the second step depend on the disentanglement assumption.
> > >
> > > Von Kügelgen, Julius, et al. "Self-supervised learning with data augmentations provably isolate content from style." Advances in neural information processing systems 34 (2021): 16451-16467.
> > >
> > >
> > > > C2: Relatedly, if the identifiability assumptions do hold, can we verify this after having run the algorithm?
> > >
> > > We again thank the reviewer for this thoughtful question. In general, verifying identification from observational data is difficult and this challenge exists for any causal effect inference task.  However, if we assume access to data from another domain that varies the correlation of spurious attribute and the task label, it may be possible to check whether our method recovers the correct non-spurious representation for predicting Y. **The test is as follows**: If the identified effect is correct, then our model’s prediction accuracy should stay invariant across the original and new domains (assuming that the noise distribution stays constant). That said, we want to emphasize that the power of such tests depends on the quality of the new domain sampled: **in general, it may be possible to reject some bad models but verifying identifiability from data alone is a rich, open question**.

---

### Official Review · Reviewer_pmkx · 2023-07-06

**Soundness:** 4 excellent
**Presentation:** 4 excellent
**Contribution:** 4 excellent
**Rating:** 7
**Confidence:** 4

**Summary:**

The paper proposes a new method to detect and remove spurious attributes. First, the paper gives the sufficient conditions that are needed for estimating the causal effects with theoretical proof. To detect the spurious attribute, the proposed method estimates the causal effect based on a deep learning-based estimator [5]. To mitigate the identified spurious attributes, the method adds a regularization term that aims at matching the classifier’s output with the estimated causal effect of an attribute on the label (Eq. 2). The paper further proves that the proposed regularization term’s robustness to noise in causal effect estimation given that the ranking of estimated treatment effect is correct. The experiments are conducted on three datasets, demonstrating the effectiveness of the proposed method.

**Strengths:**

1. The paper is well-written and easy to follow.
2. The paper presents insightful theoretical proof.
3. The proposed method achieves comparable or better empirical results than other methods.


**Weaknesses:**

**[Discussion or Comparison with Other Methods]** The paper fails to discuss or compare with other methods of identifying spurious attributes. EIIL [31] maximizes the IRM objective to infer environments. DebiAN [32] uses the violation of the fairness criterion to partition data into spurious subgroups. Domino [33] proposes error-aware clustering to detect spurious correlations. JTT [34] regards incorrectly predicted samples to detect spurious correlation.

**References**

[31] Elliot Creager, Joern-Henrik Jacobsen, and Richard Zemel, “Environment Inference for Invariant Learning,” in ICML, 2021.

[32] Zhiheng Li, Anthony Hoogs, and Chenliang Xu, “Discover and Mitigate Unknown Biases with Debiasing Alternate Networks,” in ECCV, 2022.

[33] Sabri Eyuboglu, Maya Varma, Khaled Kamal Saab, Jean-Benoit Delbrouck, Christopher Lee-Messer, Jared Dunnmon, James Zou, and Christopher Re, “Domino: Discovering Systematic Errors with Cross-Modal Embeddings,” in ICLR, 2022.

[34] Evan Z. Liu, Behzad Haghgoo, Annie S. Chen, Aditi Raghunathan, Pang Wei Koh, Shiori Sagawa, Percy Liang, and Chelsea Finn, “Just Train Twice: Improving Group Robustness without Training Group Information,” in ICML, 2021.

**Questions:**

In the rebuttal, I expect the authors to address my concerns (listed in the “weaknesses” section) by answering the following questions:

1. Add discussion or comparison with other related works [31-34]
2. Will the code be released for better reproducibility?


**Limitations:**

The paper has adequately addressed the limitations (Section 6). From my perspective, the paper does not have a potential negative societal impact.

---

> ### Author Rebuttal · Authors · 2023-08-09
>
> > **Q1: Add discussion or comparison with other related works**
>
> A: We thank the reviewer for pointing this out. Given limited time and computation constraints, we have added results on three new baselines (JTT[34], EIIL[31], and IRM [35]) on two datasets – Syn-Text-Unobs-Conf and  MNIST34 dataset considered in this work. In summary, our method outperforms all the baselines considered on both the metrics of interest – Average Group Accuracy and  $\Delta$Prob. See the “**Results on Additional Baselines**” discussion in the global comment and Fig 1a and 1b in the uploaded rebuttal document (in the global comment) for detailed discussion.
>
> DebiAN [32] proposes an alternating framework – Discoverer to find the biased subgroups and Classifier to train a model such that the bias discovered by Discoverer is fixed. This method bears similarity to the EIIL [31] framework (added as a baseline in our work) where they use IRMv1 penalty to automatically find the biased subgroup instead of group fairness violation metrics considered in this work. Though the EIIL framework doesn’t use the alternating strategy to further refine the model. In future work, it would be interesting to study the connection between these invariance or fairness constraints and the corresponding causal effect of attributes. Next, they use reweighting instead of regularization to mitigate the bias in the model. In future work, we can also explore reweighting-based techniques, such as inverse propensity weighing, for estimating the causal effect.
>
> Domino [33] mentioned by the reviewer only focuses on discovering the biases and provides no method to fix them or learn an unbiased model whereas our work aims to do both the task – discover and learn an unbiased model.
>
>
>
>
> > **Q2: Will the code be released for better reproducibility?**
>
> A: Yes, we will release the code with the camera-ready version of our paper.
>
>
>
>
> **References**
>
>
> [33] Discover and Mitigate Unknown Biases with Debiasing Alternate Networks, ECCV 2022
>
> [34] Just Train Twice: Improving Group Robustness without Training Group Information, ICML 2021
>
> [31] Environment Inference for Invariant Learning, ICML 2021
>
> [35] Invariant Risk Minimization, 2020

---

> > ### Comment · Reviewer_pmkx · 2023-08-13
> >
> > I have read the authors' responses and other reviewers' comments. The response addresses my concern. I raise my rating to "Accept." I encourage the authors to add the response to the final version.

---

### Official Review · Reviewer_nk8U · 2023-07-08

**Soundness:** 3 good
**Presentation:** 3 good
**Contribution:** 3 good
**Rating:** 5
**Confidence:** 3

**Summary:**

The authors study the problem of learning under the presence of spurious correlations, given multiple attributes, some of which may be spurious. They first propose three causal graphs to represent the data generating process, and show that two them allow for identification of the causal effect of the attributes on the label. They propose a method to identify spurious attributes by computing the causal effect of each attribute on the label, under two possible causal graphs, and then regularizing the model based on the magnitude of this treatment effect. They show that their method beats the baselines on three datasets.

**Strengths:**

- The paper tackles an important and well-studied problem in machine learning.
- The paper is generally well-written and easy to follow.
- The paper is grounded in solid causality theory.

**Weaknesses:**

1. The authors demonstrate theoretical guarantees for two causal graphs shown (DGP-1 and DGP-2). In practice, assuming the causal structure seems like a fairly strong assumption, and the authors do not really examine what happens when the graph is mis-specified. For example, does the method still work if the true data generating process is anti-causal?

2. In order to compute the regularization term, the authors require a sample from the counterfactual distribution. Learning a model that can generate such counterfactual samples seems difficult. In their real-world dataset example, the authors use GPT-3 to generate such counterfactuals. Having access to an LLM seems like it should be outside the scope of the problem, as the authors could have also just asked GPT-3 to predict the sentiment directly (and likely get decent performance). The authors should address (empirically) how to generate these counterfactuals for other text and image datasets, and how the quality of this model impacts their method.

3. The empirical evaluations feel lacking to me. For example, the authors do not evaluate on typical spurious correlation datasets such as Waterbirds, CelebA, or MNLI. In particular, CelebA contains many attribute fields which would be ideal to demonstrate their method.

4. The authors should empirically examine the case where there are multiple spurious attributes, which may be inter-correlated. Does computing the treatment effect of each attribute and then regularizing each one separately make some assumption about the independence of the spurious attributes?

5. The authors should show the results of a few more baseline methods that are popular in the spurious correlation literature, such as JTT [1] (which also does not require knowledge of the attributes), or GroupDRO [2] as an upper bound.

[1] https://arxiv.org/abs/2107.09044

[2] https://arxiv.org/abs/1911.08731

**Questions:**

Please address the weaknesses above.

**Limitations:**

The authors have adequately addressed the limitations.

---

> ### Author Rebuttal · Authors · 2023-08-09
>
> > **W1: The authors demonstrate theoretical guarantees for two causal graphs shown (DGP-1 and DGP-2).** (part of the remark left out for brevity)
>
> A1: We politely disagree with the reviewer's comment. In summary, our method does not assume the knowledge of underlying DGP and thus is invariant to any misspecification in the graph. We consider certain illustrative DGPs (DGP1 and DGP2) to show that causal effects are only identifiable in certain causal graphs. In particular, DGP1 is fairly general and covers a wide range of real-world datasets where the labels are generated by manual annotation and the input contains all the attributes sufficient to allow for such labeling without any external knowledge. But in our empirical experiments, we use a general-purpose causal effect estimator (Direct and Riesz) that doesn’t assume the knowledge of the underlying DGP of the dataset. Next, we go on to show that even when the causal effect is not identifiable (eg. in DGP3 and experimentally simulated in the Syn-Text-Unobs-Conf dataset, see Section 4.1 and Appendix E.1) our method is able to perform significantly better than all the considered baselines (results in Section 4.3 and Appendix F). Please see the “Our method does not assume knowledge of the underlying DGP” discussion in the global comment (response to all reviewers) of the rebuttal for a detailed discussion.
>
> Will our method work in the anti-causal setting? In an anti-causal setting, the causal effect of all attributes is zero by definition. Since our method depends on computing the causal effect of different attributes, it is not guaranteed to work for an anti-causal setting.
>
>
>
> > **W2: In order to compute the regularization term, the authors require a sample from the counterfactual distribution.** (part of the remark left for brevity)
>
> A2:  We agree with the reviewer’s critique that our method requires access to the counterfactual distribution and the quality of these counterfactuals could impact our method. However, we have addressed this concern in the paper with experiments on datasets with and without access to Oracle counterfactual distributions like LLMs. For a summary, see the “Learning counterfactual distribution” discussion in the global comment of the rebuttal.
> In addition, a common deployment goal is to build a model with efficient inference at test time. In such a case, it is reasonable to use an expensive model like GPT3 for generating counterfactuals during training and build a smaller, efficient prediction model for inference. Using GPT directly for real-world use cases will have scalability issues.
>
>
>
>
> > **W3: The empirical evaluations feel lacking to me. For example, the authors do not evaluate on typical spurious correlation datasets such as Waterbirds, CelebA, or MNLI.** (part of the remark left out for brevity)
>
> A3: In the appendix (E.1) we evaluated our method on three different attributes of the Civil-comments dataset, another real-world dataset considered in the spurious correlation literature [1], and part of the challenging WILDS benchmark [4] for out-of-distribution generalization. This dataset also has multiple spurious attributes out of which we subsample three – gender, race, and religion – and evaluate our method on all three attributes (see Section F.1, F.2, Table 10, and Fig 9 in the appendix for detailed discussion). These attributes are present in the sentence in a subtle way and thus cannot be easily removed by simply discarding a few words from the sentence. To summarize the result,
> Causal effect Estimation (Stage 1): Table 10 in the appendix summarizes the estimated causal effect for each of the attributes of this dataset. Since this is a real-world dataset, the true causal effect for each of the attributes is unknown but we expect them to be $0$ since these sensitive attributes should not affect the task label $y$ (toxicity of a sentence). We observe that both the causal effect estimators (Riesz and Direct) perform similarly and have a causal effect close to $0$.
> Overall Performance (Stage 2): Fig 9a, 9b, and 9c in the appendix show the overall performance on all different attributes of this dataset. Overall, we observe similar results as the TwitterAAE dataset where our method performs comparably to the baselines on the Average Group Accuracy but our method has close to optimum $Delta$Prob i.e. 0.
>
>
>
> > **W4: The authors should empirically examine the case where there are multiple spurious attributes, which may be inter-correlated.** (part of the remark left out for brevity)
>
> A4: We thank the reviewer for this suggestion. If multiple features are dependent it is possible that the regularization step might become difficult since the regularizer might need to satisfy the causal effect constraints among the features (which is unavailable). That said, given the limited time and computing resources, we plan to include this experiment in the camera-ready version of the paper if accepted.
>
>
>
>
>
> > **W5: The authors should show the results of a few more baseline methods that are popular in the spurious correlation literature, such as JTT [1], or GroupDRO [2] as an upper bound.**
>
> A5: We thank the reviewer for pointing this out. We have added results on three new baselines (JTT[1], EIIL[2], and IRM [3]) on two datasets – Syn-Text-Unobs-Conf and  MNIST34 dataset considered in this work. We find that our method outperforms all the baselines considered on both the metrics of interest – Average Group Accuracy and  $\Delta$Prob. See the “Results on Additional Baselines” discussion in the global comment and Fig 1a and 1b in the uploaded rebuttal document for detailed discussion.
>
>
>
>
>
> **References**
>
> [1] Just Train Twice: Improving Group Robustness without Training Group Information, ICML 2021
>
> [2] Environment Inference for Invariant Learning, ICML 2021
>
> [3] Invariant Risk Minimization, 2020,
>
> [4] WILDS: A Benchmark of in-the-Wild Distribution Shifts

---

> > ### Comment · Reviewer_nk8U · 2023-08-16
> >
> > Thank you for the response. Most of my concerns and questions have been addressed, and I think the paper has been improved with the new experiments that the authors have run. I have increased my score as a result. I would encourage the authors to include experiments on additional datasets in their revision, especially MNLI, which the authors reference in their response to Reviewer quUc as an example of a dataset where spurious features may have non-zero causal effect (this concept has also been discussed in [1]).
> >
> > [1] https://arxiv.org/pdf/2210.14011.pdf

---

### Author Rebuttal · Authors · 2023-08-09


> **Our method does not assume knowledge of the underlying DGP. We describe three commonly occurring DGPs only to illustrate identification properties** (Reviewer nk8u, quUc):


**Identifiability of causal effect**: We have listed different DGPs (data-generating processes) common in the real world and shown that causal effects are identifiable for some of them. However, our method doesn't require the knowledge of underlying DGP (see below). We agree that for different DGPs, there could be different correct causal effect estimators, but they don't require the knowledge of whether the attribute for which we are estimating the causal effect is causal or spurious. For example in DGP1, we don’t know beforehand whether A is causal or spurious (In Fig 2, the red arrow in DGP1 depicts that the edge is unknown) and we can use the causal effect estimator designed in the proof of Claim 1 of Proposition 3.1 to estimate the causal effect of A on X directly (equation 11 in appendix B). If the value of the estimated causal effect under an infinite sample limit is 0, then A is spurious otherwise not.


**Our method doesn’t need knowledge of underlying DGP**:  In our empirical study, we consider different datasets with different underlying DGPs. For example, MNIST34 follows DGP2, real world-datasets TwitterAAE and CivilComments follow DGP1, and Syn-Text-Unobs-Conf follows DGP3 (see Section 4.1). However, our method does not use this knowledge. To demonstrate this, we have used the same causal effect estimator (Riesz and Direct) for all our experiments (see section 4.2).






> **Results on Additional Baselines** [Reviewer nk8u,pmkx]:

We have added additional baselines to compare our method. We have limited our evaluation to two datasets, Syn-Text-Unobs-Conf and MNIST34 due to time and computing resources constraints, and will add the comparison on other datasets in the camera-ready submission if accepted. See Fig 1(a) and Fig 1(b) in the rebuttal document in the global comment for the plots. In summary,
1. **Syn-Text-Unobs-Conf dataset (1a in rebuttal doc)**: We have added three new baselines, JTT[1], EIIL[2], and IRM[3]. Unlike Mouli+CAD, JTT, EIIL, and IRM are able to considerably decrease the $\Delta$Prob (lower is better) while having better Average group accuracy compared to ERM. However, our method (CausalReg) is able to perform better than all the considered baselines on both metrics.
2. **MNIST34 dataset (1b in rebuttal doc)**: Due to time and compute constraints we have only been able to add a comparison to the JTT and IRM baseline for this dataset but plan to add other baselines in the camera-ready submission. Though JTT is able to lower the $\Delta$Prob (lower is better) compared to Mouli+CAD and ERM, our method is able to outperform all the baselines on both metrics.


[1] Just Train Twice: Improving Group Robustness without Training Group Information, ICML 2021

[2] Environment Inference for Invariant Learning, ICML 2021

[3] Invariant Risk Minimization, 2020






> **Robustness to Noise in the estimated causal effect** (Reviewer nk8u, quUc, zxmG):

1. **Theoretical Analysis**: Theorem 3.1 states that we don’t need access to the precise causal effect estimate of an attribute to learn a model robust to spurious correlation. Our method only requires the ranking of the causal and spurious attributes to be correct to learn the desired classifier.
2. **Empirical Analysis**:
    1. **Robustness to varying predictive correlation $\kappa$**: Across datasets, as we increase the predictive correlation ($\kappa$), the causal effect estimate becomes worse (see Table 6-10). In spite of this error in stage 1, our method performs better than the other baselines (see Sections 4.1 and 4.2 in the main paper and Sections F.1 and F.2 in the appendix).
    2. **Sensitivity to noise in estimated causal effect**: To evaluate the sensitivity to the noise, we regularize the model with our method using a spectrum of different causal effects for the spurious attribute to simulate the noise in the estimation step. We observe that for all the datasets, our method is able to perform better than the baselines even with a large noise. See Fig. 10, 11, and 12 in the appendix for a detailed discussion.



> **Learning counterfactual distribution** (Reviewer nk8u, zxmG):

Our method assumes access to examples sampled from a counterfactual distribution (Eq. 2 and Line 176) to train an unbiased model. Empirically, we demonstrate the performance of our method on different types of datasets (synthetic, vision, and text) with access to different qualities of counterfactual distributions. More specifically,
1. **Counterfactuals using GPT3 (Text)**: For the TwitterAAE dataset,  we use GPT3 as the oracle counterfactual distribution (see Table 4 and Section E.1 in the appendix). Given an input sentence, GPT3 is prompted to generate a new sentence where the race-specific attribute is changed from African-American (AAE) to Non-Hispanic White or vice-versa.
2. **Counterfactual distribution using Topic Model (Text)**: In Appendix E.1, we evaluate our method on three different attributes of the real-world CivilComments dataset. For this dataset, instead of using generative models like GPT3, we use a handcrafted list of words to generate the counterfactual w.r.t. three different attributes — religion, race, and gender (Table 5). We remove the words corresponding to each attribute from the sentence to generate the noisy estimate of counterfactual (see Section E.1).
3. **Counterfactuals using deterministic transformation (Image)**: For certain simple attributes in the image dataset like rotation, background color e.t.c. a deterministic transformation could be used to generate the counterfactual images (as done in our experiment on the MNIST dataset, see Section 4 and E.1). For more complicated transformations like changing the subject in the image, it would be interesting to explore existing generative models in future work.

---

### Decision · Program_Chairs · 2023-09-21

**Decision:**

Accept (poster)

**Comment:**

My recommendation is to accept the paper.

The reviewers reached consensus to accept the paper, although some points do require clarification, especially around the role of identification and disentanglement assumptions in the presentation and evaluation of the method. In cases where specific DGP's are assumed as examples, it should be stated explicitly. Further, the clarlfication about which DGP's apply to different real examples ([here](https://openreview.net/forum?id=V5Oh7Aqfft&noteId=Nxq7S5ANRQ)) should be included in the paper. Clearly, there are cases where the DGP would violate the assumptions of the causal effect estimation methods that are used, so _some_ knowledge of the DGP is necessary for the method to operate as expected; this is fine, but needs to be made very clear.

In the proof where disentanglement is assumed, it should be noted whether this assumption is made without loss of generality, or whether the theorem only covers a special case that is being used for motivation for the method, for which the justification is primarily empirical.

Please carefully review the reviewer comments about what aspects of the discussion should be further clarified in the main text of the paper at camera ready time.